# Generalization of neoantigen-based tumor vaccine by delivering peptide-MHC complex via oncolytic virus

Chenyi Wang [ID] [1,2,3,4], Yingjun Shi [ID] [2,3,4], Da Zhang[2,3], Yupeng Sun[2,3], Junjie Xie [ID] [2,3], Bingchen Wu[2,3], Cuilin Zhang[2,3 ✉] & Xiaolong Liu [ID] [1,2,3 ✉]

## Abstract

**Neoantigen vaccine is a promising breakthrough in tumor immunotherapy. However, the application of this highly personalized strategy in the treatment of solid tumors is hindered by several obstacles, including very costly and time-consuming preparation steps, uncertainty in prediction algorithms and tumor heterogeneity. Universalization of neoantigen vaccine is an ideal yet currently unattainable solution to such limitations. To overcome these limitations, we engineered oncolytic viruses co-expressing neoantigens and neoantigen-binding major histocompatibility complex (MHC) molecules to force ectopic delivery of peptide-MHC ligands to T cell receptors (TCRs), enabling specific targeting by neoantigen vaccine-primed host immunity. When integrated with neoantigen vaccination, the engineered viruses exhibited potent cytolytic activity in a variety of tumor models irrespective of the neoantigen expression profiles, eliciting robust systemic antitumor immunity to reject tumor rechallenge and inhibit abscopal tumor growth with a favorable safety profile. Thus, this study provides a powerful approach to enhance the universality and efficacy of neoantigen vaccines, meeting the urgent need for universal neoantigen vaccines in the clinic to facilitate the further development of tumor immunotherapy.**

**Keywords** Oncolytic Virus; Neoantigen Vaccines; Peptide-MHC Complexes; Tumor Immunotherapy
**Subject Categories** Cancer; Immunology

## Introduction

Tumor neoantigens are novel epitopes that arise from various tumor-specific alterations within cancer cells during the transformation process (Xie et al, 2023). With considerable potential to circumvent T cell central tolerance and elicit truly tumor-specific T cell responses, neoantigens have been considered as safe and potent targets for therapeutic cancer vaccines due to their exclusive expression in tumor cells (Lang et al, 2022). However, owing to their random nature of occurrence, neoantigens are highly personalized, posing substantial limitations to widespread accessibility and potentially narrowing the therapeutic time window for neoantigen vaccines in treating tumors (Pearlman et al, 2021). Targeting a small subset of public neoantigens, such as driver gene mutations (Hsiue et al, 2021) or viral oncogenic proteins (Kenter et al, 2009), could theoretically be applied as off-the-shelf universal vaccines to overcome the limitations and benefit a large proportion of patients. However, the discovery of public neoantigens qualified with both universality and immunogenicity is extremely challenging. Meanwhile, in terms of therapeutic efficacy, the application of public neoantigen vaccines is severely limited by the presence of tumor heterogeneity resulting from the rapidly evolving of mutational landscape within tumor tissues. Furthermore, acquired resistance to immunotherapies, as represented by the loss of MHC molecules and clonal neoantigens in tumor cells, is also emerging as a significant clinical obstacle (McGranahan and Swanton, 2019). As an alternative to public neoantigen vaccines, we proposed that the introduction of both neoantigen epitopes and MHC molecules into tumor cells through novel approaches would generalize the application of neoantigen vaccines to extend their therapeutic benefit in the clinic.

Oncolytic viruses (OVs) hold great promise for the treatment of solid tumors, as they could preferentially replicate in tumor cells, cytolytically release tumor-specific antigens, modulate the tumor microenvironment, and systemically trigger host antitumor immunity (Harrington et al, 2019). However, clinical efficacy of OVs as monotherapy remains limited due to several obstacles, such as the induction of neutralizing antibodies, the suppressive tumor microenvironment, off-target sequestration, and viral-immune balancing (Martin and Bell, 2018; Moaven et al, 2021). To potentiate the therapeutic efficacy, various OVs have been exploited as a highly versatile platform for locoregional delivery of immunomodulatory transgenes, including bispecific antibodies (Khalique et al, 2021), bispecific fusion proteins (Huang et al, 2020), full-length antibodies (Xu et al, 2021), cytokines (Xu et al, 2021), chemokines (Li et al, 2011) and immune checkpoint inhibitors (Passaro et al, 2019). Interestingly, Park et al, have recently demonstrated the effective use of vaccinia oncolytic virus to deliver chimeric antigen receptor (CAR) targets (truncated non-signaling variant of CD19) into solid tumors,

[1]College of Chemical Engineering, Fuzhou University, Fuzhou 350108, P. R. China. [2]The United Innovation of Mengchao Hepatobiliary Technology Key Laboratory of Fujian Province, Mengchao Hepatobiliary Hospital of Fujian Medical University, Fuzhou 350025, P. R. China. [3]The Liver Center of Fujian Province, Fujian Medical University, Fuzhou 350025, P. R. China. [4]These authors contributed equally: Chenyi Wang, Yingjun Shi. ✉E-mail: Cuilin_Zhang@fzu.edu.cn; xiaoloong.liu@gmail.com

promoting the intratumoral infiltration of both endogenous T cells and adoptively transferred CD19 CAR T cells as well as the release of intact viral progeny from dying tumor cells (Park et al, 2020). Inspired by this work, we reasoned that engineering OVs to transfer neoantigen peptide-MHC complex would effectively synergize with neoantigen vaccine-primed immunity to amplify antitumor activity.

Serotype 5 adenovirus (Ad5) has long been validated as a promising vector for gene therapy, vaccination, and oncolytic cancer therapy, through extensive testing in numerous animal models and clinical trials (Wold and Toth, 2013). Since single-shot immunization with moderate doses of adenovirus-based vaccines has been reported to be effective in eliciting immune protection (Wu et al, 2020) and potentiating the stemness of neoantigen-specific CD8$^+$ T cells (D'Alise et al, 2022; D'Alise et al, 2019), we produced Ad5-vectored neoantigen vaccines in this study to ensure its early application in tumor therapy. Based on the vaccine vectors, engineered viruses encoding neoantigen peptide-MHC complexes were subsequently developed as vectors for intratumoral (I.T.) virotherapy to integrate with neoantigen vaccination to generalize the applications of neoantigen vaccines. Because of the species-specific limited replication of oncolytic Ad5 (Thomas et al, 2007), for example, with negligible efficiency in mice except for epidermal cells due to unknown mechanism (Blair et al, 1989; Ganly et al, 2000), to clearly demonstrate the feasibility of the integrative immunotherapy, the engineered replication-defective adenoviruses (eAdvs) were mainly applied in mouse cell lines and immuno-competent mouse models, while the engineered oncolytic adeno-viruses (eOAds) were comparatively used in human cell lines and humanized tumor models in immunodeficient mice. Infection with the engineered viruses allowed ectopic reconstitution of neoantigen peptide-MHC complexes, thereby redirecting the cytotoxicity of vaccine-primed T cells to tumor cells beyond the resident cells of the targeted neoantigens. Integrating with the immunization by neoantigen vaccines, the therapeutic potential of the engineered viruses was intensively explored using different tumor models that were established from cell lines with and without the expression of the applied neoantigens. Furthermore, the mechanisms of this integrative therapy were thoroughly investigated through ELISpot assay, multiplex immunofluorescence assay, and immune cell depletion. Thus, this proof-of-concept study offers a viable strategy to overcome major limitations in neoantigen vaccines applications, including broad applicability and tumor heterogeneity, meriting for further exploration in the clinic to improve immunotherapeutic outcomes.

# Results

## Generation and validation of adenovirus-vectored neoantigen vaccines

In an attempt to exemplify the generalization of neoantigen vaccines, 7 validated MHC-I (H-2Kb) binding neoantigens (Chen et al, 2022) (Mapk3-S284F, Lmf1-F523V, Samd91-K752M, Traf7-C403W, Dtnb-K40T, Lbr-A341P, and Ptpn2-I383T) that were obtained from Hepa1-6 cells, a murine hepatoma cell line from the C57L/J mouse, were selected as the immunogens. Respectively, named as NF and NGS, the listed neoantigens concatenated in sequence with furin cleavage sites (RGRKRRS) (Duperret et al, 2019) or non-immunogenic glycine/

serine linkers (GGSGGGGSGG) (Sahin et al, 2017) were synthesized (General Biol, China) and fused with the signal peptide, transmembrane domain and intracellular domain of H-2Kb molecule (Fig. 1A). We further constructed replication-defective recombinant adeno-viruses by subcloning each synthesized gene cassette into a modified adenoviral shuttle plasmid (pDC315-EF1-mCherry), which was generated by introducing an EF-1α promoter-regulated mCherry expression cassette in between the multiple cloning sites (MCS) and the SV40-polyA signal of pDC315 plasmid. Specific for the visualization of neoantigen expression in mammalian cells, alternative plasmids were constructed by attaching multiple Flag tags (3×Flag) to the C terminus of the NF- or NGS-coding sequences in pDC315 plasmid. After the verification of the transgene expression in HEK293T cells by immunofluorescence staining and Western-blot analysis (Fig. 1B; Appendix Fig. S1A) following plasmid transfection, adenoviruses encoding no-tag NF (Adv-NF) or NGS (Adv-NGS) were thereafter packaged and propagated in HEK293A cells. In vitro characterization of Adv-NF and Adv-NGS revealed that the viruses were equivalent to the parental empty adenovirus (Adv-Ctrl) over a range of doses [0.01 to 10 multiplicity of infection (MOI)] in regards to their infection efficiency in mouse dendritic cells (DCs) (Appendix Fig. S1B). Compared to the phosphate-buffered saline (PBS) and Adv-Ctrl groups, infection with Adv-NF and Adv-NGS at a MOI of 2 both resulted in a 2-fold expansion of CD80$^+$CD86$^+$ mouse DCs, demonstrating a potency comparable to that of lipopolysaccharide (LPS) stimulation (Appendix Fig. S1C,D). These findings substantiated the efficacy of Ad5 as a promising vaccine vector for neoantigens.

To further validate the immunogenicity of the adenoviral neoantigen vaccines (Adv-NF and Adv-NGS), IFNγ production was measured by flow cytometry analysis and ELISpot assay to assess neoantigen-specific T cells responses following in vitro antigen presentation by the adenovirus-treated DCs. As demonstrated in the Appendix Fig. S2A,B and Fig. 1C,D, compared to PBS control and Adv-Ctrl, antigen presentation by Adv-NF and Adv-NGS both elicited a significant immune response in splenic mononuclear cells (SMNCs). Based on these results, a cytotoxicity assay was conducted by co-culturing the SMNCs with Hepa1-6 cells to evaluate their neoantigen-specific reactivity post-antigen pre-sentation. Subsequent to the co-culture, the culture medium, adherent remnant tumor cells, and non-adherent cells (comprising DCs, SMNCs and apoptotic tumor cells) were collected for further analysis. The lactate dehydrogenase (LDH) cytotoxicity assay on remnant tumor cells (Fig. 1E) demonstrated that adenovirus-mediated in vitro presentation of the neoantigens via Adv-NF and Adv-NGS, respectively, enabled SMNCs to eliminate 75% and 50% of the co-cultured Hepa1-6 cells. In addition, with the fluorescence imaging in a parallel experiment, where SMNCs were stained with CM-Dil and Hepa1-6 cells were labeled by green fluorescent protein (GFP), the cytotoxicity results were corroborated by the dramatic reduction of the GFP signal in the Adv-NF and Adv-NGS groups (Appendix Fig. S3A). On the other hand, as for T cell activation markers in the non-adherent cells, co-culture with Hepa1-6 cells significantly increased the composition of CD25$^+$ (~10%, Fig. 1F and Appendix Fig. S3B), CD69$^+$ (~10%, Fig. 1G; Appendix Fig. S3B), IFNγ$^+$ (~3–7%, Fig. 1H and Appendix Fig. S3C), and CD8$^+$IFNγ$^+$ (~3–6%, Fig. 1I; Appendix Fig. S3C) cells within the CD3$^+$ subpopulation in Adv-NF and Adv-NGS groups, demonstrating the therapeutic potential of adenovirus-vectored neoantigen vaccines.

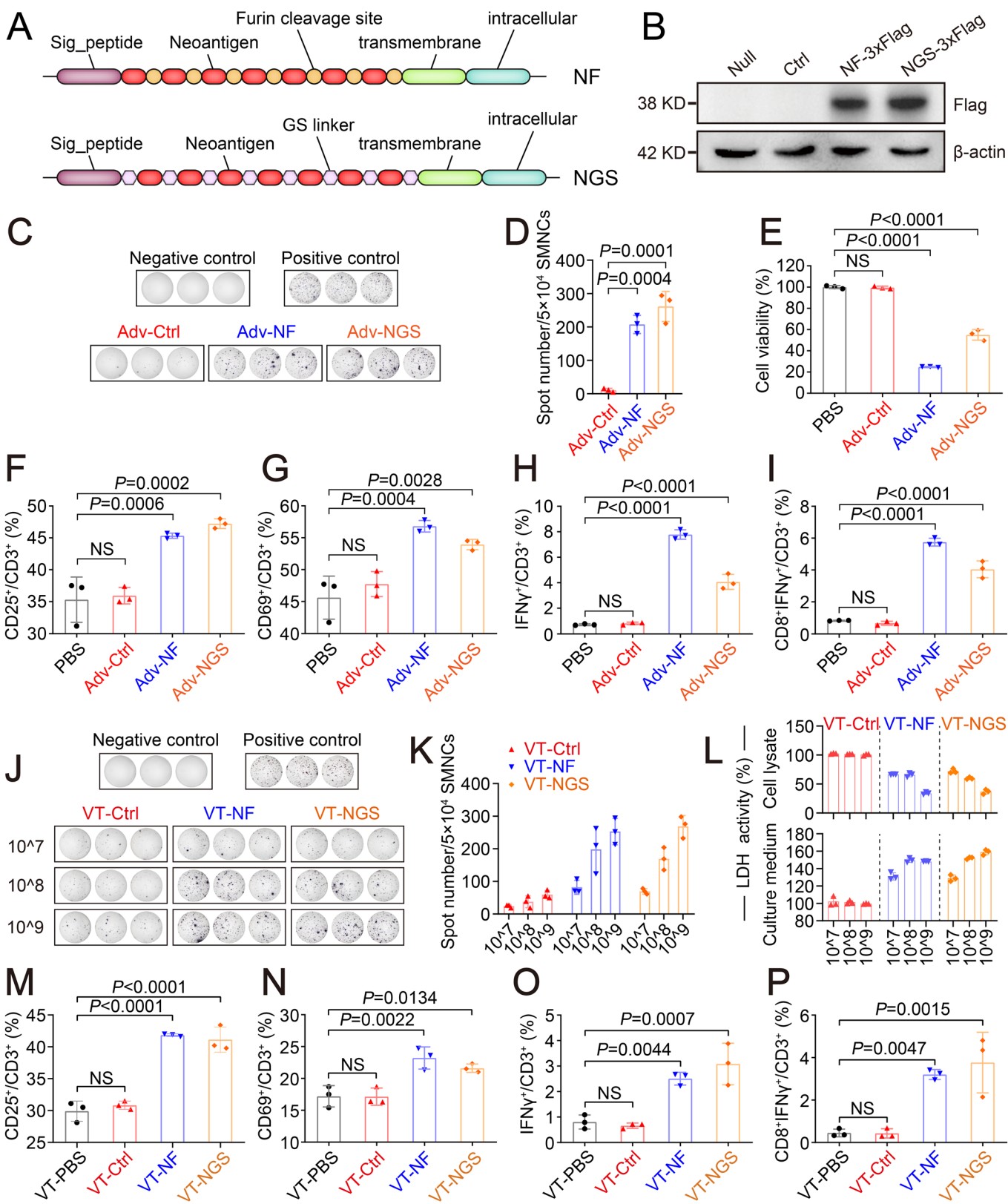

**Figure 1.  Immunization with adenoviral neoantigen vaccines allowed the generation of functional T cells.**

(A) Schematic of adenovirus-vectored neoantigen vaccine. Extracellular domain except signal peptide of mouse MHC-I (H-2Kb) was replaced by neoantigens concatenated with furin cleavage sites (NF) or glycine/serine linkers (NGS). (B) Western-blot analysis demonstrated the overexpression of NF-3×Flag and NGS-3×Flag in HEK293T cells after 24 h of plasmid transfection. (C) ELISpot results showed the in vitro immunogenicity of the adenoviral vaccines ($n = 3$). Infected by 2 MOI indicated adenoviruses for 48 h, $1 \times 10^4$ DCs were co-inoculated into ELISpot plate with $5 \times 10^4$ freshly prepared SMNCs. After 48 h of co-culture, the spots were detected by following the manufacturer's instructions. SMNCs co-cultured with naive DCs were served as the negative control, while antibody-stimulated (2.5 μg/ml anti-CD3ε and 1 μg/ml anti-CD28) SMNCs were considered as the positive control. (D) Statistical analysis of the results in (C) ($n = 3$). (E) LDH assay showed the elimination of Hepa1-6 cells by antigen-presented SMNCs ($n = 3$). After treating with PBS or 2 MOI of indicated adenoviruses (Adv-Ctrl, Adv-NF or Adv-NGS) for 48 h, DCs were co-cultured with freshly prepared SMNCs at a ratio of 1:5 for 72 h. Subsequently, the cells were harvested to co-culture with Hepa1-6 cells at a ratio of 1:3 for 48 h before assessing the LDH activity in the remnant Hepa1-6 cells. P value: Adv-Ctrl, $P = 0.993$; Adv-NF, $P = 1.528 \times 10^{-9}$; Adv-NGS, $P = 8.929 \times 10^{-8}$. (F, G) Quantification of the percentage of activated T cells (CD3$^+$CD25$^+$ in F, CD3$^+$CD69$^+$ in G) after the co-culture with Hepa1-6 cells ($n = 3$). (H, I) Quantification of the percentage of cytolytically active CD3$^+$ T cells (CD3$^+$IFNγ$^+$ in H, CD3$^+$CD8$^+$IFNγ$^+$ in I) ($n = 3$ biological replicates). P value (H): Adv-Ctrl, $P = 0.976$; Adv-NF, $P = 2.175 \times 10^{-8}$; Adv-NGS, $P = 7.290 \times 10^{-6}$. P value (I): Adv-Ctrl, $P = 0.841$; Adv-NF, $P = 9.800 \times 10^{-8}$; Adv-NGS, $P = 2.672 \times 10^{-6}$. (J) ELISpot results showed the in vivo dose-dependent immunogenicity of the adenoviral vaccines. Fourteen days after vaccination with different doses ($2 \times 10^7$ VP, $2 \times 10^8$ VP or $2 \times 10^9$ VP) of adenoviral vaccines via S.C. route, the mice were sacrificed to isolate SMNCs (VT-Ctrl, VT-NF, or VT-NGS) for ELISpot analysis with the neoantigen peptide-pulsed DCs ($n = 3$). Negative control: VT-PBS co-cultured with naive DCs. Positive control: VT-PBS co-cultured with naive DCs in antibody-supplemented (2.5 μg/ml anti-CD3ε and 1 μg/ml anti-CD28) culture medium. (K) Statistical analysis of the results in (J) ($n = 3$). (L) Measurement of LDH activity in remnant tumor cells and culture medium showing the dose-dependent elimination of Hepa1-6 cells by VT-NF or VT-NGS. (M, N) Quantifying the percentage of activated CD3$^+$ T cells (CD3$^+$CD25$^+$ in M, CD3$^+$CD69$^+$ in N) in $10^9$ VP group ($n = 3$). P value (M): Adv-Ctrl, $P = 0.724$; VT-NF, $P = 1.001 \times 10^{-5}$; VT-NGS, $P = 1.578 \times 10^{-5}$. (O, P) Quantifying the percentage of cytolytically active CD3$^+$ T cells (CD3$^+$IFNγ$^+$ in O, CD3$^+$CD8$^+$IFNγ$^+$ in P) in $10^9$ VP group ($n = 3$). Statistical analysis was performed using one-way ANOVA (D–I and M–P). Data are shown as mean ± SD for 3 biological replicates. Source data are available online for this figure.

To evaluate the in vivo immunogenicity of the adenoviral neoantigen vaccines, C57BL/6 mice were immunized with a single dose of $2 \times 10^7$ virus particles (VP) (low dose), $2 \times 10^8$ VP (medium dose), or $2 \times 10^9$ VP (high dose) of Adv-Ctrl (Vac-Ctrl), Adv-NF (Vac-NF) and Adv-NGS (Vac-NGS), respectively, via subcutaneous route (S.C.). The primed SMNCs, correspondingly designated as VT-Ctrl, VT-NF and VT-NGS, were isolated on day 14 post-vaccination and subjected to ELISpot analysis. As shown in Fig. 1J,K, in contrast to Vac-Ctrl, vaccination with both Vac-NF and Vac-NGS dose-dependently enhanced the immune activity of the SMNCs in response to the stimulation by the neoantigen peptide-pulsed DCs. In addition, as exemplified by flow cytometry analysis in high-dose ($10^9$) groups (VT-Ctrl versus VT-NF), the primed SMNCs exhibited clear T cell reactivity towards all the individual neoantigens except Samd91-K752M, revealing Traf7-C403W and Ptpn2-I383T as dominant epitopes (Appendix Fig. S4). Consistent with these results, dose-dependent cytotoxicity was observed in Hepa1-6 cells co-cultured with VT-NF or VT-NGS, whereas incubation with VT-Ctrl did not produce similar results (Fig. 1L). Flow cytometry analysis of the non-adherent cells collected from high-dose groups revealed a significant expansion of CD25$^+$ (~11%, Fig. 1M; Appendix Fig. S3D), CD69$^+$ (~4–6%, Fig. 1N; Appendix Fig. S3D), IFNγ$^+$ (~2–3%, Fig. 1O; Appendix Fig. S3E), and CD8$^+$IFNγ$^+$ (~3–4%, Fig. 1P; Appendix Fig. S3E) cells within the CD3$^+$ subpopulation of VT-NF and VT-NGS groups, indicating cytotoxic activation of the primed cells. Collectively, these results demonstrated the efficacy of Ad5 as a single-shot vaccine vector for neoantigens.

## Transfer of specific cytotoxicity of vaccine-primed T cells by delivering peptide-MHC complexes into tumor cells

The specific recognition of peptide-MHC ligands by TCRs is a key factor in determining the antitumor activity of cytotoxic T lymphocytes (Stone et al, 2009). As evidenced by the development of TCR-engineered T cell (TCR-T) therapy, transferring of TCR specificities could be achieved by transgenic expression of the subunits of TCRs (Dembic et al, 1986). Likewise, ectopic expression of the genetic components of peptide-MHC

complexes has the potential to redirect the cytotoxicity of neoantigen vaccine-primed immunity towards tumor cells beyond the resident cells of neoantigens to broaden the applicability of neoantigen vaccines. Accordingly, we engineered the aforementioned vaccine vectors by subcloning an internal ribosome entry site (IRES) regulated H-2Kb transgenic cassette into the downstream of the neoantigen-coding sequence to generate constructs for the delivery of neoantigen peptide-MHC complexes (NFH and NGSH) (Fig. 2A). Notably, beta-2 microglobulin (B2M) as an essential component within the peptide-MHC complexes was not included in the constructs in the current proof-of-concept study to simplify the demonstration of functionalities, due to its evolutionary conservation allowed cross-species association with heavy chain of MHC-I molecules (Achour et al, 2006) and quantitative real-time PCR (qPCR) validated expression in murine (Hepa1-6, B16F10, MC38 and LLC) and human cell lines (HEK293T, HepG2, SK-Hep-1, and SMMC-7721) (Fig. EV1A). The constructs were subsequently packaged into adenoviruses (Adv-NFH and Adv-NGSH) and kept in −80 °C refrigerator for long-term use. Following in vitro infection, the transgenic expression of the neoantigens and H-2Kb was validated in LLC and B16F10 cells by qPCR analysis and flow cytometry analysis (Figs. 2B and EV1B). Meanwhile, as validated by the results of qPCR analysis and sanger sequencing, all the murine and human cell lines except Hepa1-6 were noncarriers of the neoantigens, and both H-2Kb and the neoantigens were absent in human cell lines (Fig. EV1A,C).

To demonstrate the redirection of cytotoxicity in vaccine-primed T cells, we vaccinated C57BL/6 mice with 3 doses of PBS control, non-adjuvanted neoantigen peptides (PEP), or Poly(I:C)-adjuvanted neoantigen peptides (PEP + ADJ) and acquired the vaccine-primed splenic T cells (VT-PBS, VT-PEP and VT-PEP + ADJ) subsequently. As for cytotoxicity against Hepa1-6 cells, the VT-PEP and VT-PEP + ADJ, respectively, eliminated 22% and 46% of the co-cultured cells in contrast to the cells co-cultured with VT-PBS, indicating functional induction of neoantigen-specific T cells that resulted from different magnitude of vaccination (Fig. 2C). Using pre-infected murine carcinoma (LLC and B16F10) and human (HEK293T and HepG2) cell lines as target cells for cytotoxicity assay, the peptide vaccine primed T cells demonstrated

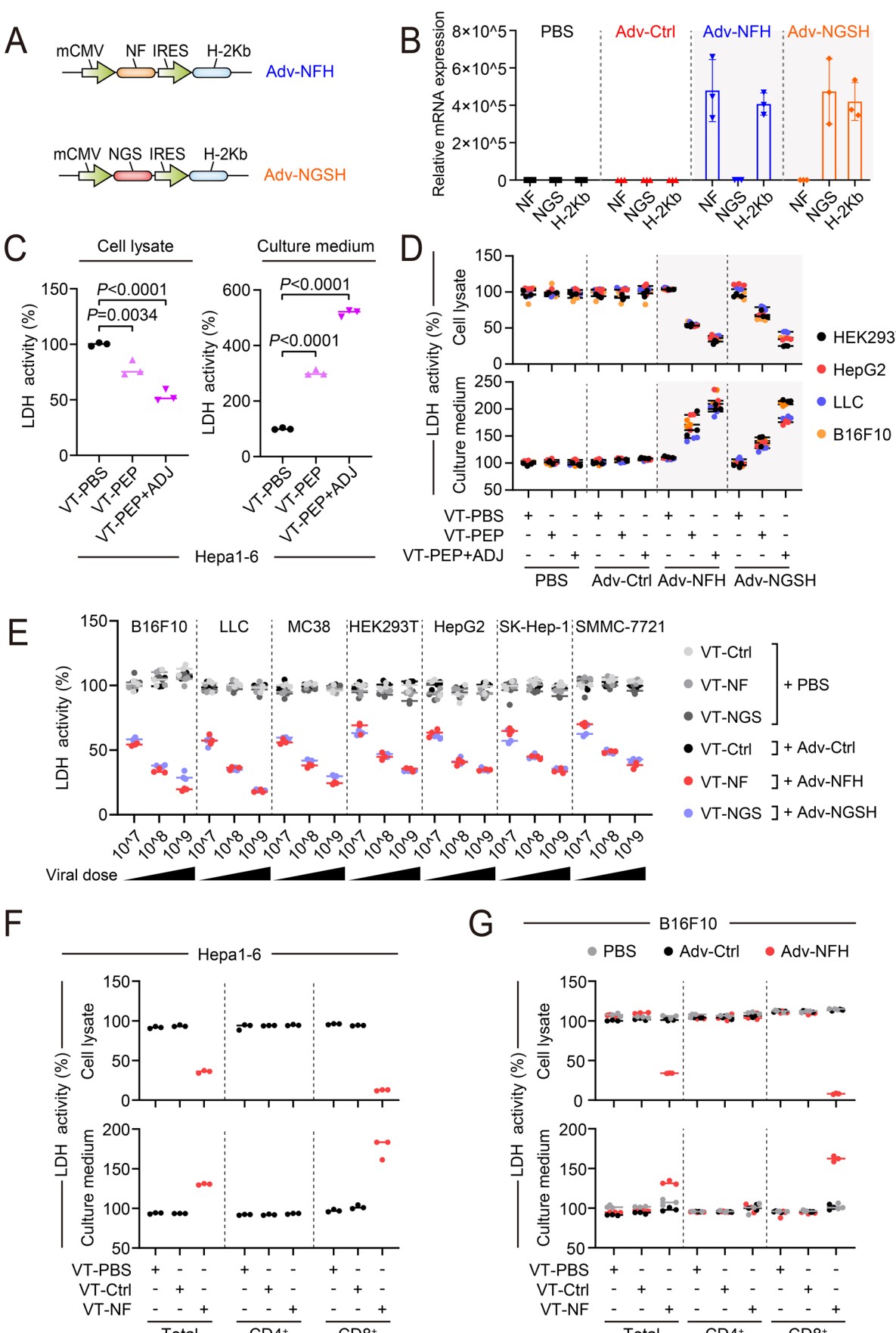

◀ **Figure 2. Transfer of specific cytotoxicity of vaccine-primed T cells by delivering peptide-MHC complexes into tumor cells.**

(A) Schematic of adenoviral vectors engineered for the delivery of neoantigen peptide-MHC complexes. NFH and NGSH, respectively, indicated the co-expression of NF and NGS with H-2Kb. (B) With indicated primers, qPCR analysis in HEK293T cells validated the transgenic expression of the neoantigens and H-2Kb ($n = 3$). Cells were treated with PBS, Adv-Ctrl (1 MOI), Adv-NFH (1 MOI), or Adv-NGSH (1 MOI) for 24 h before cell harvest and mRNA extraction. (C) LDH assay in remnant tumor cells and culture medium demonstrated effective elimination of Hepa1-6 cells with splenic T cells acquired from peptide-vaccinated mice (VT-PEP and VT-PEP + ADJ) in comparison with T cells acquired from saline-treated mice (VT-PBS) ($n = 3$). C57BL/6 mice were vaccinated for 3 doses (day 1, day 5, and day 9) before the harvest of SMNCs (day 14). The peptide powder (2 mg) was dissolved in 50 μl DMSO as stock solution before the preparation of the vaccines, which were prepared in 200 μl PBS and injected via S.C. route. PEP: 700 μg peptides. ADJ: 50 μg poly I:C (1 μg/μl in PBS). $P$ value (cell lysate): VT-PEP + ADJ, $P = 5.422 \times 10^{-5}$. $P$ value (culture medium): VT-PEP, $P = 3.036 \times 10^{-7}$; VT-PEP + ADJ, $P = 4.038 \times 10^{-9}$. (D) Results of LDH assay indicated the redirected cytotoxicity of peptide-vaccinated T cells against tumor cells infected by Adv-NFH or Adv-NGSH ($n = 3$). Following the infection by 1 MOI of indicated viruses for 48 h, HEK293T, HepG2, LLC and B16F10 cells were, respectively, co-cultured with vaccine-primed splenic T cells at a ratio of 1:3 for 48 h. (E) Redirecting dose-dependent cytotoxicity of adenoviral vaccine-primed T cells to murine and human tumor cells by infecting the cells with Adv-NFH or Adv-NGSH ($n = 3$). Mice were vaccinated with single dose of indicated dosage of adenoviruses for 14 d before harvesting SMNCs. Cytotoxicity assay was assessed in the remnant tumor cells by following the experimental procedure in (D). (F, G) Cytotoxicity assay in Hepa1-6 (F) and virus-infected B16F10 cells (G) with total or isolated splenic T cells ($n = 3$). Statistical analysis was performed using one-way ANOVA (C). Data are shown as mean ± SD for 3 biological replicates. Source data are available online for this figure.

similar cytotoxicity against Adv-NFH- and Adv-NGSH-infected target cells, while displaying negligible cytotoxicity toward PBS- or Adv-Ctrl-treated cells (Fig. 2D). Likewise, the adenovirus vaccine-primed T cells (VT-Ctrl, VT-NF or VT-NGS) were used to treat murine (B16F10, LLC, and MC38) and human (HEK293T, HepG2, SK-Hep-1, and SMMC-7721) cell lines that pre-infected with Adv-Ctrl, Adv-NFH or Adv-NGSH. Similar to the results in Fig. 2D, a dose-dependent cytotoxicity was only observed in Adv-NFH or Adv-NGSH infected cells when they were co-cultured with VT-NF or VT-NGS (Figs. 2E and EV2A,B). In addition, the cytotoxicity of VT-NF and VT-NGS was specifically confined in the tumor cells infected with viruses containing neoantigen peptide-MHC transgenes (Adv-NFH and Adv-NGSH) (Fig. EV2C). By co-culturing VT-Ctrl and VT-NF with HepG2 and LLC cells pre-treated by PBS, Adv-Ctrl, Adv-NF, Adv-H2Kb or Adv-NFH, respectively, obvious cytotoxicity was observed only in VT-NF that co-cultured with Adv-NF/Adv-NFH-infected LLC cells or Adv-NFH-infected HepG2 cells (Fig. EV2D). Importantly, VT-NF exhibited more potent cytotoxicity in Adv-NFH-treated (77% elimination) rather than Adv-NF-treated (30% elimination) LLC cells, further validating the functionality and therapeutic benefit of MHC molecules. Collectively, these results confirmed that reconstitution of neoantigen peptide-MHC complexes and redirection of cytotoxicity of neoantigen vaccine-primed T cells can be achieved by the infection of target tumor cells with Adv-NFH or Adv-NGSH.

In addition, vaccine-primed splenic T cells, isolated using the CD4+ or CD8+ T cell isolation kit, were co-cultured with Hepa1-6 or adenovirus-infected B16F10 cells to demonstrate the specificity of redirected cytotoxicity. As shown in Fig. 2F, compared to unseparated VT-NF cells (64% elimination), CD8+ VT-NF cells exhibited superior cytotoxicity against Hepa1-6 cells (87% elimination), while CD4+ VT-NF cells (5% elimination) showed minimal cytotoxicity, similar to VT-PBS (8% elimination) and VT-Ctrl (6% elimination). These results suggested a preferential priming of CD8+ T cells by MHC-I restricted neoantigen vaccines. By repeating the experiments in B16F10 cells, which originally lacked these neoantigens, the cytotoxicity of VT-NF cells in Hepa1-6 cells was reproduced in Adv-NFH-infected but not in PBS- or Adv-Ctrl-infected B16F10 cells, indicating specific recruitment of vaccine-primed CD8+ cells by transferring the neoantigen peptide-MHC complexes to the target tumor cells (Fig. 2G).

## Ectopic delivery of peptide-MHC complexes prevented tumor growth in C57BL/6 mice early immunized with the neoantigen vaccines

To investigate the in vivo effectiveness of the adenovirus-vectored neoantigen vaccines, we preliminarily tested their antitumor effect in early vaccination settings. As shown in Fig. 3A, C57BL/6 mice were, respectively, immunized with a single dose of PBS, Vac-Ctrl, Vac-NF, and Vac-NGS for 14 days before the challenge of Hepa1-6 cells. As a result, this prophylactic intervention protected 100% of Vac-NF and Vac-NGS vaccinated mice, whereas all the mice in control groups (PBS and Vac-Ctrl) developed large tumors (Fig. 3B–E), validating the therapeutic potential of the adenovirus-vectored neoantigen vaccines. Notably, due to its narrow lead over Vac-NGS in effectiveness, Vac-NF was chosen as the therapeutic vaccine for subsequent experiments.

Next, considering the exclusive expression pattern of the neoantigens of Hepa1-6 cells (Figs. 2D,E and EV1C), B16F10 cells originally without these neoantigen expression were infected with Adv-NFH to determine if ectopic delivery of neoantigen peptide-MHC complexes could recapitulate the vaccination-induced protection in C57BL/6 mice (Figs. 3F,G). Similar to the results in the challenge of Hepa1-6 cells, early vaccination of Vac-NF was highly effective in preventing tumor development of Adv-NFH-infected B16F10 cells (Fig. 3H–J). In contrast, tumor cell inoculation resulted in the development of large tumors in all the mice of control groups, including Adv-Ctrl-infected cells in Vac-NF-immunized mice (Vac-NF & Adv-Ctrl), Adv-NFH-infected cells in Vac-Ctrl-immunized mice (Vac-Ctrl & Adv-NFH), and Adv-Ctrl-infected cells in Vac-Ctrl-immunized mice (Vac-Ctrl & Adv-Ctrl). Thus, by demonstrating the utilization of Vac-NF as an effective neoantigen vaccine in preventing tumor establishment of the cells (B16F10) beyond the resident cells of the neoantigens (Hepa1-6), these results raised a possibility to put into a practice of functional integrative therapy that consist of neoantigen vaccination and intratumoral delivery of neoantigen peptide-MHC complexes.

## Exploiting vaccination-induced immunity of neoantigens via integrative therapy effectively suppressed tumor growth in C57BL/6 mice

To evaluate the therapeutic potential of the integrative therapy, considering the species-specifically limited replication of Ad5 in mice, an integrative treatment with S.C. vaccination and I.T.

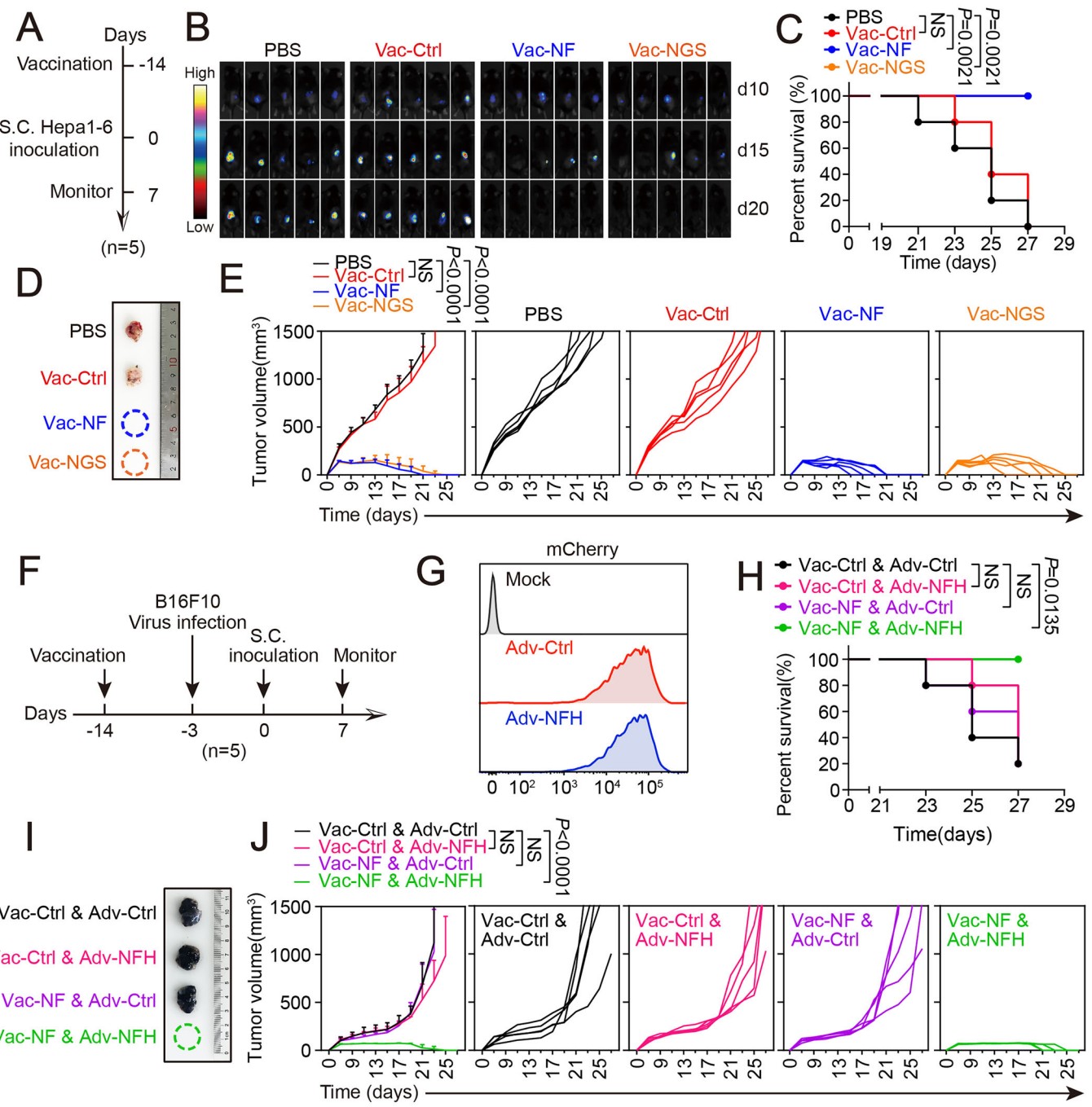

**Figure 3.  Ectopic delivery of peptide-MHC complexes prevented tumor growth in C57BL/6 mice early immunized with the neoantigen vaccines.**

(A) Treatment scheme for subcutaneous Hepa1-6 tumor model in C57BL/6 mice. (B) Noninvasive optical imaging demonstrated the development of Hepa1-6-Luc tumors in mice pre-vaccinated with PBS, Vac-Ctrl ($2 \times 10^9$ VP), Vac-NF ($2 \times 10^9$ VP) or Vac-NGS ($2 \times 10^9$ VP) ($n = 5$ mice per group). (C) Overall survival monitoring during the treatment. (D) Representative image demonstrated tumor volume after sacrifice of the mice. (E) Overall and individual tumor growth curves since the inoculation of Hepa1-6 cells ($n = 5$ mice per group). P value: Vac-Ctrl, $P = 0.398$; Vac-NF, $P = 7.747 \times 10^{-11}$; Vac-NGS, $P = 1.350 \times 10^{-10}$. (F) Treatment scheme for subcutaneous B16F10 tumor model. (G) Flow cytometry analysis evaluated the infection of B16F10 cells by the treatment of PBS (Mock), Adv-Ctrl (10 MOI) or Adv-NFH (10 MOI) before subcutaneous inoculation. (H) Survival monitoring during the treatment ($n = 5$ mice per group). (I) Representative images showing B16F10 tumor volume after sacrifice of the mice. (J) Overall and individual tumor growth curves since the inoculation of B16F10 cells ($n = 5$ mice per group). P value: Vac-Ctrl & Adv-NFH, $P = 0.316$; Vac-NF & Adv-Ctrl, $P = 0.909$; Vac-NF & Adv-NFH, $P = 1.243 \times 10^{-6}$. Statistical analysis was performed with Log-Rank test (C, H) and two-way ANOVA (E, J). Data in (E) and (J) are shown as mean ± SD (error bar). Source data are available online for this figure.

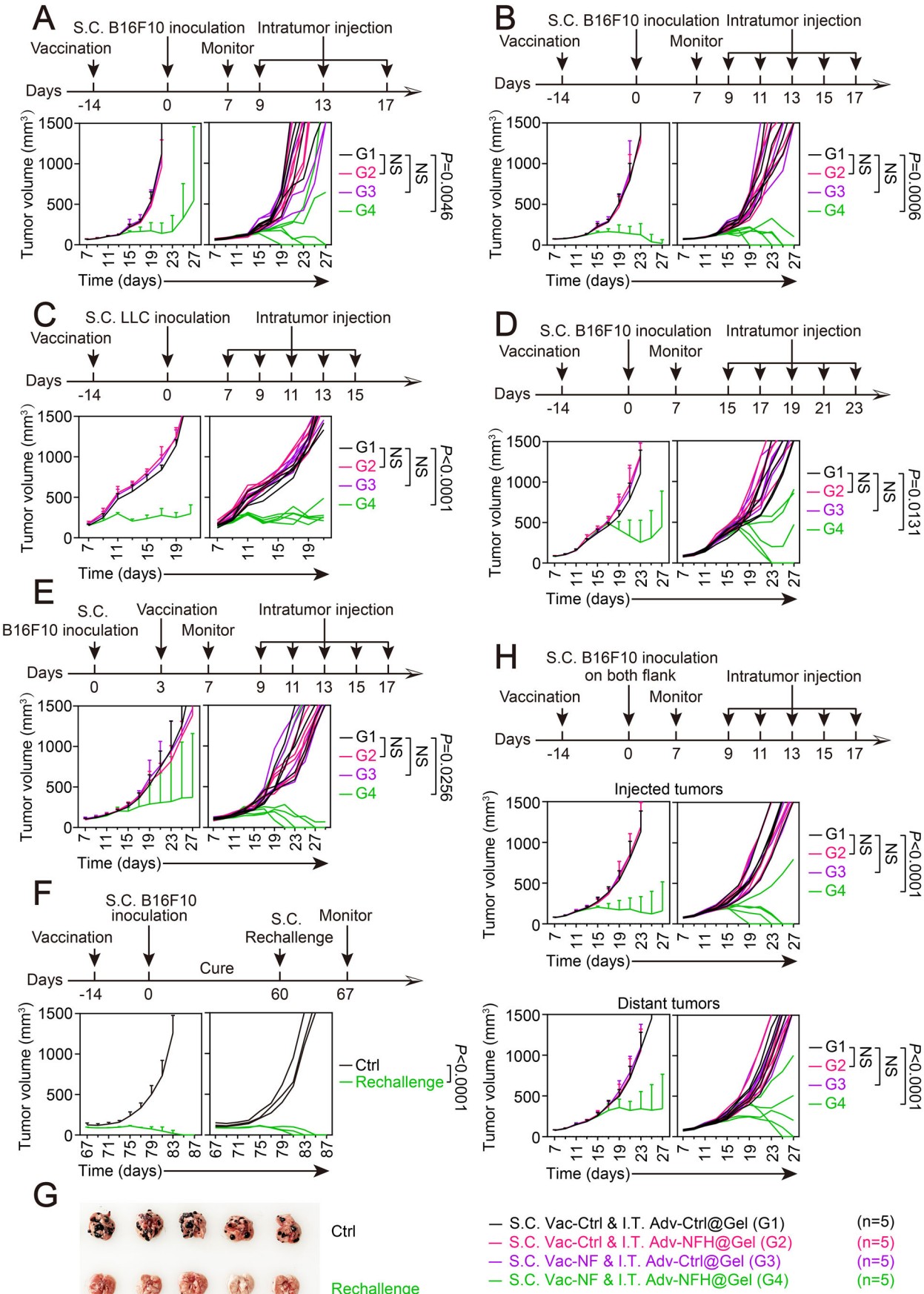

Figure 4.  Exploiting vaccination-induced immunity to neoantigens via integrative therapy effectively suppressed tumor growth in C57BL/6 mice.

(A) Overall and individual tumor growth curves demonstrated an integrative therapy of subcutaneous (S.C.) B16F10 tumor model with S.C. vaccination ($2 \times 10^9$ VP) and 3 doses I.T. injection of silk hydrogel encapsulated replication-defective adenoviruses ($n = 5$ mice per group). Noteworthy, early vaccination was performed before tumor inoculation. G1 indicated the integration between S.C. Vac-Ctrl and I.T. Adv-Ctrl@Gel, G2 represented the integration of S.C. Vac-Ctrl and I.T. Adv-NFH@Gel, G3 indicated the integration of S.C. Vac-NF and I.T. Adv-Ctrl@Gel, and G4 represented the integration between S.C. Vac-NF and I.T. Adv-NFH@Gel. (B) Increasing the doses of I.T. injection improved the therapeutic efficiency of the integrative therapy in B16F10-established tumor model ($n = 5$ mice per group). (C) Integrative therapy in LLC-established tumor model demonstrated the broad applicability ($n = 5$ mice per group). P value: G2, $P = 0.080$; G3, $P = 0.165$; G4, $P = 3.620 \times 10^{-11}$. (D) Efficacy of the integrative therapy in large advanced B16F10 tumor (~400 mm³) model ($n = 5$ mice per group). (E) Clinical simulation by reversing the order for vaccination and tumor inoculation in mouse model indicated the applicability of the integrative therapy ($n = 5$ mice per group). (F) Growth curves of subcutaneously rechallenged B16F10 ($3 \times 10^6$) tumors in cured survivors ($n = 3$ mice per group). (G) Pulmonary metastasis of tail vein injected B16F10 cells ($3 \times 10^6$) in cured survivors ($n = 5$ mice per group). (H) Growth curve of injected and distant tumors in a bilateral tumor model ($n = 5$ mice per group). P value (injected tumor): G2, $P = 0.996$; G3, $P = 0.987$; G4, $P = 8.83 \times 10^{-4}$. P value (distant tumor): G2, $P = 0.99992$; G3, $P = 0.974$; G4, $P = 0.0196$. Statistical analysis was performed using two-way ANOVA (A–F and H). Data are shown as mean ± SD (error bar). Source data are available online for this figure.

injection of replication-defective adenovirus was carried out in immunocompetent C57BL/6 mice. As the first attempt, to meet the requirement of duration for vaccination to take effect, vaccine administration was performed 14 d before the inoculation of tumor cells, while I.T. injection was carried out 9 d after the inoculation. However, initially, no obvious response to the treatment was observed in B16F10 established tumors (Fig. EV3A). According to literature analysis, this might result from virus dissemination in surrounding nontarget tissues and immunogenicity of adenoviruses that limited in vivo persistence (Jung et al, 2017). Since hydrogel encapsulation has emerged as a promising approach for over-coming of these limitations (Jung et al, 2017), an injectable silk protein-based hydrogel (Rockwood et al, 2011; Zhao et al, 2022) due to its biocompatible and biodegradable properties was utilized to encapsulate adenoviruses to enhance the therapeutic efficiency (Fig. EV3B). Remarkably, hydrogel encapsulation significantly increased the infection rate of adenovirus within tumors (Fig. EV3C,D) by maintaining a consistently high viral load (Fig. EV3E) and reducing the off-tumor dissemination (Appendix Fig. S5) and the induction of neutralizing antibodies (Fig. EV3F). This approach proved effective even in mice that had been pre-treated with adenovirus vaccines (Vac-Ctrl or Vac-NF), ensuring intratumoral expression of transgenes delivered by adenoviral vectors (Fig. EV3F,G). Consequently, hydrogel encapsulation was universally adopted for subsequent I.T. injections based on these promising results.

With the hydrogel, as shown in Fig. 4A, the integrative therapy demonstrated remarkable tumor control, enabling complete tumor regression in 3/5 of mice. The therapeutic outcome can be further improved by multiplying the I.T. injections from 3 times to 5 times, leading to complete tumor regression in 4/5 of the treated mice (Fig. 4B). Of note, both results of histological analysis in major organs and biochemical examinations showed no obvious toxicity of the treatment (Appendix Fig. S6A,B). Next, to validate the broad applicability of the integrative therapy, the therapeutic scheme outlined in Fig. 4B was repeated in LLC-established tumor model and encouragingly resulted in effective inhibition of tumor growth across all of the treated mice (Fig. 4C; Appendix Fig. S7). To evaluate the antitumor efficacy of the treatment in large advanced tumors, mice were randomized to receive I.T. injection of hydrogel-encapsulated Adv-NFH until tumor volume reached ~400 mm³ in B16F10-established tumor model. Even in this context, complete tumor regression was achieved in 2/5 of the treated mice (Fig. 4D). In addition, the order for tumor inoculation and vaccination was

reversed by performing vaccine-administration at day 3 after the inoculation of B16F10 cells to mimic actual clinical scenarios. Similarly, Adv-NFH was potently effective in eradiating the established tumors in combination with the vaccination of Vac-NF, with 3/5 of the treated mice attained complete tumor regression (Fig. 4E).

Subsequent to the completion of treatment, animals cured by the integrative therapy were rechallenged with $3 \times 10^6$ B16F10 cells in the contralateral flank, while naive and age-matched mice were employed as controls. Observing for more than 2 weeks, rejection of tumor rechallenge was achieved in all cases (Fig. 4F). In addition, in a parallel experiment, tumor-free mice from previous studies more potently suppressed the lung metastasis than control naive mice after intravenous rechallenge with un-infected B16F10 cells, indicating the induction of immunological memory after initial clearance of the tumors (Fig. 4G). Based on the results, a bilateral tumor model was carried out to assess the induction of abscopal effect. As demonstrated in Fig. 4H, the integrative therapy not only eradicated virus-injected tumors but also significantly suppressed the growth of distant untreated tumors. Furthermore, as shown in Appendix Fig. S8A,B, a B16F10-specific cytotoxicity was observed in the SMNCs of the treated mice by co-culturing with B16F10 cells or stimulating with B16F10-specific neoantigen peptides (Redenti et al, 2024), confirming the establishment of therapy-induced systemic antitumor immune response. Thus, these findings collectively established the therapeutic benefits of the integrative therapy in immunocompetent mice, despite the inability to demonstrate the oncolytic effect of Ad5 in this particular context due to the limited replication of Ad5 in murine cells.

## Tumor infiltration of neoantigen-responsive CD8+ T cells was crucial for the therapeutic benefit of the integrative therapy

Representing the backbone of currently successful cancer immu-notherapy, T cell is a central focus in the battle against cancer due to their capacity in antigen-directed cytotoxicity (Raskov et al, 2021; Waldman et al, 2020). To further elucidate the mechanism of the integrative therapy, we assessed the intratumoral infiltration of T cells following the integrative therapy in a B16F10-established C57BL/6 tumor model. As demonstrated by the results of immunofluorescence staining (Fig. 5A) and flow cytometry analysis (Fig. 5B), increased intratumoral infiltration of both CD4+ and CD8+ T cells was observed in the experimental group (G4: S.C. Vac-NF & I.T. Adv-NFH@Gel),

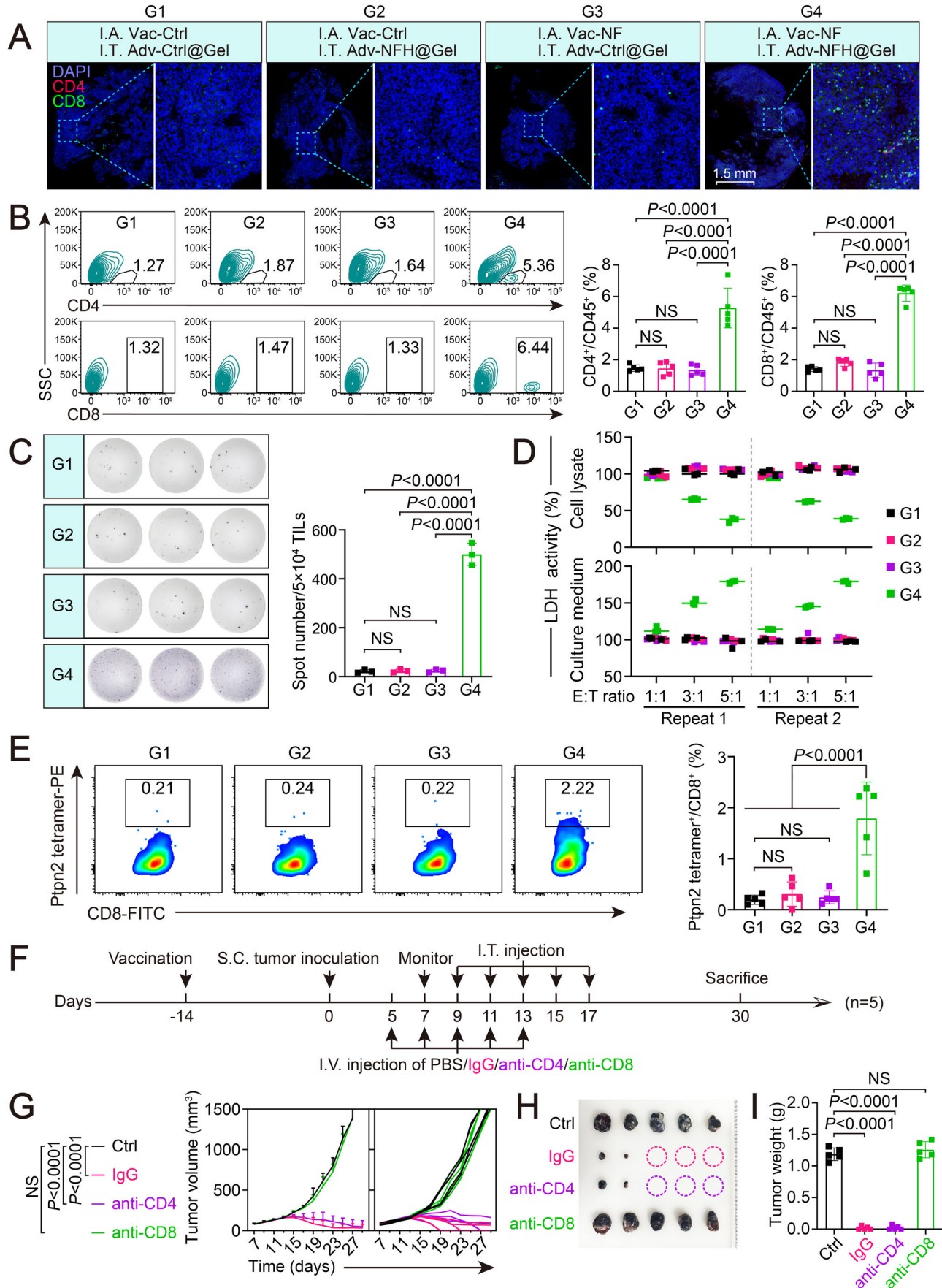

◄

**Figure 5. Tumor infiltration of neoantigen-responsive CD8$^+$ T cells was crucial for the therapeutic benefit of the integrative therapy.**

(A) Representative fluorescence images demonstrated tumor infiltration of T cells. Scale bar: 1.5 mm. (B) Flow cytometry analysis showed the percentage of intratumorally infiltrated CD4$^+$ and CD8$^+$ T cells gated from CD45$^+$ cells ($n = 5$). $P$ value (CD4$^+$/CD45$^+$): G1 vs. G2, $P = 0.99998$; G1 vs. G3, $P = 0.991$; G1 vs. G4, $P = 1.018 \times 10^{-6}$; G2 vs. G4, $P = 9.626 \times 10^{-7}$; G3 vs. G4, $P = 6.496 \times 10^{-7}$. $P$ value (CD8$^+$/CD45$^+$): G1 vs. G2, $P = 0.286$; G1 vs. G3, $P = 0.992$; G1 vs. G4, $P = 4.548 \times 10^{-12}$; G2 vs. G4, $P = 1.711 \times 10^{-11}$; G3 vs. G4, $P = 3.778 \times 10^{-12}$. (C) ELISpot assay with the neoantigen peptide-pulsed DCs demonstrate the antigen-responsiveness of the isolated TILs ($n = 3$). $P$ value: G1 vs. G2, $P = 0.9997$; G1 vs. G3, $P = 0.9996$; G1 vs. G4, $P = 2.720 \times 10^{-8}$; G2 vs. G4, $P = 2.776 \times 10^{-8}$; G3 vs. G4, $P = 2.787 \times 10^{-8}$. (D) LDH assay in remnant Hepa1-6 cells after 48 h co-culture indicated the neoantigen-specific cytotoxicity of the TILs ($n = 3$). Repeat1 and Repeat2 represented TILs isolated from different tumor samples. (E) Flow cytometry analysis determined the percentage of Ptpn2-specific TILs gated from CD8$^+$CD3$^+$ cells ($n = 5$). $P$ value: G1 vs. G2, $P = 0.966$; G1 vs. G3, $P = 0.997$; G1 vs. G4, $P = 3.430 \times 10^{-5}$; G2 vs. G4, $P = 7.967 \times 10^{-5}$; G3 vs. G4, $P = 4.905 \times 10^{-5}$. (F) Schematic view of the depletion experiment in B16F10-established tumor model. PBS (Ctrl) and isotype IgG (IgG, InVivoMAb, BE0090, 100 μg per time) were referred to as the control groups. (G–I) Depletion experiment in C57BL/6 mice demonstrated the importance of CD8$^+$ cells in the integrative therapy ($n = 5$ mice per group). Overall and individual tumor growth curves (G), image of tumor volume (H), and tumor weight (I) were demonstrated. Ctrl: S.C. Vac-NF + I.T. PBS + I.P. PBS; IgG: S.C. Vac-NF + I.T. Adv-NFH@Gel+I.P. IgG; anti-CD4: S.C. Vac-NF + I.T. Adv-NFH@Gel+I.P. anti-CD4; anti-CD8: S.C. Vac-NF + I.T. Adv-NFH@Gel+I.P. anti-CD8. $P$ value (G): IgG, $P = 4.237 \times 10^{-10}$; anti-CD4, $P = 1.405 \times 10^{-9}$; anti-CD8, $P = 0.828$. $P$ value (I): IgG, $P = 1.638 \times 10^{-12}$; anti-CD4, $P = 1.762 \times 10^{-12}$; anti-CD8, $P = 0.546$. Statistical analysis was performed using one-way ANOVA (B, C, E, I) or two-way ANOVA (G). Data are shown as mean ± SD for 3–5 biological replicates. Source data are available online for this figure.

compared to control groups (G1: S.C. Vac-Ctrl & I.T. Adv-Ctrl@Gel; G2: S.C. Vac-Ctrl & I.T. Adv-NFH@Gel; G3: S.C. Vac-NF & I.T. Adv-Ctrl@Gel). Subjecting the isolated tumor-infiltrating lymphocytes (TILs) to ELISpot analysis with the neoantigen peptide-pulsed DCs, a potent immune response was only observed in the TILs obtained from G4 (Fig. 5C). Consistent with the results of ELISpot assay, only TILs from G4 exhibited the capacity to efficiently destruct co-cultured Hepa1-6 cells (Fig. 5D), which served as the resident cells for the designated neoantigens in this study. To further assess the antigen-responsiveness of the TILs, freshly prepared TILs were stained with fluorescently labeled tetramers for neoantigen peptide Ptpn2$_{376\text{-}384}$ (RWLYWQPTL), one of the seven neoantigens used in this study. As shown in Fig. 5E, in contrast to the control groups (~0.25%), peptide-MHC tetramer staining revealed a statistically significant increase of Ptpn2-specific CD8$^+$CD3$^+$ TILs (~1.79%) in G4. Additionally, the Ptpn2-specific T cell reactivity was further validated by ELISpot assay (Appendix Fig. S9), confirming the recruitment of neoantigen-specific T cells by the integrative therapy.

To further investigate the significance of the immune cells in the integrative therapy, depletion experiments were performed with anti-CD4 (InVivoMAb, BE0003-1, 100 μg per time) and anti-CD8 (InVivoMAb, BE0117, 100 μg per time) antibodies (Fig. 5F). Consistent with the results of cytotoxicity assay with vaccine-primed CD4$^+$ and CD8$^+$ cells (Fig. 2F,G), in contrast to the ineffectiveness of CD4 depletion, CD8 depletion completely abolished the therapeutic benefits of the integrative therapy in B16F10 established tumor models (Fig. 5G–I), verifying CD8$^+$ T cells as a crucial effector for the therapeutic outcome of the integrative therapy. Thus, these results collectively validated the intratumoral recruitment of vaccine-primed CD8$^+$ T cells as a responsible mechanism for the functionality of the integrative therapy, confirming the development feasibility of generalized neoantigen vaccines.

In addition, we conducted a more extensive evaluation in B16F10-established tumor tissues following the integrative treatment. As demonstrated in the Fig. EV4A–F, in compare with the control groups (G1, G2 and G3), the percentage of the activated effector T cells (CD69$^+$CD8$^+$ in CD45$^+$ cells), the matured DCs (CD80$^+$CD86$^+$ in CD11c$^+$ cells), NK cells (NK1.1$^+$ in CD45$^+$ cells) and M1 macrophage (CD86$^+$ in F4/80$^+$CD11b$^+$ cells) were increased in G4. In contrast, the composition of M2 macrophage (CD206$^+$ in F4/80$^+$CD11b$^+$ cells) was decreased in G4. Collectively, these results indicated that the integrative therapy reshaped the immune landscape of the treated tumors to promote effective antitumor responses.

## Adoptive cell therapy with vaccine-primed T cells enabled transferring of tumor protection

In case of ineffective vaccination, for example in lymphopenic cancer patients resulting from disease progression or previous chemotherapy, adoptive cell transferring is required to achieve effective immunotherapy. To simulate this scenario, we conducted an integrative treatment involving I.T. virotherapy and intravenous (I.V.) transferring of vaccine-primed T cells in immunodeficient BALB/c nude mice with tumors established from human carcinoma cell lines (Crossland et al, 2018; Ogunnaike et al, 2021; Yamada et al, 1987). In this context, oncolytic adenovirus was applicable due to the well-studied replication and cytolysis characteristics in human cells, enabling the demonstration of a strengthened therapeutic potential of the integrative immunotherapy with these oncolytic features. To this end, the adenovirus (Adv-Ctrl and Adv-NFH) was supplemented with essential elements (E1A and E1B19K) (Matsushita et al, 2004) for replication to generate oncolytic adenoviruses, OAv-Ctrl and OAv-NFH. These were engineered with the human telomerase reverse transcriptase (hTERT) promoter (Kanaya et al, 2020; Kawashima et al, 2004) to improve the selectivity of virus replication and transgene expression of the neoantigens, thereby minimizing potential side effects (Fig. 6A).

To assess the infectivity and reproductivity of the oncolytic adenoviruses in human tumor cells, we infected HepG2 cells, a typical human hepatocellular carcinoma cell line, with the viruses at varying MOIs over a 72 h time course. The expression of mCherry, encoded by the adenoviral shuttle plasmid (pDC315-EF1-mCherry), was detected via flow cytometry analysis. As demonstrated in Fig. 6B, we observed an MOI-dependent increase in the percentage of mCherry-positive tumor cells for both the replication-defective adenoviruses (Adv-Ctrl and Adv-NFH) and oncolytic adenoviruses (OAv-Ctrl and OAv-NFH). The infection efficiency was comparable between the engineered (Adv-NFH and OAv-NFH) and control viruses (Adv-Ctrl and OAv-Ctrl). Notably, the oncolytic adenoviruses infected a significantly higher proportion of HepG2 cells than the replication-defective adenoviruses at low MOIs (ranging from 0.01 to 1 MOI). Despite this, the cytotoxicity of these viruses in HepG2 cells was almost identical,

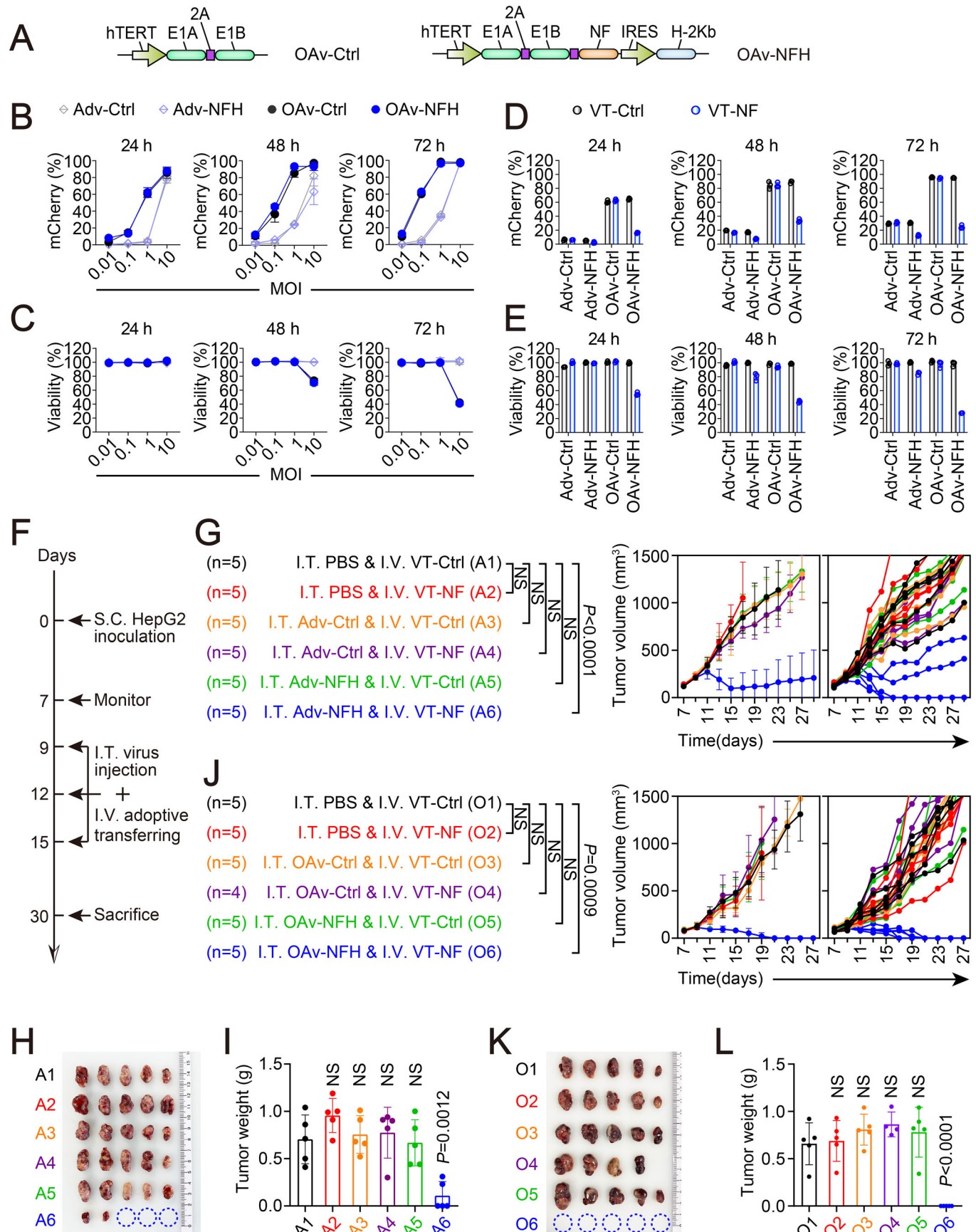

**Figure 6. Adoptive cell therapy with T cells from immunized mice enabled transferring of tumor protection.**

(A) Schematic view of oncolytic adenoviral vectors. (B, C) Quantification of mCherry$^+$ (B) and cell viability (C) of HepG2 cells following the exposure to different MOIs of viruses (Adv-Ctrl, Adv-NFH, OAv-Ctrl, or OAv-NFH) for indicated duration ($n = 3$ biological replicates). (D, E) Quantification of mCherry$^+$ (D) and cell viability (E) of virus-treated HepG2 cells following the co-culture with vaccine-primed T cells (VT-Ctrl or VT-NF) at a 1:3 ratio for indicated duration ($n = 3$ biological replicates). (F) Treatment scheme of the integrative therapy in BALB/c nude mice. (G-I) Measurement of the growth (G), final volume (H), and weight of HepG2 tumors (I) following an integrative therapy between I.T. injection of non-replicable adenoviruses ($10^9$ VP per time) and adoptive transferring of vaccine-primed T cells ($10^6$ cells per time) ($n = 5$ mice per group). $4 \times 10^6$ cells were inoculated into the right flanks of the mice to establish HepG2 tumors. (J-L) Measurement of the growth (J), final volume (K), and weight of HepG2 tumors (L) following an integrative therapy between I.T. injection of oncolytic adenoviruses ($10^9$ VP per time) and adoptive transfer of vaccine-primed T cells ($10^6$ cells per time) ($n = 4-5$ mice per group). $P$ value (L): O2, $P = 0.9992$; O3, $P = 0.603$; O4, $P = 0.375$; O5, $P = 0.762$; O6, $P = 6.029 \times 10^{-5}$. Statistical analysis was performed using one-way ANOVA (I, L) or two-way ANOVA (G, J). Data are shown as mean ± SD. Source data are available online for this figure.

except after prolonged infection (48 or 72 h) at a high MOI (10), indicating the delayed reproduction after viral infection (Bai et al, 2019; Gulbudak et al, 2021). This remarkable contrast between the infection efficiency and cytolysis activity of the oncolytic adenoviruses suggested a window of targetable opportunity for neoantigen vaccine-primed immune system (Fig. 6C). To validate this, HepG2 cells were infected with the viruses at an MOI of 1 and co-cultured with VT-Ctrl or VT-NF at a ratio of 1:3 over a 72 h time course. Due to its oncolytic activity and consequently high infection rate, OAv-NFH treatment resulted in significantly greater elimination of HepG2 cells (up to ~70% at 72 h) than Adv-NFH (~20% at 72 h) when combined with VT-NF cells (Fig. 6D). This result was further confirmed by Cell Counting Kit-8 (CCK8) assay to remnant tumor cells in a parallel experiment (Fig. 6E), highlighting the superior cytotoxicity of the integrative treatment with both OAv-NFH and VT-NF.

Based on the results, we conducted an integrative treatment in BALB/c nude mice with HepG2-established subcutaneous tumors. The treatment involved I.T. injection of viruses (PBS, Adv-Ctrl, Adv-NFH, OAv-Ctrl or OAv-NFH) immediately followed by I.V. transferring of T cells (VT-Ctrl or VT-NF) (Fig. 6F). As demonstrated in Fig. 6G–I, although the integration of Adv-NFH and VT-NF cells resulted in remarkable tumor regression, 2/5 of the mice remain tumor-bearing after three rounds of treatment. In contrast, potentiated by the oncolytic activity of OAvs, the integration of OAv-NFH and VT-NF cells led to complete tumor regression in all of the treated mice (Fig. 6J–L) while maintaining a favorable safety profile (Appendix Fig. S10A,B), indicating the therapeutic benefits and the applicability of oncolytic viruses as a preferable transgenic vector for the integrative therapy in immunocompetent individuals. Meanwhile, by demonstrating the efficacy of treatment with oncolytic virus and adoptive transfer of neoantigen vaccine-primed T cells, these results highlighted a highly effective strategy to broaden the applicability of the integrative therapy.

## Validation of the applicability in human cell-based system with HLA-A*02:01 and HLA-A*02:01-restricted neoantigens

To further demonstrate the clinical relevance of our strategy, HLA-A*02:01 and HLA-A*02:01-restricted neoantigens were used to examine its applicability in human-related settings. To this end, 7 HLA-A*02:01-restricted neoantigens that have been validated in literature were selected for the examination, including TMEM48_F > L, SEC24A_P > L, TKT_R > W, EXT2_D > Y, SLC39A11_P > L, ASTN1_P > L, and SMARCD3_H > Y (Carreno

et al, 2015; Reparaz et al, 2022; Stronen et al, 2016). Following an established protocol for inducing neoantigen-specific T cells from healthy donors (Ali et al, 2019), peptides of the selected neoantigens were synthesized (GenScript) for in vitro antigen presentation and validation of neoantigen-reactivity of the primed T cells. Subsequently, 3 HLA-A*02:01$^+$ healthy donors (HLA typing by Weihe Biotechnology INC.) were enrolled. Autologous DCs were isolated from peripheral blood mononuclear cells (PBMCs) using plastic adherence method (Huang et al, 2022), while naive T cells (CD3$^+$CD8$^+$CD62L$^+$CD45RA$^+$) were purified by flow cytometry-based cell sorting (Cieri et al, 2013). The isolated naive T cells were then stimulated twice with peptide-pulsed DCs to enable in vitro antigen presentation.

Co-culture of the resulting cell populations with T2 cells pulsed with individual peptides (Fig. 7A, incubated overnight with 100 nM peptide) revealed clear T cell reactivity against the selected neoantigens, as quantified by IFNγ staining via flow cytometry. This reactivity was consistently observed across all 3 donors, albeit with donor-specific variations in intensity and epitope preference. In contrast, such reactivity was negligible in the control cultures (Fig. 7B). Noteworthy, TMEM48_F > L and SLC39A11_P > L elicited more stable and potent T cell reactivity than other neoantigens. The specificity of these responses was further validated using peptide-HLA-A*02:01 tetramer staining, with TMEM48_F > L- and SLC39A11_P > L-specific T cells constituting 4.4–8.47% and 4.98–14.8% of total peptide-primed CD8$^+$ T cells, respectively (Fig. 7C). Consistent with these findings, T cells primed with the selected neoantigens exhibited potent cytotoxicity against peptide-pulsed T2 cells, as measured by LDH release assays and flow cytometry analysis (Fig. 7D,E). These results collectively confirm the successful priming of naive T cells and the immunogenicity of the selected neoantigens.

To determine whether forced peptide-MHC complex transfer could redirect the cytotoxicity of the peptide-primed T cells, we engineered a replication-defective adenovirus (Adv-hNFH) and an oncolytic adenovirus (OAv-hNFH), both co-expressing HLA-A*02:01 and selected HLA-A*02:01-restricted neoantigens (Fig. 7F). Adenoviruses expressing only neoantigens (Adv-hNF) or HLA-A*02:01 (Adv-HLA) were used as controls. Transgene expression was validated by qPCR analysis in virus-infected SNU-449 cells (Fig. 7G), a HLA-A*02:01-negative human hepatocellular carcinoma cell line (HLA-A*11:01, 31:01) (Scholtalbers et al, 2015). As demonstrated in Fig. 7H, when SNU449 cells were used as target cells for cytotoxicity assay, the neoantigen-primed T cells exhibited significant cytotoxicity toward Adv-hNFH- and OAv-hNFH-infected cells, while showing negligible cytotoxicity against the cells pre-treated by PBS, Adv-Ctrl, Adv-hNF, Adv-HLA, or

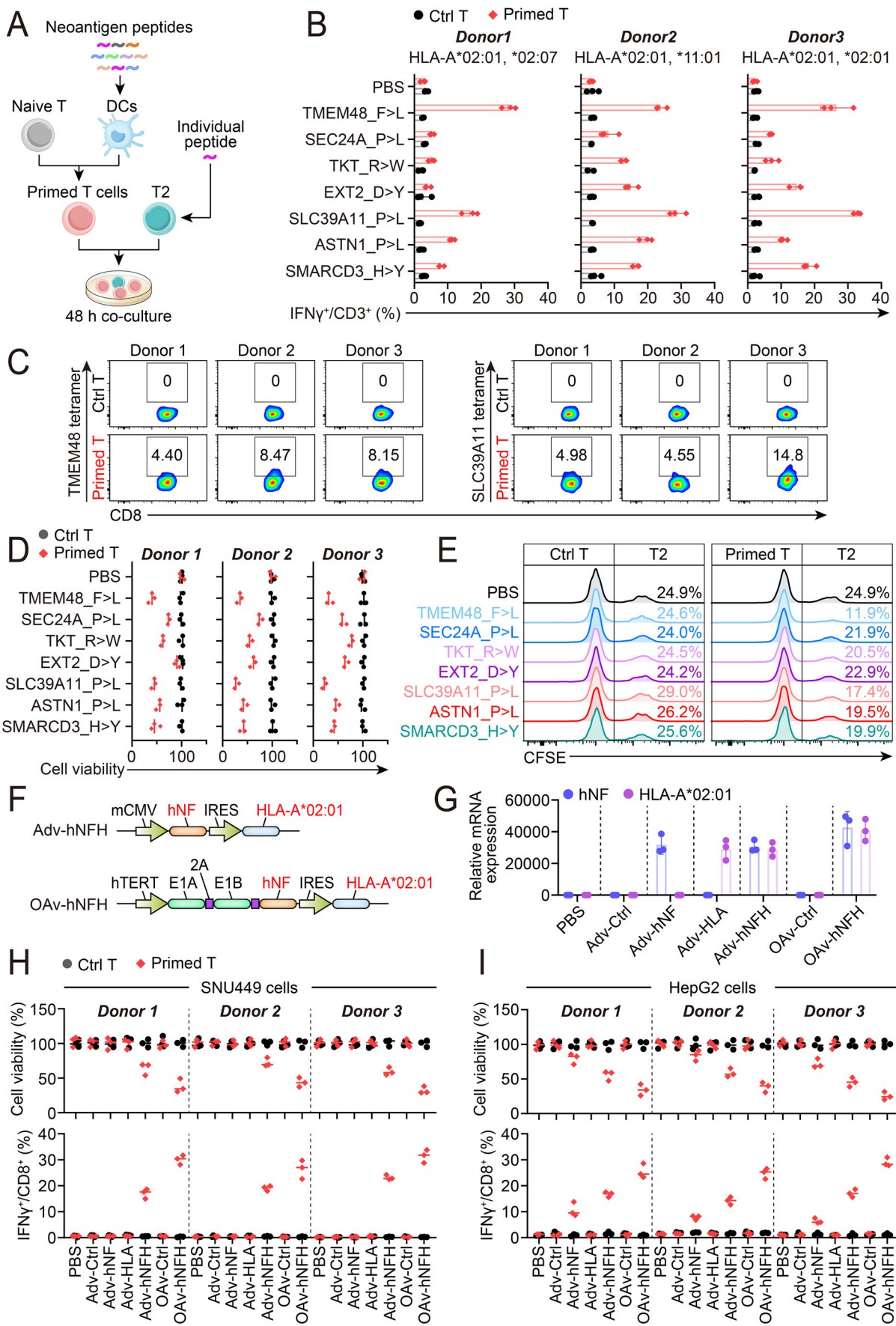

Figure 7.  Preclinical validation in human cell-based system with HLA-A*02:01 allele and HLA-A*02:01-restricted epitopes.

(A) Treatment scheme for in vitro antigen presentation and peptide-pulsed T2 re-stimulation. (B) IFNγ staining demonstrated the T cell reactivity to individual neoantigen peptides ($n = 3$). (C) Peptide-HLA-A*02:01 tetramer staining showed the specificity of T cell reactivity. (D) LDH assay showed the elimination of peptide-pulsed T2 cells by the primed T cells ($n = 3$). Following a 24 h co-culture (E:T = 3:1), the culture medium was collected for the assessment of LDH activity. Cell viability was determined by calculating the ratio of LDH activity in the culture medium to that in the lysate of the control T2 cells. (E) CFSE (Thermo Fisher Scientific, C34570) staining demonstrate the cytotoxicity of peptide-primed T cells towards peptide-pulsed T2 cells. Following the in vitro antigen presentation, T cells of donor 1 were labeled with a low concentration of CFSE (0.5 µM) and co-cultured with peptide-pulsed T2 cells labeled with a high concentration of CFSE (5 µM) at a 3:1 ratio. Selective elimination of the peptide-pulsed T2 cells was measured 24 h later by flow cytometry analysis. (F) Schematic view of adenoviral vectors. (G) qPCR analysis validated the expression of HLA-A*02:01 and HLA-A*02:01-restricted neoantigens in virus-infected SNU-449 cells ($n = 3$). (H) Cytotoxicity assay demonstrated selective elimination of Adv-hNFH- and OAv-hNFH-infected SNU-449 cells by peptide-primed T cells ($n = 3$). SNU-449 cells were pre-treated by the PBS or indicated viruses (1 MOI for 48 h) and co-cultured with the peptide-primed T cells at a ratio of 1:3 for 24 h before subjecting to LDH assay. (I) Peptide-primed T cells selectively eliminate HepG2 cells that had been infected by Adv-hNF, Adv-hNFH, and OAv-hNFH ($n = 3$). The procedure for cell treatment was identical to that described in Figure H. Data are shown as mean ± SD for 3 biological replicates. Source data are available online for this figure.

OAv-Ctrl. In contrast, no notable cytotoxicity was observed when the pre-infected SNU-449 cells were co-cultured with control T cells (PBS-treated). Notably, consistent with results shown in Fig. 6E, OAv-hNFH elicited superior cytotoxicity of the primed T cells compared to Adv-hNFH in SNU-449 cells. Such cytotoxicity of the primed T cells was similarly observed in HLA-A*02:01-positive HepG2 cells (HLA-A*02:01, 24:02) (Boegel et al, 2014), with the exception of the Adv-hNF group (Fig. 7I). Align with the findings presented in Fig. EV1D, transfer of neoantigens in tumor cells endogenously expressing neoantigen-binding MHC molecules by Adv-NF elicited cytotoxicity of the primed T cells, albeit to a lesser extent than Adv-hNFH and OAv-hNFH.

Collectively, by leveraging HLA-A02:01 and HLA-A02:01-restricted epitopes for validation, our results compellingly demonstrate the feasibility of our strategy within human system, underscoring its promising translational potential for clinical application.

# Discussion

The binding of non-self peptides to MHC-I molecules is essential for CD8+ T cell-mediated recognition and elimination of tumor cells expressing aberrant proteins. As a key immune evasion strategy employed by tumor cells, loss of the MHC-I antigen presentation machinery (APM) impairs the ability of the host immune system to effectively control tumor progression (Dhatchinamoorthy et al, 2021). Acting as key components of peptide-MHC ligands for T cell receptors (TCR), either MHC-I restoration (Gu et al, 2021; Pulido et al, 2020) or epitope delivery-mediated forced antigenicity (Cuburu et al, 2022; Millar et al, 2020; Rosato et al, 2019) have been demonstrated to be effective in potentiating tumor immunotherapy. Nevertheless, restoring peptide-MHC complex within tumor cells by co-delivery of MHC-I molecules and antigenic epitopes represents an indispensable strategy to address tumor heterogeneity, especially in the context where the intratumoral microenvironment remains poorly characterized. However, to the best of our knowledge, the benefit of the delivery of peptide-MHC complex remains unexplored to date.

In this study, to address the major challenges of neoantigen vaccines for tumor therapy, we demonstrated the ability of oncolytic virus to deliver neoantigen peptide-MHC complexes into tumor cells, enabling a safe and functional integration between oncolytic virus and neoantigen vaccines to suppress the growth of solid tumors in both local and regional metastatic tumor models.

We propose that this approach can promote the widespread application of neoantigen vaccines, while timely and economically translate into the clinic to broaden their therapeutic benefits in various intractable solid tumors. The application scenarios of the integrative therapy encompass, but are not limited to: (1) Expanding the application of the neoantigen vaccine across different patients sharing identical MHC subtypes; (2) Employing the same neoantigen vaccine across various lesions in patients with multiple primary tumors or even metastatic tumors featuring clonal neoantigen loss; (3) Overcoming tumor heterogeneity or antigen/MHC loss by force-expressing neoantigen peptide-MHC complexes in tumor cells without pre-defined neoantigens, or directly applying in recurrent tumor post neoantigen vaccination without the need for further neoantigen identification to forcibly reinstating tumor cell responsiveness to vaccine treatment; (4) Achieving synergistic therapy with oncolytic viruses overexpressing neoantigen peptide-MHC complexes by adoptive transferring neoantigen vaccine-primed T cells in immunocompromised patients. Patients who have already received neoantigen vaccines can utilize their pre-existing immunity to undergo the integrative therapy, thereby avoiding delays caused by the time required for the vaccine to take effect.

In contrast to the study combining CD19-delivering OV with CD19-CAR T cells (Park et al, 2020), our work was characterized by the introduction of neoantigen peptide-MHC complexes and neoantigen vaccination. By leveraging tumor-specific immunity of endogenous immune system for tumor therapy, our strategy offers an alternative to address limitations of CAR-T cell therapy, including high cost of treatment, long-term persistence of T cells, and adverse events associated with the on-target off-tumor effects and malignant T-cell transformation (Perica et al, 2025; Rafiq et al, 2020; Zhang et al, 2022). Notably, as shown in Fig. 6, adoptive transferring of vaccine-primed T or TCR-engineered T cells (TCR-T) can substitute for neoantigen vaccination, providing a flexible regimen for lymphopenic cancer patients. Regarding therapeutic timeliness, taking into account the manufacturing and distribution process, while vaccination typically requires a few weeks to establish protective immunity, well-stocked neoantigen vaccines could match the timeliness of CAR T-cell therapies (Yang et al, 2022). In our study, initiating I.T. injections as early as day 6 post-vaccination induced significant tumor regression in mouse models, suggesting the potential to extend the therapeutic window for neoantigen vaccines (Fig. 4E). However, further studies are warranted to fully elucidate the differences between these strategies.

In the field of neoantigens, determining the criteria for selecting the target neoantigen(s) is a key issue (Wolf and Sameuls, 2023).

The therapeutic efficacy of neoantigen vaccines is believed to be primarily determined by their qualitative characteristics, such as their ability to induce non-self-recognition and self-discrimination (Łuksza et al, 2022), rather than their quantity (Wolf and Sameuls, 2023). This distinction underscores the importance of selecting high-quality neoantigens to maximize the therapeutic potential of the integrative therapy in practice. For the generalization of neoantigen vaccines across diverse patients who share identical HLA serotypes, with the ongoing comprehensive exploration of the therapeutic potential of neoantigen vaccines, we anticipate continual accumulation of pertinent application results, as well as growing pool of validated neoantigen sequences, which will enable the selection of high-quality neoantigens. Noteworthy, being considered as an alternative class of neoantigens (Blass and Ott, 2021; Xie et al, 2023), viral proteins could be included and adapt into the integrative therapy as peptides or proteins, following comprehensive assessment of their therapeutic potential.

Aside from neoantigens, MHC molecules have been incorporated into viral vectors as a prophylactic setting to prevent MHC loss-induced immune escape. As a result, MHC matching is becoming a prerequisite for the application of our approach. MHC proteins are encoded by a highly polymorphic gene family in all vertebrates (Abualrous et al, 2021). In contrast to adoptive T cell therapies, which ideally require comprehensive match of MHC profiles between donor and recipient to maximally prevent the development of graft-versus-host disease (GVHD) and graft rejection (Perkey and Maillard, 2018), allelic matching as demonstrated in the current study is a sufficient condition for our approach, extending the applicable range of neoantigen vaccines from the individual person to the subpopulation. In line with this perspective, according to a report by Cynthia X. L. Chang et al, the principle variants of MHC-I alleles, including HLA-A (*02:03, *02:06, *02:07, *02:11, *11:01, *24:02, and *33:03), HLA-B (*15:02, *40:01, *46:01, *55:02, and *58:01), and HLA-C (*03:04, *04:01, *07:02, and *08:01), could, respectively, provide 93%, 63%, and 79% coverage in South East Asian (SEA) population (Chang et al, 2013). Likewise, we analyzed the allele frequency in a Han population residing in Shanxi Province of China (Zhao et al, 2023) (Fig. EV5A–D) and confirmed that a few HLA molecules (allele frequency > =5%) could provide coverage for the majority of the population, indicating a broad applicability of the integrative therapy in humans. Certainly, adaptive engineering based on the allele frequencies of MHC molecules in the local population is essential for the further application of the integrative therapy across different geographic regions. In addition to MHC restriction, although we performed current proof-of-concept studies by delivering the heavy chain of MHC-I proteins (H-2Kb) to avoid immune escape caused by downregulated or lost expression of MHC-I molecules in tumor cells, it is worthwhile to consider the clinical significance of B2M in future studies because of its essential role in maintaining the stability of the MHC-I complex (Del Campo et al, 2014). Furthermore, considering the various mechanisms of immune escape within the tumor microenvironment, combination with other therapeutic approaches and further modification of the oncolytic virus to co-deliver checkpoint inhibitors (Wang et al, 2020), antibodies (Xu et al, 2021), cytokines (Kemp et al, 2019) or chemokines (Li et al, 2020) are warranted to enhance the therapeutic efficacy of the integrative therapy.

Moreover, as a common pathogen in humans and the most widely used adenoviral serotype, Ad5 has been plagued by the pre-existing immunity, limiting its efficacy as a vaccine and therapeutic vector (Zaiss et al, 2009). In this sense, both neutralizing antibodies and CD8+ T lymphocytes have been reported to be contributors of the anti-Ad5 immunity, with neutralizing antibodies serving as the primary immunological effector mechanism (Sumida et al, 2004). Interestingly, by performing various in vitro and in vivo experiments, no apparent cellular immune response against Adv-Ctrl infected tumor cells was observed (Figs. 2, 3, 4, 6 and EV1–EV3) in Adv-Ctrl primed T cells (VT-Ctrl). As discussed in the report by Sumida et al (2004), this phenomenon could probably be explained by the relatively low doses of the viruses used in the current study, which may limit both the establishment of cellular immunity against Ad5 (Fitzgerald et al, 2003; Sumida et al, 2004) and the potency of the intratumoral immune response of Ad5-specific T cells. Indeed, this assumption was verified by repeating the infection-induced DC maturation (Appendix Fig. S11A), cytotoxicity assay (Appendix Fig. S11B,C), and integrative therapy in C57BL/6 mice (Appendix Fig. S11D) with a serial doses of Adv-Ctrl, confirming the dose as a key factor limiting viral backbone-induced cellular immunity. Regarding the humoral immunity for the in vivo evaluation of the integrative therapy, to at least partially mitigate host immunity-mediated clearance, we encapsulated the viruses in silk hydrogel to improve the persistence after I.T. injection and ensure demonstration of therapeutic benefit in both immunocompetent and immunodeficient mouse models. Similarly, we envision that other types of vaccines (e.g., peptides (Malonis et al, 2020), DNA (Duperret et al, 2019), mRNA (Cafri et al, 2020), and dendritic cell vaccines (Saxena and Bhardwaj, 2018)), oncolytic viruses (e.g., chimeric adenoviruses (Flickinger et al, 2020), herpes simplex viruses (Ju et al, 2022), vaccinia viruses (Park et al, 2020) and Newcastle disease viruses (Tian et al, 2023)) and biocompatible materials (Kreppel and Kochanek, 2008) can be exploited in future studies to improve the efficacy of the integrative therapy.

In conclusion, we have demonstrated the effective utilization of oncolytic adenovirus to deliver neoantigen-MHC complexes into tumor cells, rewiring vaccine-primed immunity to synergize with virotherapy for enhanced destruction of solid tumors. Consequently, our findings raised the possibility of universalizing neoantigen vaccines as a treatment paradigm for various solid tumors, regardless of the expression profile of the neoantigens. By validating and overcoming the potential limitations in future studies, this approach could serve as a fundamental platform to establish more effective immunotherapy for solid tumors.

# Methods

**Reagents and tools table**

| Reagent/Resource | Reference or Source | Identifier or Catalog Number |
|---|---|---|
| **Experimental models** | | |
| HEK293T (*H. sapiens*) | ATCC | Cat #CRL-3216 |
| HepG2 (*H. sapiens*) | ATCC | Cat #HB-8065 |
| SK-Hep-1 (*H. sapiens*) | ATCC | Cat #HTB-52 |
| SNU-449 (*H. sapiens*) | ATCC | Cat #CRL-2234 |
| T2 (*H. sapiens*) | ATCC | Cat #CRL-1992 |

| Reagent/Resource | Reference or Source | Identifier or Catalog Number |
|---|---|---|
| Hepa1-6 (*M. sapiens*) | ATCC | Cat #CRL-1830 |
| MC38 (*M. sapiens*) | Kerafast | Cat #ENH204-FP |
| HEK293A (*H. sapiens*) | Procell | Cat #CL-0003 |
| B16F10 (*M. sapiens*) | Procell | Cat #CL-0319 |
| LLC (*M. sapiens*) | Procell | Cat #CL-0140 |
| SMMC-7721 (*H. sapiens*) | Second Military Medical University (China) | N/A |
| Hepa1-6-Luc (*M. sapiens*) | This study | N/A |
| Hepa1-6-GFP (*M. sapiens*) | This study | N/A |
| C57BL/6 mice (*M. musculus*) | Shanghai Silaike Experimental Animal Co., Ltd. | N/A |
| BALB/c nude mice (*M. musculus*) | Shanghai Silaike Experimental Animal Co., Ltd. | N/A |
| **Recombinant DNA** | | |
| pDC315 | GeneBiogist Technology | N/A |
| pBHGlox (delta) E1, 3Cre | GeneBiogist Technology | N/A |
| RGRKRRS | Duperret et al, 2019 | N/A |
| GGSGGGGSGG | Sahin et al, 2017 | N/A |
| **Antibodies** | | |
| Anti-Flag antibody | Abmart | Cat #M20008M |
| Anti-β-actin antibody | Cell Signaling Technology | Cat #12262S |
| Anti-CD3ε | BioLegend | Cat #100340 |
| Anti-CD28 | BioLegend | Cat #102116 |
| FITC-conjugated F4/80 | BioLegend | Cat #123107 |
| APC-conjugated anti-CD11c | eBioscience | Cat #17-0114-82 |
| PE-Cyanine7-conjugated anti-NK1.1 | eBioscience | Cat #25-5941-82 |
| APC-conjugated CD11b | eBioscience | Cat #17-0112-82 |
| PE-conjugated anti-CD80 | eBioscience | Cat #12-0801-82 |
| PE-Cyanine7-conjugated anti-CD86 | eBioscience | Cat #25-0862-82 |
| FITC-conjugated anti-CD3 | eBioscience | Cat #11-0037-42 |
| PE-conjugated anti-CD8 | eBioscience | Cat #12-0089-42 |
| PE-conjugated CD206 | eBioscience | Cat #12-2061-82 |
| PE-Cy5-conjugated anti-CD62L | eBioscience | Cat #15-0629-42 |
| PE-Cy7-conjugated anti-CD45RA | eBioscience | Cat #25-0458-42 |
| APC-conjugated anti-CD3 | eBioscience | Cat #11-0032-82 |
| prepCP Cy5.5-conjugated anti-CD25 | eBioscience | Cat #45-0251-82 |
| PE-Cyanine7-conjugated anti-CD69 | eBioscience | Cat #25-0691-82 |
| PE-conjugated anti-CD8 | eBioscience | Cat #12-0084-82 |
| PE-Cyanine7-conjugated anti-IFNγ | eBioscience | Cat #25-7311-82 |
| FITC-conjugated anti-CD8 | MBL Life Science | Cat #D271-4 |
| anti-CD8 primary antibodies | Servicebio | Cat #GB15068 |
| anti-CD4 primary antibodies | Servicebio | Cat #GB13064-2 |
| anti-CD4 | InVivoMAb | Cat #BE0003-1 |
| anti-CD8 | InVivoMAb | Cat #BE0117 |
| IgG | InVivoMAb | Cat #BE0090 |
| H-2Kb antibody | Biolegend | Cat #116505 |
| anti-adenovirus antibody | Merck | Cat #AB1056 |
| Ptpn2-specific [Ptpn2$_{376-384}$(RWLYWQPTL):H-2Kb] tetramer-PE | This study | N/A |
| TMEM48-specific (TMEM48_F > L:HLA-A*02:01) tetramer-APC | Cancer Research Center of Xiamen University | N/A |
| SLC39A11-specific (SLC39A11_P > L: HLA-A*02:01) tetramer-APC | Cancer Research Center of Xiamen University | N/A |
| **Oligonucleotides and other sequence-based reagents** | | |
| qPCR primer | This study | Methods |
| **Chemicals, Enzymes and other reagents** | | |
| FastDigest restriction enzymes | Thermo Fisher Scientific | N/A |
| T4 ligase | Thermo Fisher Scientific | Cat #EL0014 |
| Collagenase type IV | Gibco | Cat #17104-019 |
| DNase I | Solarbio | Cat #D8070 |
| Hyaluronidase | Solarbio | Cat #H8030 |
| DMEM | Thermo Fisher Scientific | Cat #C11995500BT |
| RPMI-1640 | Thermo Fisher Scientific | Cat #C11875500BT |
| Lymphocyte serum-free medium KBM581 | Corning | Cat #88-581-CM |
| Recombinant IL-2 | Jiangsu Kingsley Pharmaceutical Co., Ltd | N/A |
| Recombinant human GM-CSF | TOPSCIENCE | Cat #TMPY-03858 |
| Recombinant mouse GM-CSF | Novoprotein | Cat #CK02 |
| Recombinant mouse IL-4 | Novoprotein | Cat #CK15 |
| Recombinant human IL-4 | Novoprotein | Cat #C050 |
| Recombinant human IL-7 | Novoprotein | Cat #C086 |
| Recombinant human TNFα | Novoprotein | Cat #DC008 |

| Reagent/Resource | Reference or Source | Identifier or Catalog Number |
| --- | --- | --- |
| Recombinant human IL-21 | Novoprotein | Cat #CC45 |
| LPS | Beyotime Biotechnology | Cat #ST1470-10mg |
| DAPI | Beyotime Biotechnology | Cat #C1002 |
| DiI | Beyotime Biotechnology | Cat #C1036 |
| CM-DiI | Yeasen Biotechnology | Cat #40718ES50 |
| CsCl | Sigma-Aldrich | Cat #C3032-500G |
| PEG8000 | Sigma-Aldrich | Cat #P5413-1KG |
| Red Blood Cell Lysis Buffer | Solarbio | Cat #R1010 |
| Ficoll PLUS | Tianjin Haoyang Biological Manufacture | Cat #P1114 |
| Taq Pro Universal SYBR qPCR Master Mix | Vazyme Biotech | Cat #Q712-02 |
| CellTrace™ CFSE | Thermo Fisher Scientific | Cat #C34570 |
| Lipofectamine™ 3000 Transfection Reagent | Thermo Fisher Scientific | Cat #L3000001 |
| Bovine serum albumin | Sigma-Aldrich | Cat #V900933-1KG |
| TMEM48_F > L | Carreno et al, 2015 | N/A |
| SEC24A_P > L | Carreno et al, 2015 | N/A |
| TKT_R > W | Carreno et al, 2015 | N/A |
| EXT2_D > Y | Reparaz et al, 2022 | N/A |
| SLC39A11_P > L | Reparaz et al, 2022 | N/A |
| ASTN1_P > L | Stronen et al, 2016 | N/A |
| SMARCD3_H > Y | Stronen et al, 2016 | N/A |
| Mapk3-S284F | Chen et al, 2022 | N/A |
| Lmf1-F523V | Chen et al, 2022 | N/A |
| Samd91-K752M | Chen et al, 2022 | N/A |
| Traf7-C403W | Chen et al, 2022 | N/A |
| Dtnb-K40T | Chen et al, 2022 | N/A |
| Lbr-A341P | Chen et al, 2022 | N/A |
| Ptpn2-I383T | Chen et al, 2022 | N/A |
| Matrigel | Corning Life Sciences | Cat #3330624 |
| **Software** | | |
| Image Lab 5.0, | https://www.bio-rad.com/zh-cn/product/image-lab-software?ID=KRE6P5E8Z | N/A |
| GraphPad Prism software V 8.0 | https://www.graphpad.com/ | N/A |
| Snapgene 6.1.1 | https://www.snapgene.com/updates/snapgene-6-1-1-release-notes | N/A |
| Flowjo | https://www.flowjo.com/ | N/A |
| **Other** | | |
| EasyPure® Genomic DNA Kit | TransGen Biotech | Cat #EE101-12 |

| Reagent/Resource | Reference or Source | Identifier or Catalog Number |
| --- | --- | --- |
| Cell Counting Kit (CCK-8) | Yeasen Biotechnology | Cat #40203ES76 |
| Mouse adenovirus antibody (IgG) ELISA Kit | CUSABIO Biotech CO., Ltd. | Cat #CSB-E13901m |
| TransZol Up Plus RNA Kit | TransGen Biotech | Cat #ER501-01-V2 |
| Hifair® II 1st Strand cDNA Synthesis Kit | Yeasen Biotechnology | Cat #11121ES60 |
| CD4+ T Cell Isolation Kit | Miltenyi Biotec | Cat #130-104-454 |
| CD8 MicroBeads | Miltenyi Biotec | Cat #130-116-478 |
| LDH cytotoxicity assay kit | Beyotime Biotechnology | Cat #C0017 |
| Cytofix/Cytoperm™ Plus Fixation/Permeabilization Solution Kit | BD Bioscience | Cat #554715 |
| ELISpot Plus: Mouse IFN-γ (ALP) | Mabtech | Cat #3321-4APT-2 |
| 40 μm cell strainer | Biosharp | Cat #BS-40-CS |
| BD FACSAria™ Fusion Cell Sorter | BD Bioscience | N/A |
| BD FACSVerse™ | BD Bioscience | N/A |
| Applied Biosystems 7500 Real-Time PCR System | Thermo Fisher Scientific | N/A |
| Nikon Eclipse C1 | Nikon | N/A |
| ELISpot Analysis System | Antai Yongxin Medical Technology | Cat #AT-Spot2200 |
| QuickSwith™ Quant H-2Kb Tetramer Kit | MBL Life Science | Cat #TB-7400-K1 |
| SCIENTZ-IID | Ningbo Scientz Biotechnology | N/A |
| IVIS® Spectrum in Vivo Imaging System | PerkinElmer | N/A |

## Materials

AdMax adenovirus system including shuttle plasmid pDC315 and backbone plasmid pBHGlox (delta) E1, 3Cre was obtained from GeneBiogist Technology (Shanghai, China). FastDigest restriction enzymes and T4 ligase were purchased from Thermo Fisher Scientific (United States). Anti-Flag antibody (M20008M) was purchased from Abmart (Shanghai, China). Anti-β-actin antibody (12262S) was purchased from Cell Signaling Technology (United States). DiI (C1036) and LPS (ST1470-10mg) were purchased from Beyotime Biotechnology (Shanghai, China). EasyPure® Genomic DNA Kit was purchased from TransGen Biotech (Beijing, China). CCK8 (40203ES76) and CM-DiI (40718ES50) were purchased from Yeasen Biotechnology (Shanghai, China). Dulbecco's modified Eagle's medium (DMEM) and Roswell Park Memorial Institute 1640 Medium (RPMI-1640) were purchased from Thermo Fisher Scientific (United States). Lymphocyte serum-free medium KBM581 was purchased from Corning (United States). Recombinant mouse granulocyte-macrophage colony-stimulating factor (GM-CSF, CK02) and Interleukin-4 (IL-4, CK15) were purchased

from Novoprotein (Shanghai, China). Mouse adenovirus antibody (IgG) ELISA Kit (CSB-E13901m) was purchased from CUSABIO Biotech CO., Ltd. (Wuhan, China).

## Cell culture

Mouse hepatocellular carcinoma cell line Hepa1-6, mouse Lewis lung carcinoma cell line LLC, HEK293A, HEK293T, human hepatoma cell lines (HepG2 and SMMC-7721), and liver carcinoma cell line SK-Hep-1 were cultured in DMEM supplemented with 10% fetal bovine serum (FBS, ExCell). Human hepatoma cell line SNU-449, lymphoblastoid cell line T2, mouse melanoma cell line B16F10 and colon adenocarcinoma cell line MC38 were maintained in RPMI-1640 containing 10% FBS. HEK293T (CRL-3216), HepG2 (HB-8065), SK-Hep-1 (HTB-52), SNU-449 (CRL-2234), T2 (CRL-1992), and Hepa1-6 (CRL-1830) were purchased from ATCC. MC38 (ENH204-FP) was purchased from Kerafast. HEK293A (CL-0003), B16F10 (CL-0319), LLC (CL-0140) were obtained from Procell, Wuhan, China. SMMC-7721 was generously provided by Professor Hongyang Wang from the Second Military Medical University (China). Hepa1-6 cells stably expressing firefly luciferase (Hepa1-6-Luc) or GFP (Hepa1-6-GFP) were generated by lentiviral transduction. All the cells were well maintained at 37 °C in a humidified incubator under a 5% $CO_2$ atmosphere and frequently checked by morphology monitoring. All the cell lines were authenticated by short tandem repeat (STR) analysis and routinely screened for the existence of mycoplasma.

## Plasmid construction

Subsequent to the construction of pDC315-EF1-mCherry, neoantigen expressing cassette was subcloned into the MCS of pDC315-EF1-mCherry plasmid in the downstream of MCMV promoter. Thereafter, an IRES-dependent MHC-I expressing cassette was subcloned into the plasmids in the downstream of neoantigen expressing cassette and in the upstream of EF1α promoter to allow co-expression of neoantigens and neoantigen-binding MHC-I molecules. To generate oncolytic adenoviral vectors, necessary genes for adenoviral replication, including E1A and E1B19k, were cloned and connected to neoepitope-expressing cassette by 2 A peptides. All the recombinant plasmids were verified by Sanger sequencing (BioSune Biotechnology, Shanghai).

## Adenovirus production

Propagating HEK293A cells in 35 mm dishes to let the cells reach 70–80% of confluency, the engineered shuttle plasmids were, respectively, co-transfected (Lipofectamine™ 3000 Transfection Reagent, Thermo Fisher Scientific, L3000001) with pBHGlox (delta) E1, 3Cre at a 1:2 mass ratio. By visualizing small cytopathic plaques after 10 to 21 days since the transfection, both the cells and the culture medium were collected and subjected to freeze-thaw cycle (freezing to −80 °C and thawing to 37 °C) for 3 times before pelleting cell debris by centrifugation at $4500 \times g$, 4 °C for 30 min. 1 ml supernatant was subsequently added into freshly pre-seeded HEK293A cells to amplify the virus. After 3 rounds of amplification, the harvested adenoviruses were condensed with PEG8000 (Sigma-Aldrich, P5413-1KG), and purified by CsCl (Sigma-Aldrich, C3032-500G) density gradient centrifugation and subsequent dialysis. The viruses were thereafter titrated via plaque assay or

qPCR analysis to the genomic DNA extracts of adenoviruses (Huang et al, 2019).

## qPCR assay

Subsequent to the indicated treatment, the mRNA of the cells was extracted with TransZol Up Plus RNA Kit (TransGen Biotech, ER501-01-V2), reverse transcribed into cDNA with Hifair® II 1st Strand cDNA Synthesis Kit (Yeasen Biotechnology, 11121ES60), and quantitatively analyzed in triplicate with Taq Pro Universal SYBR qPCR Master Mix (Vazyme Biotech, Q712-02) using the Applied Biosystems 7500 Real-Time PCR System (Thermo Fisher Scientific, United States) by following the manufacturers' instructions. 18S rRNA was referred to as internal control and the relative expression of the interested genes was calculated via $2^{-\Delta\Delta Ct}$ method (Livak and Schmittgen, 2001). Specific primers were as follows:

NF forward: 5′-AGAGGAAGAAAACGGCGAAG-3′;
NF reverse: 5′-CACAAAAGCCACCACAGCTC-3′;
NGS forward: 5′-ATCTGGAGGTGGCGGCAGCGGT-3′;
NGS reverse: 5′-GTCCCCTCCTTTTCCACCTG-3′;
H-2Kb forward: 5′-AGATTCCCCAAAGGCCCATG-3′;
H-2Kb reverse: 5′-AAGAGGCACCACCACAGATG-3′;
B2M forward: 5′-CAGCAAGGACTGGTCTTT-3′;
B2M reverse: 5′- ACATGTCTCGATCCCA-3′;
hNF forward: 5′-AGAAAGAGGAGAAGCGCCAC-3′;
hNF reverse: 5′-CCTGAGAGTAGCTTCCCCCT-3′;
HLA-A*02:01 forward: 5′-TCTACCCTGCGGAGATCACA-3′;
HLA-A*02:01 reverse: 5′-AGCTCCAAAGAGAACCAGGC-3′;
18S forward: 5′-AGAAACGGCTACCACATCCA-3′;
18S reverse: 5′-CACCAGACTTGCCCTCCA-3′.

## Isolation of mouse bone marrow-derived DCs (BMDCs)

The femur and tibia of mice were obtained after cervical dislocation by cutting the back legs and cleaning the muscle tissue. After sterilization with 70% ethanol, the bones were washed with PBS, kept in ice-cold PBS, cut with scissors to remove both ends of the bones, and flushed until completely white by a syringe needle to acquire the cells in the bone marrow. Filtering through a 40 μm cell strainer (Biosharp, BS-40-CS) and lysing the red blood cells (Red Blood Cell Lysis Buffer, Solarbio, R1010), the acquired cells were washed with FBS-containing RPMI-1640 and cultured in BMDC-specific medium, which was prepared by supplementing RPMI-1640 with 20 ng/ml mouse GM-CSF and 10 ng/ml mouse IL-4. The cells were maintained at 37 °C in a humidified 5% $CO_2$ incubator. Half of the culture medium was renewed every 2 days. The non-adherent cells were collected on 6th day for further analysis.

## Isolation of mouse SMNCs

Freshly removed mouse spleens were washed with PBS to flush out the content until mostly fibrous tissue remains. The cells were pelleted, washed twice, and resuspended with PBS. Subsequently, the cell suspension was gently layered over an equal volume of Ficoll PLUS (Tianjin Haoyang Biological Manufacture, China, P1114) and centrifuged for 20 min at $800 \times g$ without break. After completing the centrifugation, the white and cloudy layer containing mononuclear cells was carefully transferred to a new tube to

wash off remaining debris with at least 3 volumes of PBS. For ELISpot analysis and T cell activation assay, acquired SMNCs were used immediately or frozen in −80 °C refrigerator for long-term applications. For in vitro cytotoxicity assay and in vivo adoptive transferring, the cells were stimulated with anti-CD3ε/anti-CD28 antibodies and cytokine interleukin-2 (IL-2) to acquire activated splenic T cells. Concretely, the cells were cultured for 3 days in KBM581 containing 2.5 µg/ml anti-CD3ε (BioLegend, 100340), 1 µg/ml anti-CD28 (BioLegend, 102116) and 1000 U/ml IL-2 (Jiangsu Kingsley Pharmaceutical Co., Ltd.), and thereafter maintained in KBM581 containing only 1000 U/ml IL-2. To demonstrate the specificity of cytotoxicity, the activated T cells were respective sorted with CD4$^+$ T Cell Isolation Kit (Miltenyi Biotec, 130-104-454) and CD8 MicroBeads (Miltenyi Biotec, 130-104-075) according to the manufacturer's instructions before use.

## Isolation of mouse TILs

Tumor tissues were minced into small pieces (~1–3 mm) with sterile scissors, and treated with DMEM containing collagenase type IV (Gibco, 17104-019, 1 mg/ml), DNase I (Solarbio, D8070, 20 µg/ml), hyaluronidase (Solarbio, H8030, 200 µg/ml) and 2% FBS in incubator with shaking at 37 °C for 1 h. Afterwards, the tumor fragments were homogenized with pipette tips and filtered through a 40 µm cell strainer. After they were pelleted by centrifugation at $800 \times g$ for 5 min, the cells were washed twice with 15 ml DMEM and resuspended in 10 ml DMEM. The single-cell suspension was carefully layered over on 10 ml of Ficoll PLUS in a 50 ml conical tube and separated by following the centrifugation conditions as described above to acquire the TILs. For ELISpot analysis and cytotoxicity assay, the TILs were used immediately or frozen for long-term application.

## In vitro loading of neoantigens to mouse DCs with adenovirus

To evaluate the maturation of DCs, murine bone marrow-derived DCs were, respectively, cultured with PBS, LPS (1 µg/ml), or 2 MOI of indicated adenovirus in a 24-well plate for 72 h. Subsequently, the cells were pelleted by centrifugation at $800 \times g$ for 3 min, blocked with PBS containing 5% bovine serum albumin (BSA, Sigma-Aldrich, V900933-1KG) at room temperature for 30 min, and co-stained with APC-conjugated anti-CD11c (eBioscience, 17-0114-82, 2.5 µg/ml), PE-conjugated anti-CD80 (eBioscience, 12-0801-82, 0.6 µg/ml) and PE-Cyanine7-conjugated anti-CD86 (eBioscience, 25-0862-82, 2.5 µg/ml) antibodies at room temperature in the dark for 30 min. After 2 times of wash with PBS, the percentage of adenovirus-mediated infection (BD FACSAria™ Fusion Cell Sorter) and the extent of DC maturation (BD FACSVerse™) were both determined via flow cytometry analysis.

## Priming of human T cells in vitro

As with mouse SMNCs, HLA-A*02:01 positive peripheral blood mononuclear cells (PBMCs) from healthy donors were isolated from 20 ml of peripheral blood by density gradient centrifugation with Ficoll PLUS. After erythrocyte lysis, PBMCs were washed twice with PBS and resuspended in KBM581 (Corning, 88-581-CM). Monocytes were enriched via plastic adherence by plating the PBMCs in a 6-well plate for 2 h. The adherent cells were cultured in KBM581 supplemented with 10 ng/ml GM-CSF (TOPSCIENCE, TMPY-03858) and 10 ng/ml IL-4 (Novoprotein, C050) to obtain DCs, while the non-adherent cells were cultured in KBM581 containing 5 ng/ml IL-7 (Novoprotein, C086) to obtain T cells.

For the T cells, cell staining was performed on day 3 after culture with fluorescent dye-conjugated antibodies (purchased from eBioscience) to isolate naive T cells by flow cytometry-based cell sorting, including FITC-conjugated anti-CD3 (11-0037-42, 0.1 µg/ml), PE-conjugated anti-CD8 (12-0089-42, 0.006 µg/ml), PE-Cy5-conjugated anti-CD62L (15-0629-42, 0.05 µg/ml), and PE-Cy7-conjugated anti-CD45RA (25-0458-42, 0.025 µg/ml). The resulting naive T cells were maintained in KBM581 containing 5 ng/ml IL-7. For the DCs, the culture medium was half renewed on day 3 and day 5, with the addition of TNFα (Novoprotein, DC008, 10 ng/ml) and LPS (10 ng/ml) on day 5 for DC activation. Activated DCs were harvested on day 6 and pulsed with HLA-A*02:01-restricted neoantigen peptides for 4 h (10 µg/ml for each peptide), including TMEM48_F > L (CLNEYHLF<u>L</u>), SEC24A_P > L (FLYN<u>L</u>LTRV), TKT_R > W (AMF<u>W</u>SVPTV), EXT2_D > Y (KYV<u>Y</u>DFGVSV), SLC39A11_P > L (ALL<u>F</u>LESEL), ASTN1_P > L (K<u>L</u>YGLDWAEL), and SMARCD3_H > Y (KLFEFLV<u>Y</u>GV).

For antigen presentation, the obtained naive T cells were co-cultured with peptide-pulsed DCs in a ratio of 1:2 in the presence of 30 ng/ml IL-21 (Novoprotein, CC45). After 72 h, 5 ng/ml IL-7 and 5 ng/ml IL-15 were added to the culture medium. The culture medium was half renewed on day 5, and subculture was carried out on day 7. On day 10 of the co-culture, a second round of stimulation (3 days) was performed by adding peptide-pulsed DCs to the co-culture system (1:2 ratio).

## T cell activation assay

DCs that had been infected with indicated adenovirus at a MOI of 2 for 72 h were co-cultured with primary SMNCs at a ratio of 1:5 for 24 h. The non-adherent DCs and SMNCs were collected and co-cultured with pre-seeded Hepa1-6 cells ($1 \times 10^5$ per well in 24-well plate) at a ratio of 3:1 for 48 h. Subsequently, the adherent cells, which in large part consisted of remnant tumor cells, were subjected to LDH cytotoxicity assay following the manufacturer's instructions (Beyotime Biotechnology, C0017). The non-adherent cells in the supernatant, including DCs, SMNCs and apoptotic tumor cells, were collected and subjected to immunofluorescence staining with a combination of fluorescent-dye conjugated antibodies, including APC-conjugated anti-CD3 (eBioscience, 11-0032-82, 2.5 µg/ml), prepCP Cy5.5-conjugated anti-CD25 (eBioscience, 45-0251-82, 2.5 µg/ml), PE-Cyanine7-conjugated anti-CD69 (eBioscience, 25-0691-82, 5 µg/ml), PE-conjugated anti-CD8 (eBioscience, 12-0084-82, 1.25 µg/ml), and PE-Cyanine7-conjugated anti-IFNγ (eBioscience, 25-7311-82, 2.5 µg/ml). After 30 min of room temperature staining, the cells were washed twice with 1 ml PBS by centrifugation at 800 g for 3 min and suspended with 200 µl PBS before subjecting to flow cytometry analysis (BD FACSVerse™). Specific for intracellular IFNγ staining, Cytofix/Cytoperm™ Plus Fixation/Permeabilization Solution Kit with BD GolgiStop™ was used. Briefly, GolgiStop Protein Transport Inhibitor (4 µl per 6 ml culture medium) was added into the culture medium 4 h before the harvest of the cells, which were thereafter treated with Fixation/Permeabilization solution for 20 min at 4 °C before the immunofluorescence staining. Analysis of the data acquired from flow cytometry was performed with FlowJo software.

## Cytotoxicity assay

Tumor cells were seeded in a 24-well plate at a density of $1 \times 10^5$ cells per well. Other than Hepa1-6, the cells were infected with 1 MOI of indicated virus for 24 h. Following the attachment of the cells, splenic T cells acquired from vaccinated C57BL/6 mice were added into the plate and co-cultured with the tumor cells at a 3:1 ratio for 48 h. The adherent cells, non-adherent cells and culture medium were, respectively, collected. Cell death resulted from the cytotoxicity of the T cells was measured in the lysate of adherent cells and the culture medium by the LDH cytotoxicity assay or CCK8 assay in according to the manufacturer's instructions. On the other hand, T cell activation assay was carried out in the non-adherent cells by following the aforementioned procedures.

## ELISpot analysis

The experiments were carried out in SMNCs or TILs by following the manufacturer's instructions (Mabtech, 3321-4APT-2). Briefly, DCs were treated with neoantigen peptides (Genescript, $4 \mu g/10^4$ cells) for 24 h or indicated adenoviruses for 48 h. Subsequently, the 96-well plate for ELISpot was preconditioned with cell culture medium and incubated for 30 min at room temperature. By removing the medium, premixed $5 \times 10^4$ tested cells and pre-treated $1 \times 10^4$ DCs were added into the plate and incubated in humidified incubator for 48 h. Thereafter, the cells were removed and the spots were detected by sequentially incubating the plate with remaining reagents in the kit, including detection antibody (R4-6A2-biotin), Streptavidin-ALP and ready-to-use substrate solution (BCIP/NBT-plus). Number of spots in the ELISpot plate was counted with ELISpot Analysis System (AT-Spot2200, Antai Yongxin Medical Technology, Beijing, China).

## Tetramer staining

The Ptpn2-specific [Ptpn2$_{376-384}$(RWLYWQPTL):H-2Kb] tetramer-PE was prepared with QuickSwith™ Quant H-2Kb Tetramer Kit (MBL Life Science, TB-7400-K1) by following the manufacturer's instructions. For tetramer staining, freshly isolated TILs ($10^6$) were incubated in 1 ml of block buffer (PBS containing 5% BSA) for 10 min. The cells were then pelleted and resuspended with 100 μl of PBS containing 0.5% BSA. 10 μl of Ptpn2$_{376-384}$ (RWLYWQPTL):H-2Kb tetramer-PE was added into the suspension, thoroughly mixed with the cells and incubated at room temperature in the dark for 15 min. Subsequently, APC-conjugated anti-CD3 (eBioscience, 11-0032-82, 2.5 μg/ml) and FITC-conjugated anti-CD8 (MBL Life Science, D271-4, 10 μl) antibodies were mixed with the cells and incubated at 4 °C in the dark for 30 min. The cells were washed with 1 ml of PBS for twice before subjecting to flow cytometry analysis (BD FACSVerse™).

The TMEM48-specific (TMEM48_F > L:HLA-A*02:01) and SLC39A11-specific (SLC39A11_P > L: HLA-A*02:01) tetramer-APC were produced by Cancer Research Center of Xiamen University. Like the tetramer staining of mouse T cells, the neoantigen-primed human T cells ($10^6$) were collected and stained with the APC-conjugated tetramers (1:50 dilution, 15 min) and PE-conjugated anti-CD8 antibody (eBioscience, 12-0089-42, 1.5 μg/ml, 30 min) before subjecting to flow cytometry analysis.

## Silk hydrogel preparation

Silk hydrogel was prepared by following the protocol described in Danielle N Rockwood et al,'s report (Rockwood et al, 2011). Briefly, to prevent potential immune responses caused by the sericin, pieces of silk cocoons were firstly boiled in 0.02 M $Na_2CO_3$ solution for 30 min, rinsed 3 times and dried overnight to obtain extracted silk fibroin. Subsequently, the silk fibers were dissolved in 9.3 M lithium bromide at a ratio of 1:4 (wt/vol), dialyzed with ultrapure water and centrifuged at $13,000 \times g$ for 2 times to remove residual debris. After determining the concentration by dehydration, the silk solution was adjusted to a concentration of 2% (wt/vol) and sonicated in an ultrasonic homogenizer (SCIENTZ-IID, Ningbo Scientz Biotechnology, China) for 3 min (2 s ON, 3 s OFF) at 30% amplitude for gelation.

## Histology

Freshly dissected tumor and major organ tissues were fixed for 24 h in 4% paraformaldehyde and embedded in paraffin blocks following conventional procedures. After cut and mount of tissue sections (5 μm), the slides were deparaffinized and subjected to hematoxylin and eosin (HE) or immunofluorescence staining. Specific for the immunofluorescence staining, anti-CD8 (Servicebio, GB15068, 1:200) and anti-CD4 (Servicebio, GB13064-2, 1:200) primary antibodies were used before the counterstaining with DAPI (Beyotime Biotechnology, C1002, 5 μg/ml). Images were obtained using a fluorescence microscope (Nikon Eclipse C1) and processed with Adobe Photoshop.

## Animal studies

Male C57BL/6 mice and BALB/c nude mice aged 6-8 weeks were purchased from Shanghai Silaike Experimental Animal Co., Ltd. [licence number: SCXK (Shanghai) 2022-0004] and housed in the conventional environment of institutional animal facility.

For preventative intervention, C57BL/6 mice were vaccinated with PBS or $2 \times 10^9$ VP of indicated adenoviruses at a single dose by the S.C. route, including Vac-Ctrl, Vac-NF, and Vac-NGS. Fourteen days later, Hepa1-6-Luc cells ($5 \times 10^6$ cells per mouse) were prepared in PBS and engrafted into the right flanks of the mice. Tumor volume was monitored every two days by caliper measurement and every 5 days via IVIS® Spectrum in Vivo Imaging System (PerkinElmer, USA). Similarly, B16F10 cells ($3 \times 10^6$ cells per mouse) infected with Adv-Ctrl or Adv-NFH were prepared in PBS containing 1:1 (v/v) of Matrigel (Corning Life Sciences, 3330624, 50 μl), injected subcutaneously 14 days after the vaccination (Vac-Ctrl or Vac-NF) and monitored every two days via caliper measurement. Tumor volume was calculated using a formula based on the orthogonal diameters of the tumors: $V = L \times W^2 \times 0.5$.

For integrative studies in immunocompetent C57BL/6 mice, vaccination with Vac-Ctrl or Vac-NF was performed 14 days before or 3 days after the inoculation of tumor cells. B16F10 cells ($3 \times 10^6$ cells per mouse) or LLC cells ($3 \times 10^6$ cells per mouse) were prepared in PBS containing 1:1 (v/v) of Matrigel (50 μl) and subcutaneously injected into the right flank of C57BL/6 mice. Once tumor volume reached about 100 mm³ (modeling early stage) or

~450 mm³ (modeling late stage), therapeutic adenoviruses (Adv-Ctrl or Adv-NFH) were prepared in 100 μl injectable silk hydrogel and intratumorally administered at $10^9$ VP per mouse for 3 to 5 times. Tumor volume was measured with caliper every two days.

To evaluate the establishment of systemic antitumor immunity after the integrative therapy, B16F10 cells were subcutaneously ($3 \times 10^6$ cells prepared in PBS containing Matrigel) or intravenously ($1.5 \times 10^6$ cells prepared in PBS) implanted into the cured mice and age-matched control mice. Tumor volume of subcutaneous tumors was monitored with caliper every 2 days, while pulmonary metastasis was validated by removing the lungs at the time of sacrifice. In addition, in a bilateral tumor model, B16F10 cells ($3 \times 10^6$ cells per flank of a mouse) were prepared in PBS containing 1:1 (v/v) of Matrigel (50 μl) and subcutaneously injected into both of the flanks of C57BL/6 mice on the 14th day of vaccination. Once tumor volume reached about 100 mm³, therapeutic adenoviruses ($10^9$ VP) were prepared in 100 μl silk hydrogel and intratumorally injected into one flank of the mice. Both the injected and distant tumors were monitored via caliper measurement every 2 days.

For integrative studies in immunodeficient nude mice, HepG2 cells ($5 \times 10^6$ cells per mouse) were prepared in PBS containing 1:1 (v/v) of Matrigel (50 μl) and subcutaneously implanted into the right flanks of the BALB/c nude mice. Once tumor volume reached about 200 mm³, PBS, therapeutic non-replicate (Adv-Ctrl and Adv-NFH) or oncolytic adenoviruses (OAv-Ctrl and OAv-NFH) were, respectively, prepared in 100 μl silk hydrogel and intratumorally injected at $10^9$ VP per mouse every 3 days for a total of 3 injections. Immediately following the I.T. injection of PBS or therapeutic viruses, splenic T cells ($5 \times 10^6$ per mouse) isolated from Vac-Ctrl or Vac-NF vaccinated C57BL/6 mice (VT-Ctrl or VT-NF) were intravenously administered into the mice. Tumor volume was measured every two days with caliper measurement.

## Ethical regulations

All animal experiments were carried out by strictly following the protocols approved by the Institutional Animal Ethics Committee (MHCC-AEC-2023-09). The studies involving human samples were conducted in strict accordance with ethical guidelines and were formally approved by the institutional Ethics Review Committee (KESHEN 2021_100_02). All the experiments involving human sampling adhered to the principles outlined in the World Medical Association (WMA) Declaration of Helsinki and the Department of Health and Human Services' Belmont Report. Written informed consent was obtained from all participating individuals.

## Statistical analysis

The experiments were conducted as nonblind test. For in vitro experiments, all cell lines were identically treated without prior designation. For in vivo studies, animals were randomly grouped by stratifying the tumor volume or body weight. Sample sizes were estimated according to previous experience that demonstrated significance. All the in vitro experiments were representative of 3 independent experiments with similar results ($n = 3$), while the exact sample sizes of in vivo experiments were indicated in the figures and figure legends. No data was excluded from data analyses in both in vitro and in vivo experiments. Unless otherwise stated, the results were presented as means ± SD and analyzed with

### The paper explained

#### Problem

Neoantigen vaccines represent a groundbreaking approach in cancer immunotherapy by training the immune system to recognize tumor-specific mutations. However, their application in treating solid tumors faces significant challenges. The highly personalized nature of these vaccines requires costly and time-intensive processes, including tumor genome sequencing, computational prediction of immunogenic neoantigens, and custom manufacturing for each patient. These steps often delay treatment and limit accessibility. Furthermore, current prediction algorithms lack precision in identifying neoantigens that reliably trigger immune responses, and tumors frequently evolve to reduce or eliminate targetable neoantigens—a phenomenon known as tumor heterogeneity. While a universal neoantigen vaccine that bypasses these limitations would benefit cancer care, no effective solution currently exists, leaving many patients without viable treatment options.

#### Results

To address these challenges, we engineered oncolytic viruses designed to co-express neoantigens and their corresponding MHC molecules. This innovative approach forces tumors to display pre-assembled neoantigen-MHC complexes on their surface, effectively "guiding" immune cells to recognize and attack cancer cells irrespective of the endogenous neoantigen expression profiles. In preclinical models spanning multiple cancer types, including melanoma, pulmonary carcinoma, and hepatocellular carcinoma, combining these engineered viruses with neoantigen vaccines demonstrated potent tumor-killing effects. The treatment not only eradicated primary tumors but also prevented tumor recurrence and induced systemic immunity capable of shrinking distant, untreated tumors—a phenomenon called the abscopal effect. Crucially, this strategy maintained a strong safety profile, with no evidence of severe side effects, underscoring its potential for clinical translation.

#### Impact

This study offers a transformative solution to the limitations of personalized neoantigen vaccines. By decoupling therapeutic efficacy from tumor-intrinsic neoantigen expression, our engineered viral platform overcomes critical barriers of neoantigen vaccines, such as high costs, manufacturing delays, and tumor immune evasion. The ability to force neoantigen presentation ensures that even heterogeneous or evolving tumors remain vulnerable to immune attack, significantly reducing the likelihood of treatment resistance. Importantly, this approach bridges the gap between personalized and universal therapies, providing a scalable framework applicable across diverse tumor types and patient populations.

GraphPad Prism software. Statistical significance among groups was calculated by one-way or two-way analysis of variance (ANOVA) test as indicated in figure legends. Alternatively, log-rank (Mantel-Cox) test was performed in survival analysis. Results were determined to be statistically significant at $P < 0.05$. Data distribution was assumed to be normal, although this was not formally tested. Significance of the specific statistical analysis was labeled in the figures (NS: not significant).

## Data availability

This study includes no data deposited in external repositories.

The source data of this paper are collected in the following database record: biostudies:S-SCDT-10_1038-S44321-025-00225-3.

## Peer review information

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

## Acknowledgements

We thank Lingjie Wu, Geng Chen, Zhenli Li, Naishun Liao, and Yang Zhou at the Mengchao Hepatobiliary Hospital of Fujian Medical University for expertized technical and consultative assistance. This work was supported by the National Natural Science Foundation of China (62175031), Natural Science Foundation of Fujian Province (2021J011294 and 2023J011482), the Startup Fund for Scientific Research of Fujian Medical University (2020QH1239), and Fujian Key Laboratory of Advanced Technology for Cancer Screening and Early Diagnosis (CSED-ZDSYS-202302).

## Author contributions

**Chenyi Wang**: Formal analysis; Validation; Investigation; Visualization; Methodology. **Yingjun Shi**: Formal analysis; Validation; Investigation; Visualization; Methodology. **Da Zhang**: Validation; Methodology. **Yupeng Sun**: Validation; Methodology. **Junjie Xie**: Validation; Investigation. **Bingchen Wu**: Validation; Methodology. **Cuilin Zhang**: Conceptualization; Formal analysis; Investigation; Writing—original draft; Writing—review and editing. **Xiaolong Liu**: Conceptualization; Resources; Supervision; Project administration; Writing—review and editing.

Source data underlying figure panels in this paper may have individual authorship assigned. Where available, figure panel/source data authorship is listed in the following database record: biostudies:S-SCDT-10_1038-S44321-025-00225-3.

## Disclosure and competing interests statement

The authors declare no competing interests.

# Expanded View Figures

**Figure EV1. Expression of B2M, H-2Kb and the neoantigens in murine and human cell lines.**

(**A**) With indicated primers, the relative expression of B2M and H-2Kb was quantitatively analyzed in HEK293T, HepG2, SK-Hep-1, SMMC-7721, Hepa1-6, B16F10, MC38, and LLC cells ($n = 3$). (**B**) LLC and B16F10 cells were treated with PBS, Adv-Ctrl (1 MOI), Adv-NFH (1 MOI), or Adv-NGSH (1 MOI) for 24 h. The cells were collected and stained by a H-2Kb antibody (Biolegend, 116505, clone AF6-88.5, 5 µg/ml) for 30 min before subjecting to flow cytometry analysis. (**C**) Sanger sequencing results demonstrated the existence of the neoantigens in indicated cell lines. Data are shown as mean ± SD for 3 biological replicates.

▶

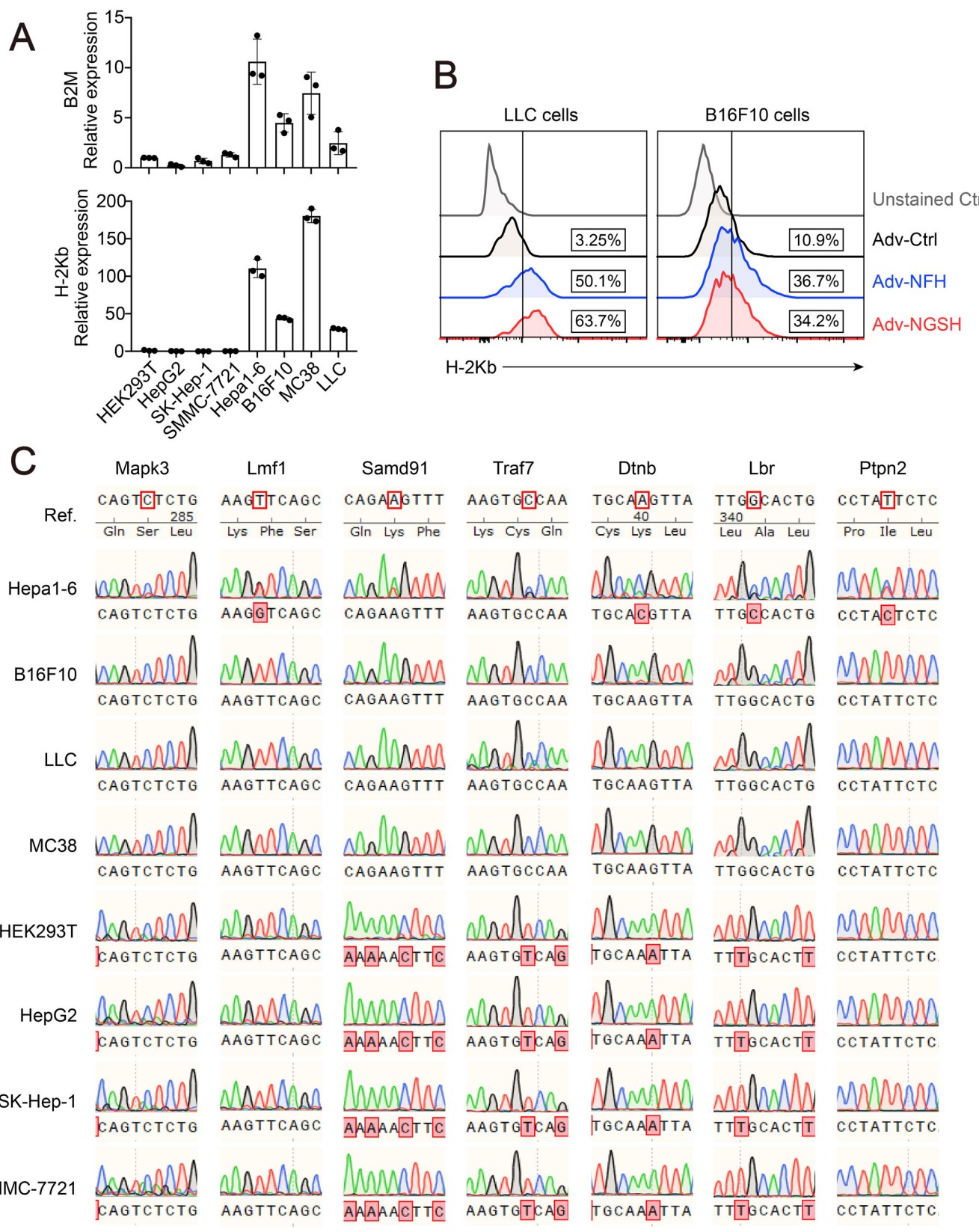

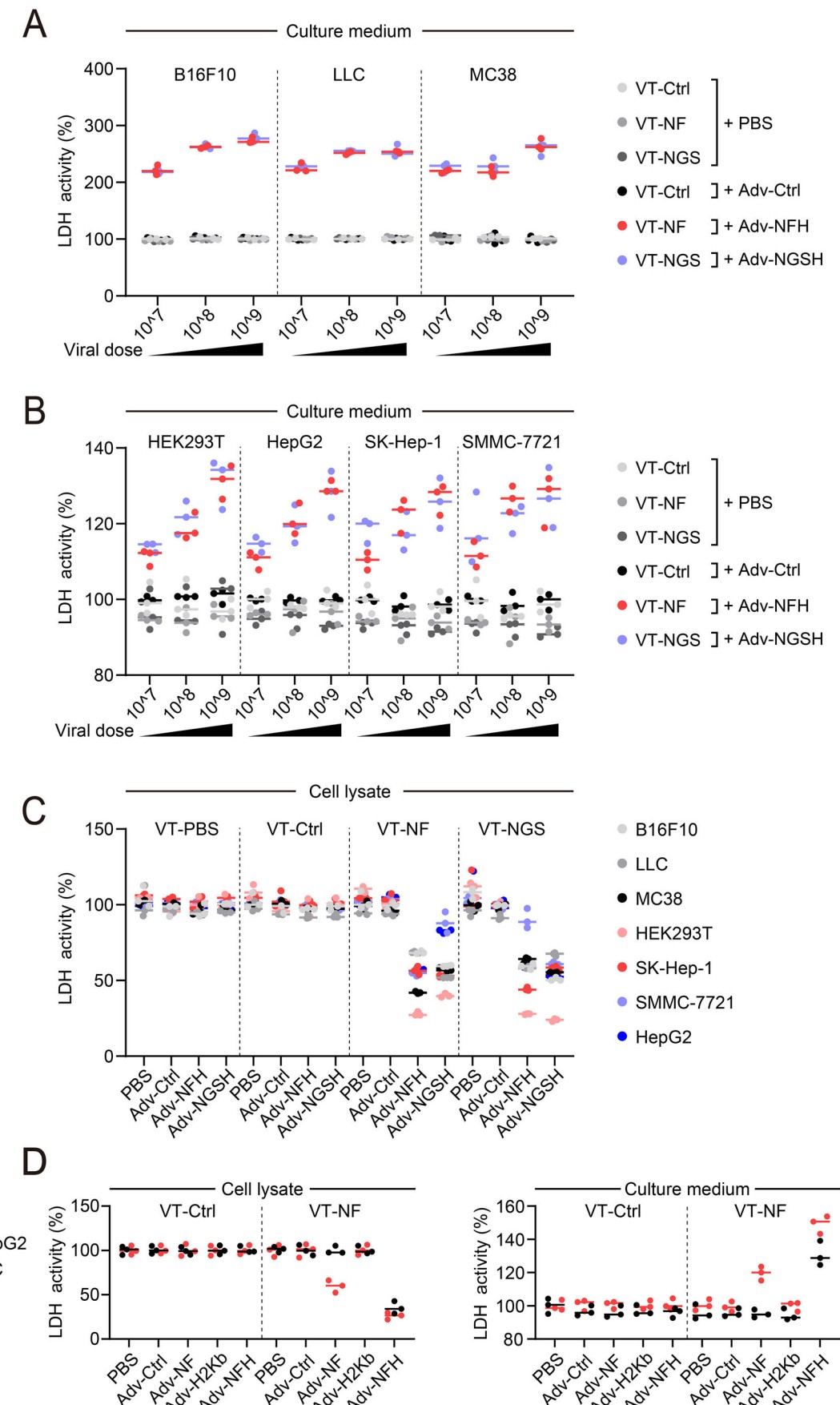

◀  **Figure EV2.  Delivery of neoantigen peptide-MHC complexes redirected the cytotoxicity of neoantigen vaccine-primed T cells.**

(**A**) LDH assay demonstrated the dose-dependent cytotoxicity of adenovirus-primed T cells in murine tumor cells that infected with Adv-NFH or Adv-NGSH. Samples were the culture medium of the murine cells assessed in Fig. 2E ($n = 3$). (**B**) Dose-dependent cytotoxicity of adenovirus-primed T cells in human tumor cells infected by Adv-NFH or Adv-NGSH. Samples were the culture medium of the human cells used in Fig. 2E ($n = 3$). (**C**) LDH assay in remnant tumor cells showed the specificity of the redirected cytotoxicity ($n = 3$). The assay was performed by following the experimental protocol in Fig. 2E. Mice were vaccinated with $2 \times 10^9$ VP of adenoviruses for 14 d before harvesting the SMNCs to prepare related T cells (VT-PBS, VT-Ctrl, VT-NF, and VT-NGS). Tumor cells, including B16F10, LLC, MC38, HEK293T, SK-Hep-1, SMMC-7721, and HepG2, were pre-treated with 1 MOI of indicated viruses and co-cultured with the prepared T cells at an E:T ratio of 1:3 for 48 h. (**D**) LDH assay demonstrated the functionality of MHC-I overexpression. After 48 h infection with 1 MOI of indicated viruses, HepG2 and LLC cells were co-cultured with VT-Ctrl or VT-NF at a ratio of 1:3 for 48 h before assessing the cell viability ($n = 3$). Data are shown as mean ± SD for 3 biological replicates.

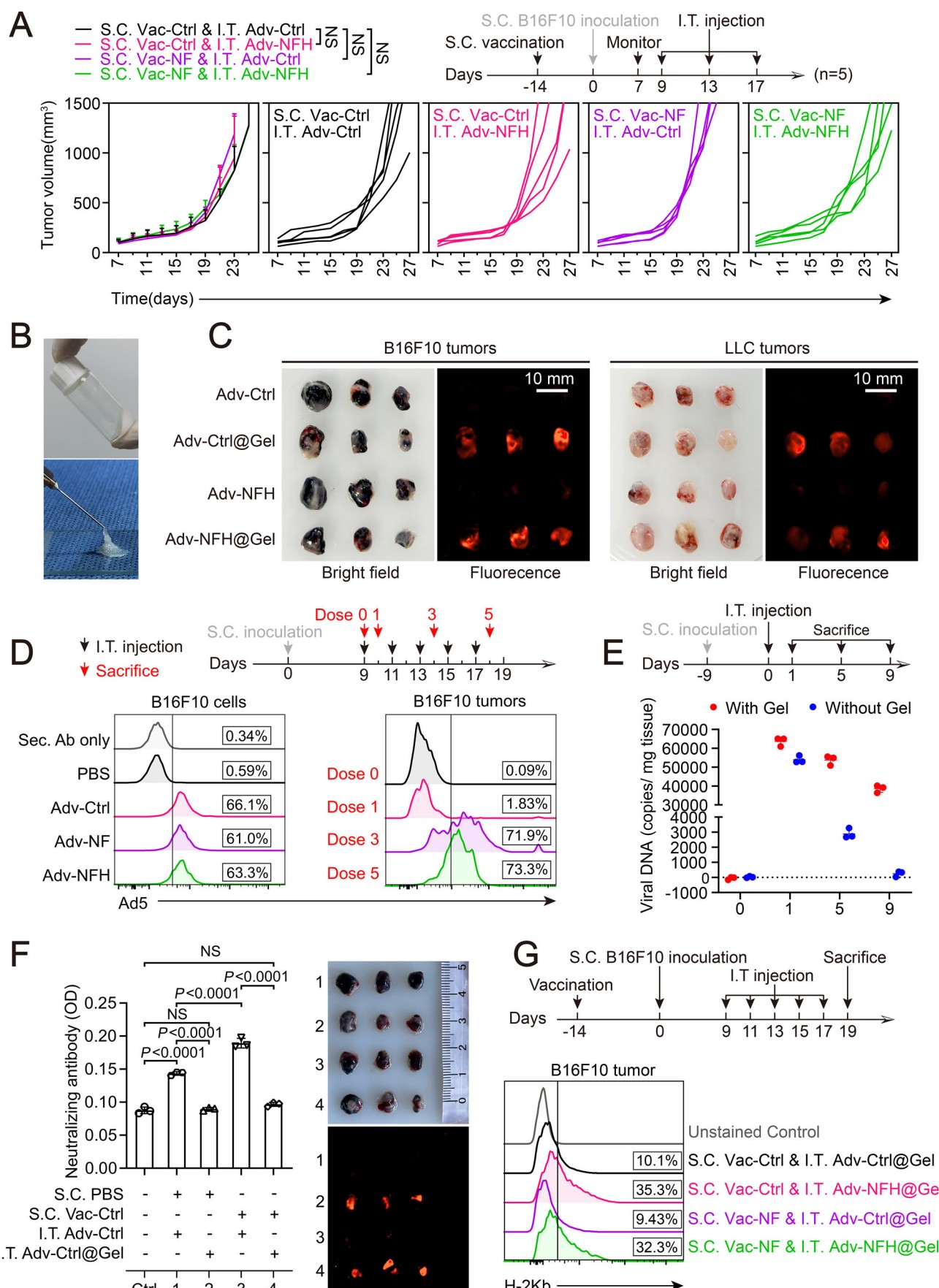

◄ **Figure EV3.** **Improving the efficacy of virotherapy with hydrogel encapsulation.**

(A) Overall and individual tumor growth curves demonstrated an integrative therapy of subcutaneous B16F10 tumor model with S.C. vaccination (Vac-Ctrl or Vac-NF, $2 \times 10^9$ VP) and I.T. virotherapy (Adv-Ctrl or Adv-NFH, $10^9$ VP) ($n = 5$). (B) Representative images demonstrated the successful preparation of injectable silk hydrogel. (C) Hydrogel encapsulation potentiated the intratumoral infection of adenoviruses ($n = 3$). (D) Quantification of intratumoral infection rate of hydrogel encapsulated Adv-Ctrl at early and late time point with an anti-adenovirus antibody (Merck, AB1056, 1:500) and Alexa FluorTM 488 secondary antibody (Thermo Fisher, A-11055, 2 µg/ml). Before the analysis, the functionality of the primary antibody was validated by staining B16F10 cells that have been infected with adenoviruses (Adv-Ctrl, Adv-NF, or Adv-NFH) in vitro at a MOI of 1 for 24 h (left panel). (E) qPCR assay quantified the viral DNA levels in the tumor tissues. Adv-Ctrl was prepared either in hydrogel (With Gel) or PBS (Without Gel) and injected into the tumors. Tumor tissues were subsequently collected at various time points for analysis. (F) Adenovirus neutralizing antibody assay and fluorescent imaging showed the protection of adenoviral vectors from host immunity by hydrogel encapsulation ($n = 3$). $P$ value: Ctrl vs. 1, $P = 3.380 \times 10^{-7}$; Ctrl vs. 2, $P = 0.982$; Ctrl vs. 4, $P = 0.199$; 1 vs. 2, $P = 4.752 \times 10^{-7}$; 1 vs. 3, $P = 2.034 \times 10^{-6}$; 3 vs. 4, $P = 1.457 \times 10^{-9}$. (G) MHC status of B16F10-established tumors following the intratumoral injection of hydrogel encapsulated adenoviruses. Statistical analysis was performed using one-way ANOVA (F) and two-way ANOVA (A). Data are shown as mean ± SD for 3 biological replicates.

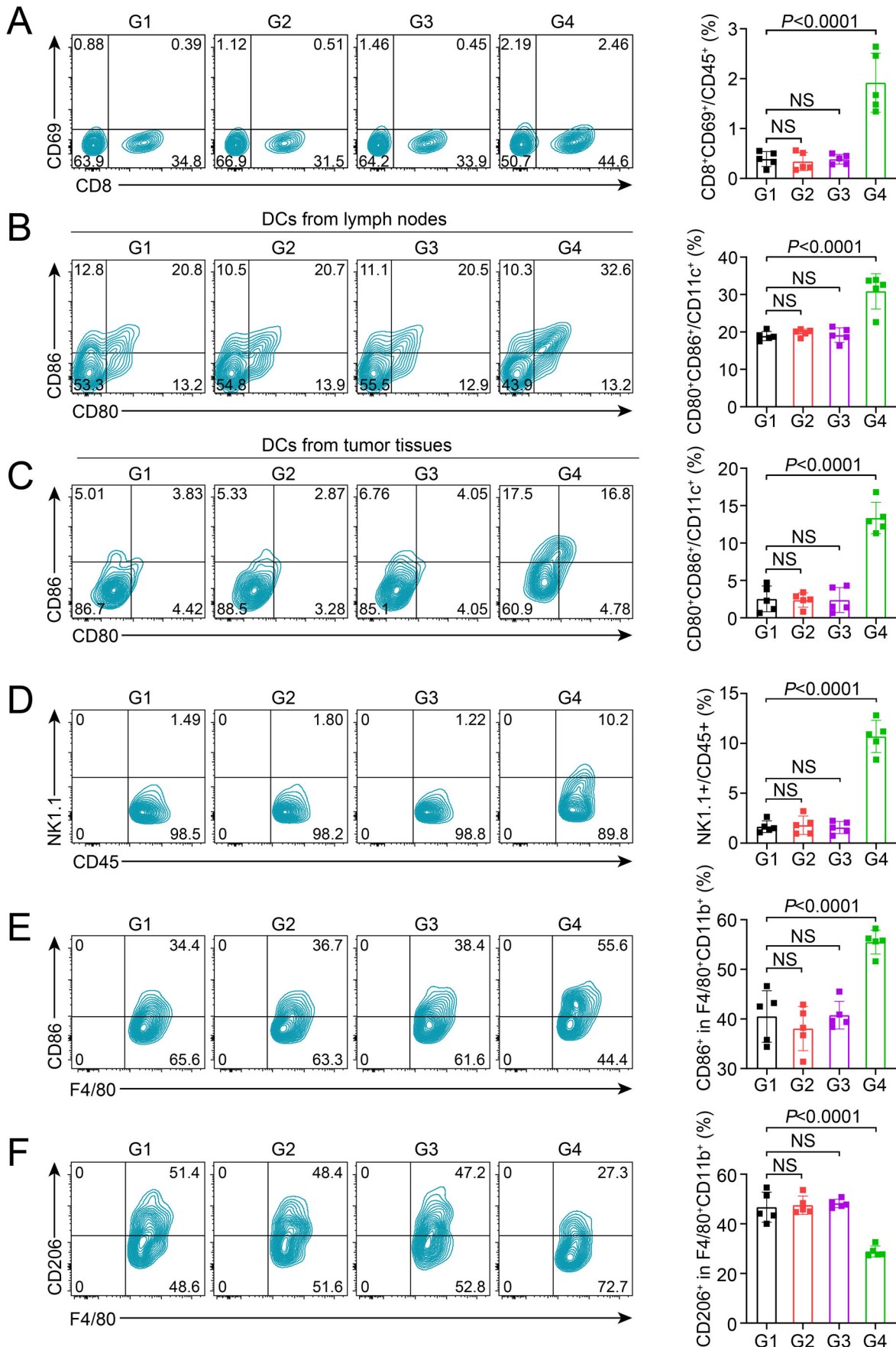

◀  **Figure EV4.  Integrative therapy remodeled the tumor microenvironment.**

Results of flow cytometry analysis showed the composition of CD69$^+$CD8$^+$ T cells in tumors (**A**), CD80$^+$CD86$^+$ DCs in lymph nodes (**B**), CD80$^+$CD86$^+$ DCs (**C**), NK1.1$^+$ NK cells (**D**), CD86$^+$ macrophages (**E**) and CD206$^+$ macrophages (**F**) in tumors ($n = 5$). PE-Cyanine7-conjugated anti-NK1.1 (eBioscience, 25-5941-82, 2.5 µg/ml), FITC-conjugated anti-F4/80 (BioLegend, 123107, 2.5 µg/ml), APC-conjugated anti-CD11b (eBioscience, 17-0112-82, 1.25 µg/ml), and PE-conjugated anti-CD206 (eBioscience, 12-2061-82, 1.25 µg/ml) antibodies were used for analysis. $P$ value (**A**): G2, $P = 0.995$; G3, $P = 0.999997$; G4, $P = 6.875 \times 10^{-6}$. $P$ value (**B**): G2, $P = 0.946$; G3, $P = 0.999$; G4, $P = 1.359 \times 10^{-5}$. $P$ value (**C**): G2, $P = 0.999$; G3, $P = 0.999$; G4, $P = 1.063 \times 10^{-7}$. $P$ value (**D**): G2, $P = 0.996$; G3, $P = 0.999$; G4, $P = 1.202 \times 10^{-9}$. $P$ value (**E**): G2, $P = 0.755$; G3, $P = 0.9996$; G4, $P = 8.186 \times 10^{-5}$. $P$ value (**F**): G2, $P = 0.987$; G3, $P = 0.924$; G4, $P = 8.309 \times 10^{-6}$. Statistical analysis was performed using one-way ANOVA (**A–F**). Data are shown as mean ± SD for 3 biological replicates.

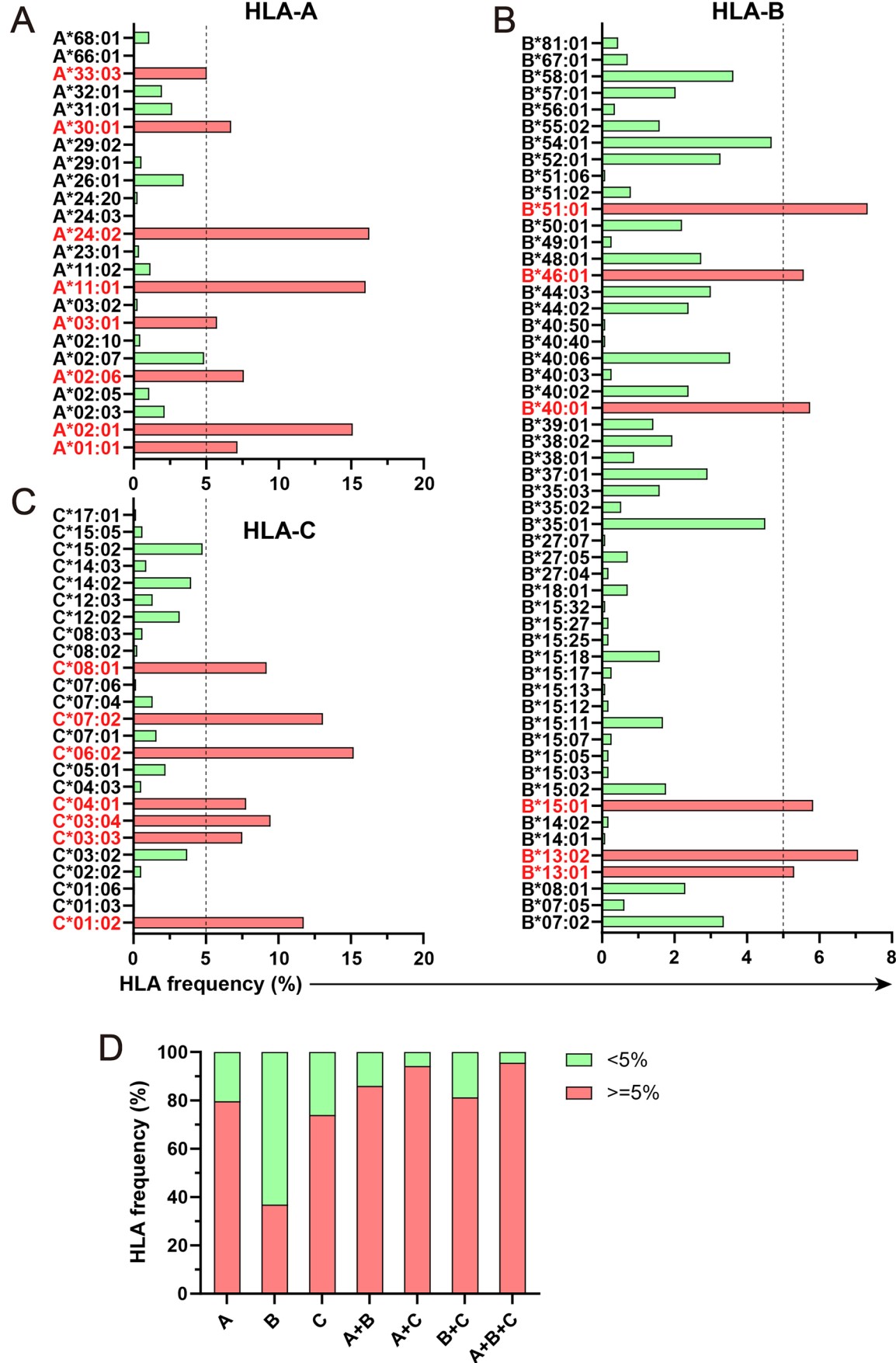

**Figure EV5. Coverage of HLA molecules indicates the applicability of the integrative therapy in human.**

Allele frequencies of HLA-A (**A**), HLA-B (**B**), and HLA-C (**C**) in the 741 individuals from Han population residing in Shanxi Province of China. The data was downloaded from PGG.MHC website: https://pog.fudan.edu.cn/pggmhc/#/population, and all the alleles were plotted. The 5% cut-off value was indicated by the dotted line. Alleles with frequencies <5% and >=5% were, respectively, indicated by green and red columns. The coverage of alleles with frequencies >=5% was calculated and demonstrated as cumulative histogram (**D**), in which HLA-A, HLA-B, and HLA-C were, respectively, abbreviated as A, B, and C. The cumulative results (>=5%) were as follows: A (79.59%), B (36.84%), C (73.94%), A + B (85.94%), A + C (94.17%), B + C (81.27%), A + B + C (95.58%).

