## [Peer Review File · EMBO Molecular Medicine]

Generalizing neoantigen-based tumor vaccine by delivering peptide-MHC complex via oncolytic virus

Chenyi Wang, Yingjun Shi, Da Zhang, Yupeng Sun, Junjie Xie, Bingchen Wu, Cuiling Zhang, and Xiaolong Liu

Corresponding author(s): Xiaolong Liu (liuxl@fjirsm.ac.cn) , Cuiling Zhang (sailingzhang2016@hotmail.com)

Review Timeline:

Submission Date:	22nd Aug 24
Editorial Decision:	2nd Oct 24
Revision Received:	20th Feb 25
Editorial Decision:	3rd Mar 25
Revision Received:	6th Mar 25
Accepted:	14th Mar 25

Editor: Zeljko Durdevic

Transaction Report:

2nd Oct 2024

Dear Prof. Liu,

Thank you for the submission of your manuscript to EMBO Molecular Medicine. We have now received feedback from the two of the three reviewers who agreed to evaluate the manuscript. As the referee #1 will unfortunately not be able to return his/her report in a timely manner, we prefer to make a decision now in order to avoid further delay in the process. As you will see from the reports below, both referees acknowledge potential interest of the study but also raise serious and partially overlapping concerns, particularly regarding the limited clinical implications of the study.

Overall, it is clear that publication of the paper cannot be considered at this stage. I also note that addressing the reviewers concerns in full will be necessary for further considering the manuscript in our journal and this appears to require a lot of additional work and experimentation. I am unsure whether you will be able or willing to address those and return a revised manuscript within the 6 months deadline. On the other hand, given the potential interest of the findings, I would be willing to consider a revised manuscript with the understanding that the referee concerns must be fully addressed.

Please note that further consideration of a revision that addresses reviewers' concerns in full will entail a second round of review. EMBO Molecular Medicine encourages a single round of revision only and therefore, acceptance or rejection of the manuscript will depend on the completeness of your responses included in the next, final version of the manuscript. For this reason, and to save you from any frustrations in the end, I would strongly advise against returning an incomplete revision and would also understand your decision if you chose to rather seek rapid publication elsewhere at this stage. If you would like to discuss further the points raised by the referees, I am available to do so via email or video. Let me know if you are interested in this option.

I look forward to seeing a revised form of your manuscript.

Should you find that the requested revisions are not feasible within the constraints outlined here and choose, therefore, to submit your paper elsewhere, we would welcome a message to this effect.

Yours sincerely,

Zeljko Durdevic

We require:

- 1) A .docx formatted version of the manuscript text (including legends for main figures, EV figures and tables). Please make sure that the changes are highlighted to be clearly visible.
- 2) Individual production quality figure files as .eps, .tif, .jpg (one file per figure). For guidance, download the 'Figure Guide PDF' (<https://www.embopress.org/page/journal/17574684/authorguide#figureformat>).
- 3) A .docx formatted letter INCLUDING the reviewers' reports and your detailed point-by-point responses to their comments. As part of the EMBO Press transparent editorial process, the point-by-point response is part of the Review Process File (RPF), which will be published alongside your paper.
- 4) A complete author checklist, which you can download from our author guidelines

(<https://www.embopress.org/page/journal/17574684/authorguide#submissionofrevisions>). Please insert information in the checklist that is also reflected in the manuscript. The completed author checklist will also be part of the RPF.

6) It is mandatory to include a 'Data Availability' section after the Materials and Methods. Before submitting your revision, primary datasets produced in this study need to be deposited in an appropriate public database, and the accession numbers and database listed under 'Data Availability'. Please remember to provide a reviewer password if the datasets are not yet public (see <https://www.embopress.org/page/journal/17574684/authorguide#dataavailability>).

7) For data quantification: please specify the name of the statistical test used to generate error bars and P values, the number (n) of independent experiments (specify technical or biological replicates) underlying each data point and the test used to calculate p-values in each figure legend. The figure legends should contain a basic description of n, P and the test applied. Graphs must include a description of the bars and the error bars (s.d., s.e.m.).

8) We would also encourage you to include the source data for figure panels that show essential data. Numerical data should be provided as individual .xls or .csv files (including a tab describing the data). For blots or microscopy, uncropped images should be submitted (using a zip archive if multiple images need to be supplied for one panel). Additional information on source data and instruction on how to label the files are available at

9) Our journal encourages inclusion of *data citations in the reference list* to directly cite datasets that were re-used and obtained from public databases. Data citations in the article text are distinct from normal bibliographical citations and should directly link to the database records from which the data can be accessed. In the main text, data citations are formatted as follows: "Data ref: Smith et al, 2001" or "Data ref: NCBI Sequence Read Archive PRJNA342805, 2017". In the Reference list, data citations must be labeled with "[DATASET]". A data reference must provide the database name, accession number/identifiers and a resolvable link to the landing page from which the data can be accessed at the end of the reference. Further instructions are available at

13) Author contributions: the contribution of every author must be detailed in a separate section (before the acknowledgments).

14) A Conflict of Interest statement should be provided in the main text.

Please also suggest a striking image or visual abstract to illustrate your article as a PNG file 550 px wide x 300-800 px high.

**** Reviewer's comments ****

Referee #2 (Remarks for Author):

In this manuscript, Wang et al claim a treatment modality aimed at universalizing neoantigen vaccine. By engineering oncolytic viruses that can deliver neoantigens and MHC molecules to tumour cells, this forced delivery enables targeted recognition by neoantigen-primed immune cells. When combined with neoantigen vaccination, these engineered viruses showed potent anti-tumour activity in various tumour models, regardless of endogenous neoantigen expression. The proposed strategy is a combination of previously reported tumor-modifying immunotherapeutic approaches that either elevates endogenous MHC levels [Cancers (Basel). 2020 Jun; 12(6): 1563] or induces artificial antigenicity in tumors [PNAS, 2022, 119 (26) e2116738119]. Integrating these strategies to the highly anticipated neoantigen vaccine field presents novelty on surface, but upon rumination it does not address the fundamental challenge in functional neoantigen identification. As the approach can be implemented with any immunogenic epitopes, the notion of adopting neoantigens seems less relevant. For instance, in the 2022 article [PNAS, 2022, 119 (26) e2116738119], Cuburu et al. leveraged a similar concept with antigens of cytomegalovirus, to which 70% of adults have preexisting immunity. In contrast, in the context of neoantigens, it is well acknowledged in the community that the fundamental barrier is the poor immunogenicity, particularly in terms of CD8 T cell induction, with predicted neoantigens. In other words, there are more concerns about whether a neoantigen could induce immunogenic responses at all in the clinical setting, and much less so for whether a proven immunogenic neoantigen may induce an anticancer effect. These challenges and concerns could render the authors' approach ineffective and dampen enthusiasm toward the authors' effort toward "universalizing neoantigen vaccines". Although the authors have provided substantive discussion, it is suggested that the authors further refine their narrative, acknowledge prior literature on MHC restoration and forced antigenicity, and in turn further differentiate the value of the present work. Based on the title of the work, I would expect the authors to at least rationalize a human-relevant neoantigen therapy design with broad applicability potential.

In terms of experimental and data presentation, further comments are provided below:

Major:

1. The authors are commendable for using several different cell lines to demonstrate the viability of the viral transduction therapy. In particular, Figure 6 shows human HepG2 cells in an immunocompromised model, which provides some clinical relevance. However, the use of mouse H2-Kb allele for the entirety of the study dampens the relevance of the work. The authors should examine the use of HLA-A02:01 alleles and HLA-A02:01-restricted epitopes in HepG2 (J Immunother Cancer. 2022; 10(2): e003978) for validation to highlight clinical relevance of the approach.
2. The text and figures do not align. There are only six figures and figure legends, but the results and discussion section contain seven figures. Mislabeled figure callouts begins at Figure 3 in the text and should be fixed.
3. The overall presentation of the cytotoxicity LDH assay is very misleading. The assay is routinely displayed in literature with x-y plot, but the authors use a heat map to display the results. Such display takes away the dose-dependent response, data points, and variance, which makes interpretation very difficult. For Figure 2D-H, Figure 5D, and several supplementary figures, cytotoxicity studies should be shown in line charts.
4. In Fig. EV1, the authors examined the activation of DCs following adenovirus infection. It is interesting to know that the control adenovirus (Adv-Ctrl) hardly activated DCs in terms of CD80/86 upregulation. As the only difference in the Adv-Ctrl is its missing of neoantigen expression, can the authors explain why it did not activate DCs?

Minor comments

1. In Fig. 5F, the vaccine-induced immunity was demonstrated by rechallenging cured mice with B16F10. However, it's intriguing to see that the tumours in the protected mice grew to ~100 mm³, then diminished subsequently. This contrasts with studies usually showing that pre-existing immunity can suppress tumour growth from the beginning. Is there any explanation for this phenomenon? In addition, can the B16F10-specific immune responses be observed in the cured mice?
2. Some of the T cell cytotoxicity studies with "culture medium" are confusing as they show LDH activity over 100% (160-180%). Could the authors elaborate on these results?
3. There are a few typos in the manuscript. One instance is in line 362 that shows "flow flow cytometry".

Referee #3 (Comments on Novelty/Model System for Author):

The value of the concept should be more detailed in models without the same CMH as the complex CMH neoepitope transferred to mimic the clinical human context.

Referee #3 (Remarks for Author):

The idea of a universal vaccination associated with an oncolytic virus expressing an MHC-neopeptide complex is original. But further experiments are needed to validate the concept.

Allogenic MHC transfer carries the risk of an allogeneic reaction that is not well controlled in the present study, particularly in the Balb/c nude mouse model without endogenous T cells (Fig 6).

The majority of experiments were performed using syngeneic tumours (B16F10) which is relevant for assessing the value of transducing neoepitopes but not the MHC-neopeptide complex. The results shown in Fig 4C in the LLC model should be investigated further as they are the most extrapolable to the clinical situation in humans.

Major Concerns

1 There is a lack of ADV control with MHC-Kb alone without neoepitopes to demonstrate the value of transferring the MHC-Kb-neoepitope complex.

2 The induction of a specific T cell response against neoepitopes needs to be better analyzed.

- What are the immunogenic neoepitopes? (Dissociate peptides in DC-T cell co-culture experiments)

- The experiment with the ptpn2 tetramer is not very convincing (Fig 5E). A clear population of specific T cell cannot be distinguished. This should be confirmed by an Elispot assay with the ptpn2 peptide or a control tetramer should be included.

- And whether there is a spreading epitope to reinforce the interest in inducing a specific T response beyond the neoepitopes.

3 In the LLC model (Fig 4C): Add a control with I.T ADV-NF (role of Kb) and I.T ADV-Kb alone.

Is there a memory response after rechallenge in this model?

4 Fig 6D: It is not clear why there is no difference between ADV-NFH and ADV-Ctrol.

5 Line 211: ' in Adv-NFH-treated (77% elimination) rather than Adv-NF-treated (30% elimination) LLC cells ' . How can 30% elimination be explained without Kb transfer because LLC expresses H2D, which does not bind to neoepitopes?

6 Line 131: Fig 1C and 1D: Is it not surprising that the presentation of neoepitopes is recognized by splenocytes only after 48h of coculture with DCs? Does this mean that the anti-neoepitope T cells were pre-existing?

7 In all the experiments with an Elispot read out, the ratio between the number of spot forming cells (SFC) and the total number of cells counted is not indicated (SFC number / nb cells?).

8 : The numbering of the figures is usually wrong (e.g. line 281, 316, 318330) There is no figure 7 as mentioned in the text.

***** Reviewers' comments *****

Referee #2

Remarks for Authors:

In this manuscript, Wang et al claim a treatment modality aimed at universalizing neoantigen vaccine. By engineering oncolytic viruses that can deliver neoantigens and MHC molecules to tumour cells, this forced delivery enables targeted recognition by neoantigen-primed immune cells. When combined with neoantigen vaccination, these engineered viruses showed potent anti-tumour activity in various tumour models, regardless of endogenous neoantigen expression. The proposed strategy is a combination of previously reported tumor-modifying immunotherapeutic approaches that either elevates endogenous MHC levels [Cancers (Basel). 2020 Jun; 12(6): 1563] or induces artificial antigenicity in tumors [PNAS, 2022, 119 (26) e2116738119]. Integrating these strategies to the highly anticipated neoantigen vaccine field presents novelty on surface, but upon rumination it does not address the fundamental challenge in functional neoantigen identification. As the approach can be implemented with any immunogenic epitopes, the notion of adopting neoantigens seems less relevant. For instance, in the 2022 article [PNAS, 2022, 119 (26) e2116738119], Cuburu et al. leveraged a similar concept with antigens of cytomegalovirus, to which 70% of adults have preexisting immunity. In contrast, in the context of neoantigens, it is well acknowledged in the community that the fundamental barrier is the poor immunogenicity, particularly in terms of CD8 T cell induction, with predicted neoantigens. In other words, there are more concerns about whether a neoantigen could induce immunogenic responses at all in the clinical setting, and much less so for whether a proven immunogenic neoantigen may induce an anticancer effect. These challenges and concerns could render the authors' approach ineffective and dampen enthusiasm toward the authors' effort toward "universalizing neoantigen vaccines". Although the authors have provided substantive discussion, it is suggested that the authors further refine their narrative, acknowledge prior literature on MHC restoration and forced antigenicity, and in turn further differentiate the value of the present work. Based on the title of the work, I would expect the authors to at least rationalize a human-relevant neoantigen therapy design with broad applicability potential.

Response: We sincerely appreciate for the reviewer's careful reading and the extremely helpful comments. Considering the extensive volume of the comments, we choose to address them in the following bullet points.

1. The fundamental challenge in functional neoantigen identification

For sure, we do agree with the reviewer that " it does not address the fundamental challenge in functional neoantigen identification". This is a focal research area in the broader scientific community, with numerous novel strategies being employed to screen for highly functional neoantigens. Our group also put lots of efforts to improve the identification of functional neoantigens with new algorithms and advanced

mass-spectrum based strategies. However, the identification of functional neoantigens is out of the scope of our manuscript. Actually, our current study focuses on how to expand the utilization of the immunogenic neoantigens that have already been identified and well-validated. In this context, a substantial body of existing studies has accumulated valuable neoantigen data, offering strong support for the implementation of the strategy proposed in this study. Consistent with relevant literature and reviewer's comments, the reactivity toward neoantigens might be heterogeneity in different individuals due to various mechanisms. Therefore, our strategy might not be able to achieve universalization of one particular neoantigen to a subpopulation, but actually could expand its utilization range. Accordingly, as demonstrated in our manuscript and in the response to Question 1 of the Major comments, we propose that co-delivery of multiple neoantigens by viral vectors could probably serve as a preventative strategy to increase error-tolerant rate of the neoantigen vaccines, thereby addressing potential inter-patient heterogeneity. Nevertheless, due to the concerns raised by the reviewer, we turn-down the description from "Universalization" to "Generalization" in the revised manuscript to preclude any unnecessary misunderstanding.

2. Implementation with any immunogenic epitopes

Meanwhile, we agree with the reviewer that the approach can be implemented with any immunogenic epitopes. We also recognized that other immunogenic epitopes, including antigens of human cytomegalovirus (HCMV) mentioned by the reviewer, may represent valuable alternatives for improving the therapeutic efficacy of the proposed strategy. Importantly, this does not contradict the findings of our strategy; rather, it underscores the versatility of our proposed strategy to some extent. In our current study, the neoantigens are selected due to their distinctive characteristics, including clinic-relevance, tumor specificity, and therapeutic potential. Particular for the therapeutic potential, the advantages of neoantigens in contrast to other immunogenic epitopes are evident in the following aspects:

(1) Avoidance of immune tolerance

The induction of immune tolerance by persistent infection is a critical mechanism in the development of some virus-mediated lesions (Oncology Letters, 2015, PMID: 26622540; Journal of Gasenterology and Hepatology, 2013, PMID: 23855309). In contrast, highly personalized nature and selective expression of neoantigens in tumor cells may reduce immune tolerance and the potential for autoimmunity (Nature Reviews Cancer, 2017, PMID: 28233802; Signal Transduction and Targeted Therapy, 2023, PMID: 36604431; Journal of Hematology & Oncology, 2019, PMID: 31492199).

(2) Dealing with HLA heterogeneity

HLA heterogeneity introduces significant uncertainty in antigen application due to allele-specific variations in epitope presentation. For instance, HLA-A03:01 and B07:02 are associated with elevated cervical cancer risks (Cancer Research, 2008, PMID: 18451182; Frontiers in Oncology, 2014, PMID: 24995157), while HLA-A*01:01 correlates with impaired SARS-CoV-2-specific CD8⁺ T-cell responses

and disease severity (Communications Biology, 2022, PMID: 36068292). These findings underscore the necessity for HLA-specific antigen customization. From this perspective, the expanding clinical data from neoantigen vaccine trials and established neoantigen database (Genomics, Proteomics & Bioinformatics, 2023, PMID: 36209954) now enable systematic HLA-epitope pairing analyses, forming a robust foundation to allow the achievement of this customization.

While HCMV antigens, as commented by the reviewer, hold promise for epitope selection due to high seroprevalence and robust T-cell responses, their clinical application may encounter notable challenges, especially in contexts where pre-existing immunity is heavily relied upon. In a study by Claire L. Gordon et al. (Journal of Experimental Medicine, 2017, PMID: 28130404), ~16.7% of HCMV-seropositive individuals (healthy before brain death) exhibited undetectable CMV-specific T cells across multiple tissues (blood, bone marrow, spleen, lymph nodes, lungs, intestines). Furthermore, HCMV gene products are detected in malignancies including glioblastoma, colorectal/prostate/breast/colon cancers (Current Opinion in Virology, 2019, PMID: 31525538). Mechanistically, HCMV drives oncogenesis through epithelial cell transformation, EMT/MET transitions, tumor angiogenesis, and immune evasion (Current Opinion in Virology, 2019, PMID: 31525538; Cellular & Molecular Immunology, 2014, PMID: 25418469). These findings suggest intratumoral delivery of HCMV epitopes may be redundant and therapeutically ineffective.

Therefore, the neoantigens in current study is just one choice of the therapeutic target selection over other immunogenic epitopes. Once the neoantigens are selected, the principles of selection and the working mechanisms of the neoantigens are totally related to “neoantigen” itself in our current study. Our group’s accumulated expertise in this field—including the prediction, validation, delivery, therapeutic application, mechanism study and clinical application of neoantigens—has laid the foundation for current study design of target selection and therapeutic application (Journal for ImmunoTherapy of Cancer, 2020, PMID: 32439798; Chinese Journal of Cancer Research, 2021, PMID: 34321833; Molecular Cancer, 2022, PMID: 34903219; Journal for ImmunoTherapy of Cancer, 2022, PMID: 36113894; EMBO Molecular Medicine, 2023, PMID: 37552209).

3. Poor immunogenicity of neoantigens

We fully acknowledge the reviewer's critical concern regarding the inherent immunogenicity limitations of neoantigens, which indeed represents a fundamental barrier in the development of neoantigen-based therapies. However, accumulating clinical evidence has demonstrated that a subset of neoantigens can elicit potent anti-tumor immunity, indicating at least some of the tumor neoantigens possess strong enough immunogenicity to effectively activate immune system for tumor therapy in clinical settings. Importantly, the therapeutic strategy proposed in this study aims to leverage existing dataset of neoantigens to select appropriate antigens from those that have already been validated as immunogenic for broader application. From this perspective, even though the majority of neoantigens may have poor immunogenicity,

it would not impede the application of the therapeutic strategy proposed in this study.

Interestingly, by elucidating the global landscape of anti-tumor T-cell responses in complete regression of HPV-associated metastatic cervical cancer after tumor-infiltrating adoptive T-cell therapy, Sanja Stevanović et al. found out that the immunodominant T-cell reactivity was directed against mutated neoantigens or a cancer-germline antigen, rather than the canonical viral antigens underlying the disease. In addition, T cells targeting viral tumor antigens did not display preferential *in vivo* expansion during cancer regression. As discussed by the authors, these results might suggest that viral proteins are not necessarily more immunogenic than alternative classes of tumor antigens (Science, 2017, PMID: 28408606). Accordingly, we believe that selection of immunogenic neoantigens for the expanded application, as proposed in our current study, is highly achievable.

4. Further value differentiation of our work

Being aware of the necessity to further differentiate the value of our work, as suggested by the reviewer, we have refined our narrative and acknowledged prior literature on MHC restoration and forced antigenicity in the revised manuscript (Page 11, Line 435-447). Following the comments of the reviewer (“*a human-relevant neoantigen therapy design*”), we have recruited 3 HLA-A*02:01⁺ donors and examined the use of HLA-A*02:01 allele and HLA-A*02:01-restricted epitopes in HepG2 and SNU-449 cells to validate the clinical relevance of our approach. By priming naïve T cells (CD3⁺CD8⁺CD62L⁺CD45RA⁺) with a pool of 7 selected neoantigen peptides, including 2 neoantigens from the 2022 article recommended by the reviewer (KYV and ALL, *J Immunother Cancer*. 2022; 10(2): e003978), we observed neoantigen-specific T cell reactivity in all the 3 donors, with varying specificity and intensity. Delivery of HLA-A*02:01 and the HLA-A*02:01-restricted neoantigens by recombinant adenovirus could effectively elicit the cytotoxicity of the neoantigen-primed T cells to eliminate the virus-infected tumor cells (HepG2 and SNU-449 cells). For the detailed experimental results, please refer to the response to the Question 1 of the Major comments.

Major comments:

*Q1. The authors are commendable for using several different cell lines to demonstrate the viability of the viral transduction therapy. In particular, Figure 6 shows human HepG2 cells in an immunocompromised model, which provides some clinical relevance. However, the use of mouse H2-Kb allele for the entirety of the study dampens the relevance of the work. The authors should examine the use of HLA-A02:01 alleles and HLA-A02:01-restricted epitopes in HepG2 (*J Immunother Cancer*. 2022; 10(2): e003978) for validation to highlight clinical relevance of the approach.*

Response: Thanks very much for the reviewer’s constructive comments. We appreciate the reviewer for recognition of the benefit of using different cell lines and the clinical relevance of the Figure 6 in the manuscript. Following the suggestion of

the reviewer, we have examined the use of HLA-A*02:01 allele and HLA-A*02:01-restricted epitopes in HepG2 cells for validation to highlight clinical relevance of our approach.

By carefully analyzing the article indicated by the reviewer (J Immunother Cancer. 2022; 10(2): e003978), we chose two HLA-A*02:01-restricted epitopes for validation, including KYV (EXT2_D>Y, KYVYDFGVSV) and ALL (SLC39A11_P>L, ALLFLESEL). However, the results presented in the Fig. 6 of the study showed that in vitro antigen presentation of KYV and ALL peptides to T cells from HLA-A*02:01⁺ healthy donors induced neoantigen-specific immune responses in only 25% (two out of eight) of the donors. To further ensure the validation, we consulted additional reports (Science, 2015, PMID: 25837513; Science, 2016, PMID: 27198675) and accordingly expanded our analysis to 5 more HLA-A*02:01-restricted epitopes, such as TMEM48_F>L (CLNEYHLFL), SEC24A_P>L (FLYNLLTRV), TKT_R>W (AMFWSVPTV), ASTN1_P>L (KLYGLDWAEL), SMARCD3_H>Y (KLFEFVLVYGV). Noteworthy, in contrast to the report from J Immunother Cancer, the 2016 Science article reported that analysis of T cell reactivity from four donors revealed three to five neoantigen-specific T cell response in all cases (CDK4_R>L, ASTN1_P>L, and GNL3L_R>C), confirming the existence of qualified neoantigens for broad application of our strategy.

Next, peptides of the selected 7 HLA-A*02:01-restricted neoantigens were synthesized (GenScript) for in vitro antigen presentation and validation of neoantigen-reactivity of the primed T cells. Meanwhile, the sequence of the selected neoantigens were concatenated with furin cleavage sites, flanked by the signal peptide and transmembrane domain of HLA-A*02:01, and synthesized for the construction of recombinant adenoviruses.

Afterwards, we recruited 10 healthy donors, 3 of whom were validated to be HLA-A*02:01⁺ (Weihe Biotechnology INC.). In vitro antigen presentation was conducted following the methodology described by Muhammad Ali et al. (Nature Protocols, 2019, PMID: 31101906). This approach emphasizes the importance of T cell naivety and the DC:T cell ratio for the efficient induction of neoantigen-reactive T cells. Minor modifications were made to adapt the protocol to the specific conditions of our laboratory. Briefly, autologous DCs and naïve T cells (CD3⁺CD8⁺CD62L⁺CD45RA⁺) were isolated from peripheral blood mononuclear cells (PBMCs) by plastic adherence method (Molecular Cancer, 2022, PMID: 35148751) and flow cytometry-based cell sorting (Blood, 2013, PMID: 23160470), respectively. Activated by TNF α (10 ng/ml) and LPS (10 ng/ml), the DCs were pulsed with neoantigen peptides (10 μ g/ml for each peptide) (Cancer Research, 2005, PMID: 15930317) and co-cultured with naïve T cells at a ratio of 1:2 for 9 days in KBM581 supplemented with IL-21 (30 ng/ml), IL-7 (5 ng/ml) and IL-15 (5 ng/ml). On day 10, the co-cultured cells were re-stimulated with peptide-pulsed DCs at a ratio of 2:1 for additional 3 d before subjecting to subsequent analysis.

Co-culturing the resulting cell populations of three HLA-A*02:01⁺ donors with T2 cells pulsed with individual peptides (Fig. 1A) revealed clear T cell reactivity toward the selected neoantigens, as evidenced by flow cytometry analysis of IFN γ

production. This reactivity was observed across all three donors, albeit with donor-specific variations in intensity and epitope preference. In contrast, such reactivity was negligible in the control cultures (Fig. 1B). Noteworthy, TMEM48_F>L and SLC39A11_P>L elicited more stable and potent T cell reactivity than other neoantigens. The specificity of these responses was further validated using peptide-HLA-A*02:01 tetramer staining, with TMEM48_F>L- and SLC39A11_P>L-specific T cells constituting 4.4-8.47% and 4.98-14.8% of total peptide-primed CD8⁺ T cells, respectively (Fig. 1C). Align with the observations, T cells primed with the selected neoantigens exhibited potent cytotoxicity against peptide-pulsed T2 cells, as measured by LDH release assays and flow cytometry analysis (Fig. 1D and 1E). These results collectively confirm the successful priming of naïve T cells and the immunogenicity of the selected neoantigens.

To determine whether forced transfer of peptide-MHC complex could redirect the cytotoxicity of the peptide-primed T cells, recombinant replication-defective adenovirus (Adv-hNFH) and oncolytic adenovirus (OAv-hNFH) co-expressing HLA-A*02:01 and the selected HLA-A*02:01-restricted neoantigens were constructed to infect tumor cells (Fig. 1F). As controls, recombinant adenoviruses expressing only the selected neoantigens (Adv-hNF) or HLA-A*02:01 (Adv-HLA) were also constructed. The expression of the transgenes was validated by qPCR analysis in virus-infected SNU-449 cells (HLA-A*11:01, 31:01, Genome Medicine, 2015, PMID: 26589293), a human hepatocellular carcinoma cell line negative for HLA-A*02:01 (Fig. 1G). As demonstrated in the Fig. 1H, when HLA-A*02:01-negative SNU449 cells were used as target cells for cytotoxicity assay, the peptide-primed T cells exhibited significant cytotoxicity toward Adv-hNFH- and OAv-hNFH-infected cells, while showing negligible cytotoxicity against the cells pre-treated by PBS, Adv-Ctrl, Adv-hNF, Adv-HLA, or OAv-Ctrl. In contrast, no notable cytotoxicity was observed when the pre-infected SNU-449 cells were co-cultured with PBS-treated control T cells. Notably, consistent with results shown in Fig. 6E of our manuscript, OAv-hNFH elicited superior cytotoxicity of the peptide-primed T cells compared to Adv-hNFH in SNU-449 cells. Such cytotoxicity of the peptide-primed T cells was similarly observed in HLA-A*02:01-positive HepG2 cells (HLA-A*02:01, 24:02, Oncoimmunology, 2014, PMID: 25960936), with the exception of the Adv-hNF group (Fig. 1I). Align with the findings presented in Fig. EV3D in our previous version of manuscript, transfer of neoantigens in tumor cells endogenously expressing neoantigen-binding MHC molecules by Adv-NF elicited cytotoxicity of the primed T cells, albeit to a lesser extent than Adv-hNFH and OAv-hNFH.

Overall, by leveraging HLA-A*02:01 alleles and HLA-A*02:01-restricted epitopes for validation, our results compellingly demonstrate the feasibility of our strategy within human system, underscoring its promising translational potential for clinical application. Noteworthy, consistent with the findings reported by Erlend Strønen et al. (Science, 2016, PMID: 27198675), the immunogenic reactivity toward a given neoantigen peptide can vary significantly among individuals. While some neoantigens elicit reactivity in only a subset of individuals, others may demonstrate

reactivity in the majority of individuals (if not all). Although the factors driving this variability remain unclear, the use of highly immunogenic neoantigens or the inclusion of multiple neoantigens within the expression vector can effectively address these variations, thereby broadening the applicability of the therapeutic strategy proposed in this study.

We have included the results as **Fig. 7** in the revised manuscript (**Page 10, Line 385-418; Page 11, Line 419-433; Page 29, Line 1200; Page 30, Line 1201-1220**).

Fig. 1 | Preclinical validation in human cell-based system with HLA-A*02:01 allele and HLA-A*02:01-restricted epitopes (also as Fig. 7 in the revised manuscript).

(A) Treatment scheme for in vitro antigen presentation and peptide-pulsed T2 re-stimulation. (B) IFN γ staining demonstrated the T cell reactivity to individual neoantigen peptides (n=3). (C) Peptide-HLA-A*02:01 tetramer staining showed the

specificity of T cell reactivity. (D) LDH assay showed the elimination of peptide-pulsed T2 cells by the primed T cells (n=3). Following a 24 h co-culture (E:T=3:1), the culture medium was collected for the assessment of LDH activity. Cell viability was determined by calculating the ratio of LDH activity in the culture medium to that in the lysate of the control T2 cells. (E) CFSE staining demonstrate the cytotoxicity of peptide-primed T cells towards peptide-pulsed T2 cells. Following the in vitro antigen presentation, T cells of donor 1 were labeled with a low concentration of CFSE (0.5 μ M) and co-cultured with peptide-pulsed T2 cells labeled with a high concentration of CFSE (5 μ M) at a 3:1 ratio. Selective elimination of the peptide-pulsed T2 cells was measured 24 h later by flow cytometry analysis. (F) Schematic view of adenoviral vectors. (G) qPCR analysis validated the expression of HLA-A*02:01 and HLA-A*02:01-restricted neoantigens in virus-infected SNU-449 cells (n=3). (H) Cytotoxicity assay demonstrated selective elimination of Adv-hNFH- and OAv-hNFH-infected SNU-449 cells by peptide-primed T cells (n=3). SNU-449 cells were pre-treated by the PBS or indicated viruses (1 MOI for 48 h) and co-cultured with the peptide-primed T cells at a ratio of 1:3 for 24 h before subjecting to LDH assay. (I) Peptide-primed T cells selectively eliminate HepG2 cells that had been infected by Adv-hNF, Adv-hNFH, and OAv-hNFH (n=3). The procedure for cell treatment was identical to that described in Figure H. Data are shown as mean \pm SD for 3 biological replicates.

Q2. The text and figures do not align. There are only six figures and figure legends, but the results and discussion section contain seven figures. Mislabelled figure callouts begins at Figure 3 in the text and should be fixed.

Response: We sincerely appreciate the reviewer's constructive comments and have revised the manuscript accordingly.

Q3. The overall presentation of the cytotoxicity LDH assay is very misleading. The assay is routinely displayed in literature with x-y plot, but the authors use a heat map to display the results. Such display takes away the dose-dependent response, data points, and variance, which makes interpretation very difficult. For Figure 2D-H, Figure 5D, and several supplementary figures, cytotoxicity studies should be shown in line charts.

Response: Thanks very much. Following the reviewer's suggestion, we have revised the presentation of the LDH assay results in the manuscript, changing it from a heat map to line charts.

Q4. In Fig. EV1, the authors examined the activation of DCs following adenovirus infection. It is interesting to know that the control adenovirus (Adv-Ctrl) hardly activated DCs in terms of CD80/86 upregulation. As the only difference in the Adv-Ctrl is its missing of neoantigen expression, can the authors explain why it did not activate DCs?

Response: Thanks a lot for the reviewer's insightful comments. According to the results in the Figure EV10A of our manuscript, which demonstrated the upregulation of CD80/86 in DCs following treatment with a serial doses of Adv-Ctrl, the dose of Adv-Ctrl plays a critical role in limiting viral backbone-induced DC activation. Our results indicate that DC maturation exhibits a clear upward trend with increasing viral titers. However, at viral titers ≤ 5 MOI, no significant difference in DC maturation was observed between the virus-treated and PBS-treated groups. Notably, in Figure EV1, a viral titer of 2 MOI was used for DC infection, which may have been insufficient to effectively activate DCs.

Our results align well with a report by Jing Ma et al. (Cell Death & Disease, 2020, PMID: 31969562). In their study (Supplementary Figure 10), the authors verified that direct infection of human DCs with various viruses, including vaccinia virus, wild-type Adenovirus serotype 5 (Ad5), Semliki Forest virus (SFV), and Vaccinia virus (VV), as assessed by examining the up-regulation of CD80/86, did not promote DC maturation.

Minor comments:

Q1. In Fig. 5F, the vaccine-induced immunity was demonstrated by rechallenging cured mice with B16F10. However, it's intriguing to see that the tumours in the protected mice grew to ~ 100 mm³, then diminished subsequently. This contrasts with studies usually showing that pre-existing immunity can suppress tumour growth from the beginning. Is there any explanation for this phenomenon? In addition, can the B16F10-specific immune responses be observed in the cured mice?

Response: Thanks a lot for the reviewer's insightful comments. For the rechallenge of B16F10 cells in the cured mice, the appearance of tumor growth could probably be attributed to the use of Matrigel, which is well-documented to gel at the site of injection and increase the prevalence and the growth rate of the inoculated tumor cells (British Journal of Cancer, 1994, PMID: 8054270). The gelation of the injected Matrigel could have delayed the absorption as well as the elimination of tumor cells, contributing to the pseudoprogession of the inoculated tumor. Additionally, the infiltration of immune cells into the gelled Matrigel may induce local inflammation and edema, resulting in an apparent increase in tumor size.

For the evaluation of B16F10-specific immune response, as commented by the reviewer, we repeated the integrative therapy in B16F10-established tumor model and obtained the SMNCs from different groups of mice for analysis. As shown in the Fig. 2, compare to the T cells obtained from control groups (G1: S.C. Vac-Ctrl & I.T. Adv-Ctrl@Gel; G2: S.C. Vac-Ctrl & I.T. Adv-NFH@Gel; G3: S.C. Vac-NF & I.T. Adv-Ctrl@Gel), T cells obtained from the experimental group (G4: S.C. Vac-NF & I.T. Adv-NFH) exhibited significant cytotoxicity toward co-cultured B16F10 cells, indicating the existence of B16F10-specific immune responses in the cured mice. To further validate the results, neoantigen peptides of B16F10 cells were synthesized according to a recent report from Andrew Redenti et al. (Nature, 2024, PMID:

39415001), including Pcmt1_P222L (VSFAPLVQL), Haus6_L176V (VARNRVQI), Ncor1_H673P (FNYKRRPNL), Vps13c_S1256P (SSLPTNAVVV), Ctsd_G397S (VSFANAVVL), and Nsun2_K72M (KILRMSPL). By stimulating the obtained SMNCs with peptide-pulsed DCs, as demonstrated in the Fig. 3), cytosolic IFN γ staining revealed obvious T cell reactivity toward the neoantigens of B16F10 cells in the experimental group (G4). These findings were highly in line with the results of B16F10 rechallenge in C57BL/6 mice, confirming the induction of B16F10-specific immune response by the integrative immunotherapy.

We have included the results as **Appendix Fig. S8** in the revised manuscript (Page 7, Line 290-293).

Fig. 2 | T cells obtained from mice treated by integrative immunotherapy exhibited B16F10-specific cytotoxicity (also as Appendix Fig. S8A in the revised manuscript).

B16F10 cells were subcutaneously inoculated into the right flanks of C57BL/6 mice, which were pre-immunized with one dose of Vac-Ctrl or Vac-NF (2×10^9 VP per mouse) for 14 d. Adv-Ctrl or Adv-NFH (10^9 VP/dose) were intratumorally injected into the established tumor tissues on day 9, day 11, day 13, day 15 and day 17 post tumor inoculation, respectively. SMNCs were obtained from the treated mice on day 20, stimulated with anti-CD3 ϵ /anti-CD28 antibodies and IL-2 for 3 d, and maintained in KPM581 containing 1000 U/ml IL-2. Thereafter, the obtained T cells were co-cultured with B16F10 (10^5 cells) at a ratio of 3:1 in a 24-well plate for 48 h before subjecting to LDH cell cytotoxicity assay (n=3). Data are shown as mean \pm SD for 3 biological replicates.

Fig. 3 | T cell reactivity toward B16F10-derived neoantigens (also as Appendix Fig. S8B in the revised manuscript).

DCs obtained from naïve mice were treated by a pool of neoantigen peptides (10 nM/peptide) for 24 h, and co-cultured with the freshly prepared SMNCs of treated mice at a ratio of 1:5 for additional 24 h. Subsequently, the cells were treated with GolgiStop Protein Transport Inhibitor (4 ul/6 ml culture medium) for 4 h before the collection of the cells, stained with fluorophore-conjugated antibodies, and subjected to flow cytometry analysis (n=5). Data are shown as mean±SD for 5 biological replicates.

Q2. Some of the T cell cytotoxicity studies with "culture medium" are confusing as they show LDH activity over 100% (160-180%). Could the authors elaborate on these results?

Response: Thanks very much for the reviewer's comments. These results represented the relative LDH levels released into the culture medium as a result of tumor cell lysis, expressed as a percentage relative to the control culture medium, which was standardized to 100%.

Q3. There are a few typos in the manuscript. One instance is in line 362 that shows "flow flow cytometry".

Response: Thanks very much for the reviewer's comments. We have revised the manuscript accordingly (Page 9, Line 355).

Referee #3

Comments on Novelty/Model System for Author:

The value of the concept should be more detailed in models without the same CMH as the complex CMH neoepitope transferred to mimic the clinical human context.

Response: Thanks very much for the reviewer's insightful comments. We fully agree with the reviewer's suggestion that the value of our concept should be validated in models without the same MHC to mimic the clinical human context. Actually, the neoantigens used in this study were identified from Hepa1-6 cells, which have an

MHC-I haplotype of H-2Kb/H-2Db (Christian F. Grimm et al., *Gastroenterology*, 2000, PMID: 11040197). To emulate clinical settings, we have already demonstrated the feasibility of our strategy in models established with human tumor cell lines, including HepG2, SK-Hep-1, and SMMC-7721 cells, none of which share the MHC haplotype of Hepa1-6 cells, as described in our manuscript.

The MHC type of the human cell lines are as follows:

HepG2 (Sebastian Boegel et al., *Oncoimmunology*, 2014, PMID: 25960936): HLA-A (A*02:01, 24:02), HLA-B (B*35:14, 51:01), HLA-C (C*04:01, 16:02);

SK-Hep-1 (https://www.cellosaurus.org/CVCL_0525): HLA-A (A*02:01, 24:02), HLA-B (B*35:04, 44:03:01), HLA-C (C*04:01);

SMMC-7721 (Hai-Long Dong et al., *Cancer Biol Ther.*, 2004, PMID: 15219945): HLA-A2+.

Remarks for Author

The idea of a universal vaccination associated with an oncolytic virus expressing an MHC-neopeptide complex is original. But further experiments are needed to validate the concept.

Allogenic MHC transfer carries the risk of an allogeneic reaction that is not well controlled in the present study, particularly in the Balb/c nude mouse model without endogenous T cells (Fig 6).

The majority of experiments were performed using syngeneic tumours (B16F10) which is relevant for assessing the value of transducing neoepitopes but not the MHC-neoepitope complex. The results shown in Fig 4C in the LLC model should be investigated further as they are the most extrapolable to the clinical situation in humans.

Response: Grateful for the reviewer's constructive comments. We sincerely appreciate the reviewer for recognition of the originality of our current study.

For the risk of allogeneic reaction as commented by the reviewer, we have included several preventive measures in our manuscript, which are outlined below for clarification:

1. As illustrated in the Fig. 6A of our manuscript, a tumor-specific promoter (hTERT) has been used to limit the expression of neoantigens within tumor cells.

2. Intratumoral injection, which has been demonstrated to be the most efficient and safest method for administering oncolytic viruses in both basic and clinical research (*Journal of Clinical Investigation*, 2019, PMID: 30829653; *Pharmaceutics*, 2021, PMID: 34959474), was applied in the current study for allogenic MHC transfer.

3. Adenoviruses have been encapsulated within silk hydrogel to restrict virus dissemination in surrounding nontarget tissues. To validate the restricted dissemination of the encapsulated virus, we conducted a repetition of intratumoral injection in C57BL/6 mice using Adv-Ctrl@Gel and obtained tumor tissues (established with LLC or B16F10 cells), organs, and serum samples. By extracting the

genomic DNA, we quantified the titration of Ad5 in the samples by qPCR method, employing purified Adv-Ctrl that titrated via plaque assay as a standard curve. As illustrated in the Fig. 4 below, after 5 times of intratumoral injection, viral distribution was exclusively observed within the tumor tissues. We have included the results as **Appedix Fig. S5** in the revised manuscript (**Page 7, Line 260**).

4. The safety of the therapeutic treatment has been validated by histological and biochemical analysis in both C57BL/6 and BALB/c nude mice.

5. The primary goal of MHC transfer was to overcome intratumoral heterogeneity and prevent immune escape caused by downregulated or lost expression of MHC-I molecules. Accordingly, the animal models in our manuscript were designed to demonstrate the feasibility and applicability of our strategy. Nevertheless, to minimize the risk of allogeneic reactions and enable effective integrative therapy with neoantigen vaccines in clinical settings, only autologous MHC molecules would theoretically be transferred into tumor cells via oncolytic viruses.

For the assessment of the value of transducing neoantigen-MHC complex, as suggested by the reviewer, we have constructed adenovirus vectors expressing neoantigens (Adv-NF) or H-2Kb (Adv-H2Kb) alone to compare with the therapeutic effect of Adv-NFH. **To further investigate the results shown in Fig. 4C in the LLC model**, we included both Adv-NF and Adv-H2Kb for intratumoral injection to repeat the experiments in the Fig. 4C. Align with the results of Fig. EV3D in the previous version of our manuscript, the results demonstrated that both Adv-NF and Adv-NFH significantly inhibited tumor growth in Vac-NF-immunized mice, while Adv-H2Kb exhibited no antitumor activity. Importantly, Adv-NFH was more effective than Adv-NF ($P < 0.001$), confirming the therapeutic potential of transducing neoantigen-MHC complex. For the detailed experimental results, please refer to the response to the Question 3 of Major Concerns.

In addition to LLC model, we further validated the applicability of our approach in human cell-based system with HLA-A*02:01 allele and HLA-A*02:01-restricted epitopes validated in published reports (Science, 2015, PMID: 25837513; Science, 2016, PMID: 27198675; The Journal for ImmunoTherapy of Cancer, 2022, PMID: 35193931), including TMEM48_F>L (CLNEYHLFL), SEC24A_P>L (FLYNLLTRV), TKT_R>W (AMFWSVPTV), EXT2_D>Y (KYVYDFGVSV), SLC39A11_P>L (ALLFLESEL), ASTN1_P>L (KLYGLDWAEL), and SMARCD3_H>Y (KLFEFVYGV).

By priming naïve T cells ($CD3^+CD8^+CD62L^+CD45RA^+$) of HLA-A*02:01⁺ donors with neoantigen peptide-pulsed autologous DCs for 2 rounds (Fig. 5A), we successfully obtained neoantigen-specific human T cells, as validated by flow cytometry analysis of IFN γ production (Fig. 5B), tetramer staining (Fig. 5C), and cytotoxicity assay in T2 cells pulsed with individual peptides (Fig. 5D-E). Albeit with donor-specific variations in intensity and epitope preference, the neoantigen-specific reactivity was observed across all the enrolled three HLA-A*02:01⁺ donors, demonstrating TMEM48_F>L and SLC39A11_P>L as dominant epitopes within the

selected neoantigens.

To determine whether forced transfer of peptide-MHC complex could redirect the cytotoxicity of the peptide-primed T cells, recombinant replication-defective adenovirus (Adv-hNFH) and oncolytic adenovirus (OAv-hNFH) co-expressing HLA-A*02:01 and the selected HLA-A*02:01-restricted neoantigens were constructed to infect tumor cells (Fig. 5F). Adenoviruses expressing only the selected neoantigens (Adv-hNF) or HLA-A*02:01 (Adv-HLA) were referred as controls. HLA-A*02:01-negative SNU-449 (HLA-A*11:01, 31:01, Genome Medicine, 2015, PMID: 26589293) and HLA-A*02:01-positive HepG2 (HLA-A*02:01, 24:02, Oncoimmunology, 2014, PMID: 25960936) cells were used as target tumor cells. Following the validation of adenovirus-mediated transgene delivery in virus-infected SNU-449 cells using qPCR analysis (Fig. 5G), cytotoxicity assay was performed by co-culturing the virus-infected SNU-449 or HepG2 cells with the neoantigen-primed T cells. As demonstrated in the Fig. 5H, in HLA-A*02:01-negative SNU449 cells, the primed T cells exhibited significant cytotoxicity toward Adv-hNFH- and OAv-hNFH-infected cells, while showing negligible cytotoxicity against the cells pre-treated by PBS, Adv-Ctrl, Adv-hNF, Adv-HLA, or OAv-Ctrl. In contrast, no notable cytotoxicity was observed when the pre-infected SNU-449 cells were co-cultured with PBS-treated control T cells. Notably, consistent with results shown in Fig. 6E of our manuscript, OAv-hNFH elicited superior cytotoxicity of the peptide-primed T cells compared to Adv-hNFH in SNU-449 cells. Such cytotoxicity of the peptide-primed T cells was similarly observed in HLA-A*02:01-positive HepG2 cells, with the exception of the Adv-hNF group (Fig. 5I). Align with the findings presented in Fig. EV3D in our previous version of manuscript, transfer of neoantigens in tumor cells endogenously expressing neoantigen-binding MHC molecules by Adv-NF elicited cytotoxicity of the primed T cells, albeit to a lesser extent than Adv-hNFH and OAv-hNFH.

Overall, by leveraging HLA-A*02:01 alleles and HLA-A*02:01-restricted epitopes for validation, our results compellingly demonstrate the feasibility of our strategy within human system, underscoring its promising translational potential for clinical application.

We have included the results as **Fig. 7** in the revised manuscript (Page 10, Line 385-418; Page 11, Line 419-433; Page 29, Line 1200; Page 30, Line 1201-1220).

Fig. 4 | Tissue distribution of intratumorally injected adenovirus (also as Appendix Fig. S5 in the revised manuscript). PBS and Adv-Ctrl (10^9 VP) were injected into LLC- or B16F10-established tumors five times before the mice were sacrificed. Genomic DNA was extracted from the tumors and primary organs of the mice, and a qPCR assay was performed to quantify the amount of viral DNA. Treatment with PBS and Adv-Ctrl is referred to as I.T. dose 0 and I.T. dose 5, respectively ($n=4$). Data are shown as mean \pm SD.

Fig. 5 | Preclinical validation in human cell-based system with HLA-A*02:01 allele and HLA-A*02:01-restricted epitopes (also as Fig. 7 in the revised manuscript).

(A) Treatment scheme for in vitro antigen presentation and peptide-pulsed T2 re-stimulation. (B) IFN γ staining demonstrated the T cell reactivity to individual neoaantigen peptides (n=3). (C) Peptide-HLA-A*02:01 tetramer staining showed the

specificity of T cell reactivity. (D) LDH assay showed the elimination of peptide-pulsed T2 cells by the primed T cells (n=3). Following a 24 h co-culture (E:T=3:1), the culture medium was collected for the assessment of LDH activity. Cell viability was determined by calculating the ratio of LDH activity in the culture medium to that in the lysate of the control T2 cells. (E) CFSE staining demonstrate the cytotoxicity of peptide-primed T cells towards peptide-pulsed T2 cells. Following the in vitro antigen presentation, T cells of donor 1 were labeled with a low concentration of CFSE (0.5 μ M) and co-cultured with peptide-pulsed T2 cells labeled with a high concentration of CFSE (5 μ M) at a 3:1 ratio. Selective elimination of the peptide-pulsed T2 cells was measured 24 h later by flow cytometry analysis. (F) Schematic view of adenoviral vectors. (G) qPCR analysis validated the expression of HLA-A*02:01 and HLA-A*02:01-restricted neoantigens in virus-infected SNU-449 cells (n=3). (H) Cytotoxicity assay demonstrated selective elimination of Adv-hNFH- and OAv-hNFH-infected SNU-449 cells by peptide-primed T cells (n=3). SNU-449 cells were pre-treated by the PBS or indicated viruses (1 MOI for 48 h) and co-cultured with the peptide-primed T cells at a ratio of 1:3 for 24 h before subjecting to LDH assay. (I) Peptide-primed T cells selectively eliminate HepG2 cells that had been infected by Adv-hNF, Adv-hNFH, and OAv-hNFH (n=3). The procedure for cell treatment was identical to that described in Figure H. Data are shown as mean \pm SD for 3 biological replicates.

Major Concerns

Q1. There is a lack of ADV control with MHC-Kb alone without neoepitopes to demonstrate the value of transferring the MHC-Kb-neoepitope complex.

Response: Thanks a lot for the reviewer's constructive comments. Although the lack of a control adenovirus expressing H-2Kb alone initially limited the comprehensiveness of our analysis, we previously investigated and discussed the functionality of the transduced neoantigen peptide-MHC complex in the earlier version of our manuscript. Specifically, we compared the ability to redirect the cytotoxicity of neoantigen-immunized T cells (VT-NF) among PBS, Adv-Ctrl, Adv-NF, and Adv-NFH in HepG2 and LLC cells, as illustrated in Fig. EV3D of the earlier version of our manuscript. According to the reviewers' comments, we constructed a control adenovirus expressing H-2Kb alone (Adv-H2Kb) and incorporated it into the above experiments, enabling a more detailed analysis of the functionality of the neoantigen peptide-MHC molecules.

As demonstrated in the Fig. 6A, both Adv-H2Kb and Adv-NFH significantly upregulated the expression H-2Kb in LLC cells, validating the functionality of Adv-H2Kb in transferring H-2Kb molecules. Noteworthy, LLC (Lewis lung carcinoma) cell line originated spontaneously in the lung of a C57BL mouse. According to a report from Lea Eisenbach et al. (International Journal of Cancer, 1983, PMID: 6862690), the MHC type of LLC cell line was H-2Kb/H-2Db. In our results, compared to the unstained group, the flow cytometry histogram for LLC cells showed a rightward shift following the staining by a H-2Kb antibody (Biolegend, 116505,

clone AF6-88.5, 5 µg/ml), with an H-2Kb positivity rate of approximately 4%. These findings are consistent with the qPCR validation shown in Fig. EV3A and the results of flow cytometry analysis presented in Fig. EV3B of our previous manuscript, validating low-level expression of H-2Kb protein in LLC cells.

As illustrated in the Fig. 6B, HepaG2 and LLC cells were pre-treated with PBS, Adv-Ctrl, Adv-NF, Adv-H2Kb, or Adv-NFH before subjecting to the co-culture with VT-Ctrl and VT-NF cells. Align with the results in the Fig. EV3D of our previous manuscript, obvious cytotoxicity was observed only in VT-NF that co-cultured with Adv-NF/Adv-NFH-infected LLC cells or Adv-NFH-infected HepG2 cells, and neither VT-Ctrl nor VT-NF cells exhibited cytotoxicity against Adv-H2Kb-infected cells. Importantly, Adv-NFH demonstrates superior efficacy by inducing a stronger VT-NF cytotoxicity in LLC cells, which express H-2Kb but lack Hepa1-6 neoantigens, compared to Adv-NF (74% elimination versus 40% elimination). Meanwhile, it elicited robust VT-NF cytotoxicity in HepG2 cells (~65% elimination), which express neither H-2Kb nor Hepa1-6 neoantigens. These findings underscored the value of transferring neoantigen peptide-MHC complexes.

We have included the results as **Fig. EV1D** in the revised manuscript (Page 5, Line 200).

Fig. 6 | LDH assay validated the functionality of transferred peptide-MHC complex (also as Fig. EVID in the revised manuscript).

(A) LLC cells were treated with PBS, Adv-Ctrl (1 MOI), Adv-NF (1 MOI), Adv-H2Kb (1 MOI), or Adv-NFH (1 MOI) for 24 h before subjecting to flow cytometry analysis. (B) Splenic T cells, including VT-Ctrl and VT-NF, were obtained from Vac-Ctrl- and Vac-NF-immunized C57BL/6 mice (2×10^9 VP per mice), respectively. The cells were co-cultured with HepG2 cells or LLC cells, which were pre-treated with 1 MOI of indicated viruses for 48 h, at a ratio of 3:1 for 48 h before the measurement of cell viability ($n=3$). Statistical analysis was performed using two-way ANOVA. Data are shown as mean \pm SD for 3 biological replicates.

Q2. The induction of a specific T cell response against neoepitopes needs to be better analyzed.

- What are the immunogenic neoepitopes? (Dissociate peptides in DC-T cell co-culture experiments)
- The experiment with the ptpn2 tetramer is not very convincing (Fig 5E). A clear

population of specific T cell cannot be distinguished. This should be confirmed by an Elispot assay with the ptpn2 peptide or a control tetramer should be included.

- And whether there is a spreading epitope to reinforce the interest in inducing a specific T response beyond the neoepitopes.

Response: Thanks very much for the reviewer’s careful reading and the very constructive comments. According to the comments, we have performed additional experiments and listed the results in the following bullet points.

1. What are the immunogenic neoepitopes? Dissociate peptides in DC-T cell co-culture experiments.

Actually, all the 7 neoantigens used in the current study have been validated to be immunogenic in a previous publication of our lab (*Journal of Immunotherapy of Cancer*, 2022, PMID: 36113894). In that report, we synthesized 20 long peptides of neoantigen candidates (17 amino acids in length) to immunize C57BL/6 mice. Following the obtaining of the splenic T cells, the immunogenicity of the neoantigen candidates were assessed by subjecting the cells to ELISpot assay with individual peptides, which revealed that 7 out of 20 neoantigen peptides could elicit significant immune responses in the immunized splenic T cells.

According to the reviewer’s comments, we immunized C57BL/6 mice with Adv-Ctrl/Adv-NF and obtained the splenic T cells to reassess the immunogenicity of the neoepitopes within the adenoviral vector. Concretely, the T cells from Adv-Ctrl-treated mice (VT-Ctrl) or Adv-NF-treated mice (VT-NF) were co-cultured with DCs pulsed with individual neoantigen peptides, and the IFN γ production in the T cells was evaluated via flow cytometry analysis. As demonstrated in the Fig. 7, except Samd91_K752M, production of intracellular IFN γ was significantly upregulated in VT-NF cells in response to the stimulation by all the other peptides, with Traf7_C403W and Ptpn2_I383T showing particularly pronounced effects. Overall, these results are consistent with our previous report, confirming the immunogenicity of the selected neoantigens.

We have included the results as **Appedix Fig. S4** in the revised manuscript (Page 4, Line 148-151).

Fig. 7 | Immunogenicity of neoepitopes within adenoviral vaccine (also as

Appedix Fig. S4 in the revised manuscript).

DCs were pulsed with 100 nM of indicated neoantigen peptides for 24 h before subjecting to the co-culture with VT-Ctrl or VT-NF cells at a ratio of 1:5 for additional 24 h. The cell culture medium was supplemented with GolgiStop Protein Transport Inhibitor (4 μ l per 6 ml culture medium) 4 h before the collection of the cells. Subsequent to the collection, the cells were stained with fluorophore-conjugated antibodies for flow cytometry analysis (n=5). Statistical analysis was performed using two-way ANOVA. Data are shown as mean \pm SD for 5 biological replicates.

2. *The experiment with the ptpn2 tetramer is not very convincing (Fig 5E). A clear population of specific T cell cannot be distinguished. This should be confirmed by an Elispot assay with the ptpn2 peptide or a control tetramer should be included.*

According to the reviewer's comments, we repeated the experiment in Fig. 5E of our manuscript and subjected the isolated TILs to ELISpot assay with the Ptpn2 peptide. As shown in the Fig. 8, TILs obtained from the experimental group (G4: S.C. Vac-NF+I.T. Adv-NFH@Gel) exhibited a more potent immune response than control groups (G1: S.C. Vac-Ctrl + I.T. Adv-Ctrl@Gel; G2: S.C. Vac-Ctrl + I.T. Adv-NFH@Gel; G3: S.C. Vac-NF + I.T. Adv-Ctrl@Gel) in response to the stimulation of Ptpn2 peptide, corroborating the tetramer staining results in the Fig. 5E of our manuscript.

We have included the results as **Appedix Fig. S9** in the revised manuscript (Page 8, Line 316-318).

Fig. 8 | ELISpot assay validated the neoantigen responsiveness of the TILs obtained from mice treated with the integrative immunotherapy (also as Appedix Fig. S9 in the revised manuscript).

C57BL/6 mice were immunized with Vac-Ctrl or Vac-NF 14 days before the inoculation of B16F10 cells (3×10^6 cells per mouse). The established tumors were

intratumorally treated with therapeutic adenoviruses (Adv-Ctrl or Adv-NFH) at 10^9 VP for 5 times. The spleens and treated tumors were obtained on day 20 post the inoculation of B16F10 cells for further analysis. Following the cell isolation, TILs (5×10^4 cells) were subjected to ELISpot assay by co-culturing with DCs pulsed with Ptpn2 peptides ($4 \mu\text{g}$ per 10^4 cells, 24 h) for 24 h. As described in the manuscript, experiments were performed according to the manufacturer's instructions (Mabtech, 3321-4APT-2). NC: negative control in which DCs were treated with PBS instead of peptides. PC: positive control, where TILs were stimulated with anti-CD3 ϵ ($2.5 \mu\text{g}/\text{ml}$) and anti-CD28 ($1 \mu\text{g}/\text{ml}$) ($n=3$). Data are shown as mean \pm SD for 3 biological replicates.

3. And whether there is a spreading epitope to reinforce the interest in inducing a specific T response beyond the neoepitopes.

To assess whether there is a spreading epitope following the integrative immunotherapy, we reviewed relevant literature to identify neoantigens uniquely expressed by B16F10 cells, which have been extensively used in our current study. According to a report from Andrew Redenti et al. (Nature, 2024, PMID: 39415001), we selected and synthesized 6 neoantigen peptides of B16F10 cells based on their predicted MHC-I IC50 (14.98 nM to 72.41 nM), including Pcmt1_P222L (VSFAPLVQL), Haus6_L176V (VARNRFVQI), Ncor1_H673P (FNYKRRPNL), Vps13c_S1256P (SSLPTNAVVV), Ctsd_G397S (VSFANAVVL), and Nsun2_K72M (KILRMSPL). These peptides were mixed in equal proportion as a pool to pulse naïve DCs to stimulate splenic mononuclear cells (SMNCs), which were obtained in parallel with the TILs as described in the Fig. 8.

As shown in the Fig. 9, splenic CD8⁺ T cells obtained from the experimental group (G4: S.C. Vac-NF + I.T. Adv-NFH@Gel) exhibited a significantly stronger immune response to B16F10 peptide stimulation compared to the control groups (G1: S.C. Vac-Ctrl + I.T. Adv-Ctrl@Gel; G2: S.C. Vac-Ctrl + I.T. Adv-NFH@Gel; G3: S.C. Vac-NF + I.T. Adv-Ctrl@Gel), confirming the occurrence of epitope spreading following the integrative immunotherapy with the neoantigens of Hepa1-6 cells.

We have included the results as **Appendix Fig. S8B** in the revised manuscript (Page 7, Line 290-293).

Fig. 9 | Antigen responsiveness of the splenic CD8⁺ T cells obtained from therapeutically treated mice (also as Appendix Fig. S8B in the revised manuscript).

The synthesized B16F10 peptides were prepared in the culture medium as a pool to treat naïve DCs for 24 h. The final concentration of each peptide in the culture medium was 10 nM. Meanwhile, following the obtaining of SMNCs from the treated mice, the SMNCs were co-cultured with the peptide-pulsed DCs at a ratio of 5:1 for 24 h. The cells were thereafter collected and stained by fluorophore-conjugated antibodies for flow cytometry analysis (n=5). Statistical analysis was performed using two-way ANOVA. Data are shown as mean±SD.

Q3. In the LLC model (Fig 4C): Add a control with I.T ADV-NF (role of Kb) and I.T ADV-Kb alone.

Reponse: Thanks very much for the reviewer’s very constructive comments. According to the comments, we added both I.T. Adv-NF and I.T. Adv-H2Kb in the LLC model, and repeated the experiments in the Fig. 4C of our manuscript. As demonstrated in the Fig. 10, I.T. Adv-NF significantly inhibited the growth of LLC tumor in Vac-NF-immunized mice rather than in Vac-Ctrl-immunized mice. In contrast, no therapeutic effect of I.T. Adv-H2Kb was observed in both Vac-Ctrl- and Vac-NF-immunized mice. These results were well in line with the results in the Fig. 4, confirming the therapeutic value of transferring neoantigen peptide-MHC complexes.

We have included the results as **Appendix Fig. S7** in the revised manuscript (Page 7, Line 272-273).

Fig. 10 | Integrative immunotherapy in LLC-established tumor model (also as Appendix Fig. S7 in the revised manuscript).

The experiments were performed by following the procedure as described in the Fig. 4C (n=5 mice per group). Statistical analysis was performed using two-way ANOVA

(tumor growth curve) or one-way ANOVA (tumor weight). Data are shown as mean±SD.

Q4. Fig 6D: It is not clear why there is no difference between ADV-NFH and ADV-Ctrol.

Response: Thanks very much for the reviewer’s insightful comments. Actually, in the Fig. 6D of our manuscript, a significant difference between Adv-NFH and Adv-Ctrl was observed when infected HepG2 cells were co-cultured with VT-NF cells, although this difference was relatively small compared to the difference between the groups of oncolytic adenoviruses (OAv-NFH vs OAv-Ctrl). By placing the results of replication defective adenoviruses (Adv-NFH and Adv-Ctrl) and oncolytic adenoviruses on the same axis for comparison, the visual prominence of the difference between Adv-NFH and Adv-Ctrl was reduced. As shown in the Fig. 11, when the data of Adv-NFH and Adv-Ctrl from Fig. 6D of our manuscript are plotted and analyzed separately, the difference between the two groups becomes more pronounced.

Fig. 11 | Data of Adv-Ctrl and Adv-NFH from Fig. 6D of our manuscript. Statistical analysis was performed using one-way ANOVA.

Q5. Line 211: ' in Adv-NFH-treated (77% elimination) rather than Adv-NF-treated (30% elimination) LLC cells '. How can 30% elimination be explained without Kb transfer because LLC expresses H2D, which does not bind to neoepitopes?

Response: Thanks very much for the reviewer’s comments. As described above in the response to the Question 1 of major concerns, the LLC (Lewis lung carcinoma) cell line, which originated spontaneously in the lung of a C57BL mouse, has been reported by Lea Eisenbach et al. (International Journal of Cancer, 1983, PMID: 6862690) to express the H-2Kb/H-2Db MHC type. In our experiments, flow cytometry analysis

revealed a rightward shift in the histogram for LLC cells stained with an H-2Kb antibody (Biolegend, 116505, clone AF6-88.5, 5 µg/ml) compared to the unstained control. As demonstrated in the Fig. 12, the H-2Kb positivity rate was approximately 4%. These observations align with the qPCR data shown in Fig. EV3A and the flow cytometry results presented in Fig. EV3B of our previous version of manuscript, collectively confirming the low-level expression of the H-2Kb protein in LLC cells.

Therefore, even though without H-2Kb transferring, 30% elimination in Adv-NF-treated LLC cells is reasonable.

Fig. 12 | H-2Kb staining in LLC cells.

Q6. Line 131: Fig 1C and 1D: Is it not surprising that the presentation of neoepitopes is recognized by splenocytes only after 48h of coculture with DCs? Does this mean that the anti-neoepitope T cells were pre-existing?

Response: Thanks a lot for the reviewer's comments. The experiments in Fig. 1C and 1D were conducted using Mabtech ELISpot plates (Catalog number 3321-4APT-2) to assess the ability of T cells to secrete IFN γ following in vitro antigen presentation. According to the literature, dendritic cells (DCs) present antigens to naive T cells, which can undergo at least 7 times of division and differentiate into effector and memory T cells without further antigenic stimulation (Susan M. Kaech et al., *Nature Immunology*, 2001, PMID: 11323695). IFN- γ secretion by T cells can occur either during the antigen presentation process or following antigen re-stimulation after differentiation. Concretely, naive murine T cells can rapidly and transiently produce low levels of IFN γ upon antigen stimulation, peaking at ~8 h and declining by 24 h (Julie M. Curtsinger et al., *The Journal of Immunology*, 2012). After differentiation into effector or memory T cells, expression of IFN γ can be detected by flow cytometry as early as 5 h following antigen re-stimulation (Susan M. Kaech et al., *Nature Immunology*, 2001, PMID: 11323695).

Although our results in the Fig. 1C and 1D of the manuscript only demonstrated ELISpot data at the 48 h time point, this does not imply that splenocytes only recognize the presented neoantigens after 48 h of co-culture with DCs. To further investigate the dynamic changes in IFN γ expression within antigen-presented T cells in our experimental condition, we repeated the in vitro antigen presentation experiment and performed flow cytometry analysis of T cells at 24, 48, and 72 h

post-co-culture. As illustrated in the Fig. 13A and 13B, the number of IFN γ ⁺ T cells in the experimental groups (Adv-NF and Adv-NGS) was significantly higher than in the control groups (PBS and Adv-Ctrl), displaying a clear time-dependent trend. These results suggest that splenocytes recognize the presentation of neoantigens earlier than 48 h, and this recognition may progressively intensify with prolonged stimulation by DCs overexpressing neoantigens. Therefore, the selection of 48 h time point in the Fig. 1C and 1D of our manuscript represented a time window where IFN γ secretion was sufficiently established during antigen presentation and reliably measurable using the ELISpot analysis.

To verify whether the anti-neoepitope T cells were pre-existing, we co-cultured the freshly prepared splenocytes with B16F10 cells pretreated with PBS, Adv-Ctrl, Adv-NF or Adv-NGS. As shown in Fig. 13C, flow cytometry analysis of IFN γ production revealed no obvious T cell reactivity toward the overexpression of the selected neoantigens, suggesting there's no pre-existing anti-neoepitope T cells in the freshly prepared splenocytes.

We have included the results as **Appendix Fig. S2** in the revised manuscript (Page 3, Line 120-122; Page 4, Line 123-125).

Fig. 13 | Recognition of the presentation of neoepitopes by splenocytes (also as Appendix Fig. S2 in the revised manuscript).

(A) Flow cytometry analysis demonstrated T cell reactivity toward the antigen presentation by pre-treated DCs. Treated with PBS or 2 MOI indicated adenoviruses for 72 h, 1×10^4 DCs were co-cultured with 5×10^4 freshly prepared SMNCs for in vitro antigen presentation. The cells were collected at 24 h, 48 h, or 72 post the co-culture and subjected to flow cytometry analysis. (B) Numeric results of Figure A (n=3). (C) T cell reactivity toward the overexpression of neoantigens in B16F10 cells. B16F10 cells were pre-treated with PBS or 1 MOI indicated viruses before the co-culture. Data are shown as mean \pm SD for 3 biological replicates.

Q7. In all the experiments with an Elispot read out, the ratio between the number of

spot forming cells (SFC) and the total number of cells counted is not indicated (SFC number / nb cells?).

Response: Thanks very much for the reviewer's comments. We have revised the manuscript according to the comments.

Q8. The numbering of the figures is usually wrong (e.g. line 281, 316, 318330) There is no figure 7 as mentioned in the text.

Response: Thanks very much for the reviewer's comments. We have revised the manuscript accordingly.

3rd Mar 2025

Dear Dr. Liu,

Thank you for the submission of your revised manuscript to EMBO Molecular Medicine. I am pleased to inform you that we will be able to accept your manuscript pending the following final amendments:

- 1) Authors: Please provide institutional email address for the co-corresponding author Cuiling Zhang. Also, add institutional e-mail addresses of both corresponding authors on the title page of the manuscript.
- 2) Author Checklist: Please complete the form by entering information about Author, Journal and Manuscript Number at the top of the page.
- 3) In the main manuscript file, please do the following:
 - Please address all comments suggested by our data editors listed below:
 - o Figure legends:
 1. Please note that the exact p values are not provided in the legends of figures 1E, H, I, M; 2C, 3E, 4C, H; 5B, C, E, G, I; 6L, EV3 F, EV4 A-F.
 2. Please note that information related to n is missing in the legends of figures 3E, J.
 3. Please note that the error bars are not defined in the legends of figures 3E, J; 4A, B, C, D, E, F, H.
 - Figure callouts should be in sequential order. Currently, Fig EV2 is called out before Fig EV1C, D. Please correct.
 - Author contributions: Please remove it from the manuscript and specify author contributions in our submission system. CRediT has replaced the traditional author contributions section because it offers a systematic machine-readable author contributions format that allows for more effective research assessment. You are encouraged to use the free text boxes beneath each contributing author's name to add specific details on the author's contribution. More information is available in our guide to authors:
<https://www.embopress.org/page/journal/17574684/authorguide#authorshipguidelines>
 - Please include structured Methods section that includes a Reagents and Tools Table (should be uploaded as a separate file) followed by a Methods and Protocols section. More information on how to adhere to this format as well as downloadable templates (.docx) for the Reagents and Tools Table can be found in our author guidelines:
<https://www.embopress.org/page/journal/17574684/authorguide#structuredmethods>
An example of a paper with Structured Methods can be found here:
<https://www.embopress.org/doi/full/10.1038/s44320-024-00037-6#sec-4>
 - In data availability section please replace the current sentence with following; This study includes no data deposited in external repositories.
- 4) Appendix: Please submit the file in PDF format.
- 5) The Paper Explained: Please add it to the main manuscript file.
- 6) Synopsis:
 - Synopsis image: Please format the image to 550 px-wide x (300 - 600)-px high and upload it as a high-resolution JPEG file.
 - Please check your synopsis text and image before submission with your revised manuscript. Please be aware that in the proof stage minor corrections only are allowed (e.g., typos).
- 7) As part of the EMBO Publications transparent editorial process initiative (see our Editorial at <http://embomolmed.embopress.org/content/2/9/329>), EMBO Molecular Medicine will publish online a Review Process File (RPF) to accompany accepted manuscripts. This file will be published in conjunction with your paper and will include the anonymous referee reports, your point-by-point response and all pertinent correspondence relating to the manuscript. Let us know whether you agree with the publication of the RPF and as here, if you want to remove or not any figures from it prior to publication. Please note that the Authors checklist will be published at the end of the RPF.
- 8) Please provide a point-by-point letter INCLUDING my comments as well as the reviewer's reports and your detailed responses (as Word file).

I look forward to reading a new revised version of your manuscript as soon as possible.

Yours sincerely,

Zeljko Durdevic

*** Instructions to submit your revised manuscript ***

- 1) a .docx formatted version of the manuscript text (including Figure legends and tables)
 - 2) Separate figure files*
 - 3) supplemental information as Expanded View and/or Appendix. Please carefully check the authors guidelines for formatting Expanded view and Appendix figures and tables at <https://www.embopress.org/page/journal/17574684/authorguide#expandedview>
 - 4) a letter INCLUDING the reviewer's reports and your detailed responses to their comments (as Word file).
 - 5) The paper explained: EMBO Molecular Medicine articles are accompanied by a summary of the articles to emphasize the major findings in the paper and their medical implications for the non-specialist reader. Please provide a draft summary of your article highlighting
 - the medical issue you are addressing,
 - the results obtained and
 - their clinical impact.This may be edited to ensure that readers understand the significance and context of the research. Please refer to any of our published articles for an example.
 - 6) Author contributions: the contribution of every author must be detailed in a separate section.
 - 7) EMBO Molecular Medicine now requires a complete author checklist (<https://www.embopress.org/page/journal/17574684/authorguide>) to be submitted with all revised manuscripts. Please use the checklist as guideline for the sort of information we need WITHIN the manuscript. The checklist should only be filled with page numbers where the information can be found. This is particularly important for animal reporting, antibody dilutions (missing) and exact values and n that should be indicated instead of a range.
 - 8) Every published paper now includes a 'Synopsis' to further enhance discoverability. Synopses are displayed on the journal webpage and are freely accessible to all readers. They include a short stand first (maximum of 300 characters, including space) as well as 2-5 one sentence bullet points that summarise the paper. Please write the bullet points to summarise the key NEW findings. They should be designed to be complementary to the abstract - i.e. not repeat the same text. We encourage inclusion of key acronyms and quantitative information (maximum of 30 words / bullet point). Please use the passive voice. Please attach these in a separate file or send them by email, we will incorporate them accordingly.
- You are also welcome to suggest a striking image or visual abstract to illustrate your article. If you do please provide a jpeg file 550 px-wide x 300-600px high.
- 9) A Conflict of Interest statement should be provided in the main text
 - 10) Please note that we now mandate that all corresponding authors list an ORCID digital identifier. This takes <90 seconds to complete. We encourage all authors to supply an ORCID identifier, which will be linked to their name for unambiguous name identification.

Currently, our records indicate that the ORCID for your account is 0000-0002-3096-4981.

Please click the link below to modify this ORCID:
Link Not Available

11) Include a Reagents and Tools Table as part of the Methods section, which can be downloaded from our author guidelines (<https://www.embopress.org/page/journal/17574684/authorguide#structuredmethods>)

Photos 400-800 DPI

*Additional important information regarding figures and illustrations can be found at <https://bit.ly/EMBOPressFigurePreparationGuideline>. See also figure legend preparation guidelines: <https://www.embopress.org/page/journal/17574684/authorguide#figureformat>

***** Reviewer's comments *****

Referee #2 (Comments on Novelty/Model System for Author):

The authors have added an HLA-A02:01 model system that adequately establishes the clinical relevance of the proposed approach. Cytolysis study has also been updated to include proper statistical analysis.

Referee #2 (Remarks for Author):

I appreciate the significant efforts the authors have made to address the concerns, refine the manuscript, and incorporate additional experiments to strengthen the study. The update of the article narrative from "universalization" to "generalization" better contextualizes the study in the broader literature of neoantigen research. Furthermore, the inclusion of HLA-A*02:01-restricted epitope validation substantially enhances the translational relevance of their findings. Overall, the revisions and additional experiments sufficiently address my concerns, and I find the manuscript suitable for publication.

Referee #3 (Remarks for Author):

The authors have satisfactorily addressed my various concerns

Dear Dr. Zeljko Durdevic,

Thanks very much for your editorial team's thorough evaluation of our manuscript entitled "Generalizing neoantigen-based tumor vaccine by delivering peptide-MHC complex via oncolytic virus" (Manuscript reference number: EMM-2024-20458-V2). We sincerely appreciate the extremely constructive comments from you and the reviewers.

We have carefully revised the manuscript and supplemented with required information according to your and data editors' comments, with all changes highlighted in red for your convenience. Noteworthy, in order to ensure that the figure callouts are in sequential order, Fig. EV1D has been relocated to Fig. EV2 and is now designated as Fig. EV2D in the revised manuscript. Meanwhile, the point-to-point responses to all the specific comments arisen by the reviewers have been listed below for your editorial checking.

According to your comments, we have added institutional E-mail addresses of both corresponding authors on the title page of the manuscript. However, we respectfully ask for your permission to use the corresponding authors' personal email addresses instead of the institutional email addresses for this submission, as we have consistently used personal email addresses for our submissions, including our previous articles published in *Nature Communications*, *Cell Reports Medicine*, as well as *EMBO Molecular Medicine*. More importantly, the corresponding authors' personal email addresses that we provide have been associated with the corresponding authors' unique ORCID. Therefore, in order to maintain our publication record to facilitate academic communication, we hope to obtain your consent to use the corresponding authors' personal email address for this submission.

Thanks again for considering accepting our manuscript for publication in *EMBO Molecular Medicine*. We are sincerely grateful for the time and effort you've dedicated to the review of our work.

Sincerely yours,

Xiaolong Liu, Ph.D., Prof.,

Mengchao Hepatobiliary Hospital of Fujian Medical University

***** Reviewers' comments *****

Referee #2 (Comments on Novelty/Model System for Author):

The authors have added an HLA-A02:01 model system that adequately establishes the clinical relevance of the proposed approach. Cytolysis study has also been updated to include proper statistical analysis.

Response: Thanks very much for the reviewer's careful reading and the valuable comments.

Referee #2 (Remarks for Author):

*I appreciate the significant efforts the authors have made to address the concerns, refine the manuscript, and incorporate additional experiments to strengthen the study. The update of the article narrative from "universalization" to "generalization" better contextualizes the study in the broader literature of neoantigen research. Furthermore, the inclusion of HLA-A*02:01-restricted epitope validation substantially enhances the translational relevance of their findings. Overall, the revisions and additional experiments sufficiently address my concerns, and I find the manuscript suitable for publication.*

Response: We are sincerely grateful for the reviewer's favorable evaluation of our revision. The highly insightful comments raised by the reviewer have profoundly improved the scientific rigor and clinical relevance of this work.

Referee #3 (Remarks for Author):

The authors have satisfactorily addressed my various concerns.

Response: We deeply appreciate the reviewer's positive assessment of our revision. Your extremely constructive comments have significantly enhanced the scholarly quality of our work.

Dear Dr. Zeljko Durdevic,

Thanks very much for the comprehensive evaluation of our manuscript entitled “Generalizing neoantigen-based tumor vaccine by delivering peptide-MHC complex via oncolytic virus” (Manuscript reference number: EMM-2024-20458-V3). Please accept our apologies for unintentional oversights emerged during the revision process.

We have carefully revised the manuscript according to your feedback, including the addition of “The Paper Explained” section in the main manuscript (Page 1, Line 35-36; Page 2, Line 37-67), renaming of “Materials and methods” (Page 15, Line 590), and the sentence replacement in data availability section (Page 22, Line 880). All modifications are clearly highlighted in red in the revised manuscript for your convenience. A point-by-point response addressing each editorial comment is provided below for your verification. Notably, we confirm our full support for publishing the Review Process File (RPF) alongside the manuscript and have no requests for redaction of figures or content prior to publication.

We remain deeply grateful for your consideration of our work for publication in *EMBO Molecular Medicine* and truly value the time and expertise invested in evaluating our submission.

Sincerely yours,

Xiaolong Liu, Ph.D., Prof.,

Mengchao Hepatobiliary Hospital of Fujian Medical University

***** Editors' comments *****

Q1. Authors: Please provide institutional email address for the co-corresponding author Cuiling Zhang. Also, add institutional e-mail addresses of both corresponding authors on the title page of the manuscript.

Response: Thanks very much for the kind reminder. Accordingly, we have added institutional email addresses of both corresponding authors on the title page of the revised manuscript (Page 1, Line 11-14).

Q2. Author Checklist: Please complete the form by entering information about Author, Journal and Manuscript Number at the top of the page.

Response: Thanks very much for the valuable comments. We have revised the “Author Checklist” accordingly.

Q3. In the main manuscript file, please do the following:

- Please address all comments suggested by our data editors listed below:

o *Figure legends:*

1. Please note that the exact *p* values are not provided in the legends of figures 1E, H, I, M; 2C, 3E, 4C, H; 5B, C, E, G, I; 6L, EV3 F, EV4 A-F.

2. Please note that information related to *n* is missing in the legends of figures 3E, J.

3. Please note that the error bars are not defined in the legends of figures 3E, J; 4A, B, C, D, E, F, H.

- Figure callouts should be in sequential order. Currently, Fig EV2 is called out before Fig EV1C, D. Please correct.

- Author contributions: Please remove it from the manuscript and specify author contributions in our submission system. CRediT has replaced the traditional author contributions section because it offers a systematic machine-readable author contributions format that allows for more effective research assessment. You are encouraged to use the free text boxes beneath each contributing author's name to add specific details on the author's contribution. More information is available in our guide to authors:

<https://www.embopress.org/page/journal/17574684/authorguide#authorshipguidelines>

- Please include structured Methods section that includes a Reagents and Tools Table (should be uploaded as a separate file) followed by a Methods and Protocols section. More information on how to adhere to this format as well as downloadable templates (.docx) for the Reagents and Tools Table can be found in our author guidelines:

<https://www.embopress.org/page/journal/17574684/authorguide#structuredmethods>

An example of a paper with Structured Methods can be found here:

<https://www.embopress.org/doi/full/10.1038/s44320-024-00037-6#sec-4>

- In data availability section please replace the current sentence with following; This study includes no data deposited in external repositories.

Response: Thanks very much for the thorough review of our manuscript. We have carefully revised the manuscript according to the very helpful comments. The concrete revisions have been highlighted in red in the manuscript for your convenience. And a detailed point-by-point response to each comment has been provided below for your reference.

1. The exact p values for the indicated figures have been provided in the figure legends, including 1E (Page 28, Line 1121-1122), 1H-1I (Page 28, Line 1125-1127), 1M (Page 28, Line 1136-1137), 2C (Page 29, Line 1154-1155), 3E (Page 29, Line 1173-1174), 3J (Page 29, Line 1179-1180), 4C (Page 30, Line 1194-1195), 4H (Page 30, Line 1201-1203), 5B (Page 30, Line 1209-1212), 5C (Page 30, Line 1213-1215), 5E (Page 30, Line 1218-1219), 5G-5I (Page 30, Line 1226-1228), 6L (Page 31, Line 1245-1246), EV3F (Page 32, Line 1309-1311), EV4 A-F (Page 33, Line 1321-1325).

2. The information related to n has been added to the legends of figures 3E, J (Page 29, Line 1173; Page 29, Line 1179).

3. The error bars have been defined in the legends of figures 3E, J (Page 29, Line 1181-1182); 4A, B, C, D, E, F, H (Page 30, Line 1204).

4. To ensure that the figure callouts are in sequential order, the description of Fig. 1C was moved up by one paragraph (Page 6, Line 218-220). And Fig. EV1D has been relocated to Fig. EV2 and is now designated as Fig. EV2D in the revised manuscript (Page 6, Line 241; Page 32, Line 1291-1293).

5. Author contributions: We have accordingly removed the section from the manuscript and specified author contributions in the submission system.

6. Structured Methods section: According to the comments, we have uploaded a Reagents and Tools Table in the submission system.

7. Data availability section: Following the comments, we have replaced the sentence with “This study includes no data deposited in external repositories” (Page 22, Line 880).

Q4. Appendix: Please submit the file in PDF format.

Response: Thanks very much for the helpful comments. We have accordingly uploaded a PDF format of “Appendix” in the submission system.

Q5. The Paper Explained: Please add it to the main manuscript file.

Response: Thanks very much for the kind reminder. In response to the comments, we have incorporated “The Paper Explained” into the revised main manuscript, positioned immediately after the “Abstract” section (Page 1, Line 35-36; Page 2, Line 37-67).

Q6. Synopsis:

- Synopsis image: Please format the image to 550 px-wide x (300 - 600)-px high and upload it as a high-resolution JPEG file.

Response: Thanks very much for the helpful comments. Accordingly, we have formatted the “Synopsis Image” to 550 px-wide×420 px-high and uploaded it as a high-resolution JPEG file in the submission system. Meanwhile, the synopsis text and image have been carefully checked before the submission of our revised manuscript.

Q7. As part of the EMBO Publications transparent editorial process initiative (see our Editorial at <http://embomolmed.embopress.org/content/2/9/329>), EMBO Molecular Medicine will publish online a Review Process File (RPF) to accompany accepted manuscripts. This file will be published in conjunction with your paper and will include the anonymous referee reports, your point-by-point response and all pertinent correspondence relating to the manuscript. Let us know whether you agree with the publication of the RPF and as here, if you want to remove or not any figures from it prior to publication. Please note that the Authors checklist will be published at the end of the RPF.

Response: Thanks very much for the kind and prompt reminder. Without doubt, we fully support EMBO Molecular Medicine's efforts to enhance transparency in scientific publishing and hereby agree to the publication of the Review Process File (RPF) alongside our manuscript. At this time, we do not wish to remove any figures or content from the RPF prior to its publication.

Q8. Please provide a point-by-point letter INCLUDING my comments as well as the reviewer's reports and your detailed responses (as Word file).

Response: Thanks very much for the kind and timely reminder. Accordingly, the point-by-point response to editorial comments has been contained in this Word file, while the response to the reviewer's reports has been duly uploaded through the online submission system.

14th Mar 2025

Dear Dr. Liu,

We are pleased to inform you that your manuscript is accepted for publication and is now being sent to our publisher to be included in the next available issue of EMBO Molecular Medicine.

Zeljko Durdevic
Senior Editor
EMBO Molecular Medicine
